# Prosaposin maintains lipid homeostasis in dopamine neurons and counteracts experimental parkinsonism in rodents

Yachao He [1] ✉, Ibrahim Kaya [2], Reza Shariatgorji [2,3], Johan Lundkvist[4,5], Lars U. Wahlberg[4], Anna Nilsson[2,3], Dejan Mamula[1], Jan Kehr[6], Justyna Zareba-Paslawska [1], Henrik Biverstål [5,7], Karima Chergui[8], Xiaoqun Zhang[1], Per E. Andren [2,3] & Per Svenningsson [1,9] ✉

Prosaposin (PSAP) modulates glycosphingolipid metabolism and variants have been linked to Parkinson's disease (PD). Here, we find altered PSAP levels in the plasma, CSF and post-mortem brain of PD patients. Altered plasma and CSF PSAP levels correlate with PD-related motor impairments. Dopaminergic PSAP-deficient (cPSAP^DAT) mice display hypolocomotion and depression/anxiety-like symptoms with mildly impaired dopaminergic neurotransmission, while serotonergic PSAP-deficient (cPSAP^SERT) mice behave normally. Spatial lipidomics revealed an accumulation of highly unsaturated and shortened lipids and reduction of sphingolipids throughout the brains of cPSAP^DAT mice. The overexpression of α-synuclein via AAV lead to more severe dopaminergic degeneration and higher p-Ser129 α-synuclein levels in cPSAP^DAT mice compared to WT mice. Overexpression of PSAP via AAV and encapsulated cell biodelivery protected against 6-OHDA and α-synuclein toxicity in wild-type rodents. Thus, these findings suggest PSAP may maintain dopaminergic lipid homeostasis, which is dysregulated in PD, and counteract experimental parkinsonism.

Parkinson's disease (PD) is a progressive neurodegenerative disorder featured by dopaminergic neuronal loss in the substantia nigra pars compacta (SNc) and accumulation of intracellular α-synuclein (α-syn)-containing Lewy bodies[1]. PD patients are inflicted by motor symptoms and numerous non-motor symptoms[2]. Dopamine (DA) replacement therapies and deep brain stimulation relieve motor symptoms but do not slow down the progressive neurodegeneration[2]. The next breakthrough in the treatment of PD would be a disease-modifying treatment[3].

The pleiotropic protein prosaposin (PSAP) is a secreted neurotrophic factor and a lysosomal protein serving as the precursor of saposins (A-D), which are cofactors for hydrolases of sphingolipids[4,5]. PSAP is decreased, and α-syn is increased in iPSC-derived DA neurons of sporadic PD patients[6]. Variants in the saposin D domain of the PSAP gene are associated with PD in Japanese patients[7], but not in other ethnicities[8–12]. A thorough examination of saposin D-mutated PD patient-derived cells from the Japanese patients reported endoplasmic reticulum PSAP retention but preserved sphingolipid hydrolase

[1]Translational Neuropharmacology, Department of Clinical Neuroscience, Karolinska Institutet, Stockholm, Sweden. [2]Department of Pharmaceutical Biosciences, Medical Mass Spectrometry Imaging, Uppsala University, Uppsala, Sweden. [3]Science for Life Laboratory, Spatial Mass Spectrometry, Uppsala University, Uppsala, Sweden. [4]Division of Neurogeriatrics, Department of Neurobiology, Care Science and Society, Karolinska Institutet, Stockholm, Sweden. [5]Sinfonia Biotherapeutics AB, Huddinge, Sweden. [6]Section of Pharmacological Neurochemistry, Department of Physiology and Pharmacology, Karolinska Institute, Solna, Sweden. [7]Department of Biosciences and Nutrition, Karolinska Institutet, Huddinge, Sweden. [8]Laboratory of Molecular Neurophysiology, Department of Physiology and Pharmacology, Karolinska Institutet, Stockholm, Sweden. [9]Department of Basic and Clinical Neuroscience, Institute of Psychiatry, Psychology and Neuroscience, King's College London, London, UK. ✉e-mail: yachao.he@ki.se; per.svenningsson@ki.se

activity, which is inconsistent with the canonical theory of PSAP and saposins[7]. The trafficking of PSAP is facilitated by forming hetero-dimers with progranulin (PGRN)[13–15]. PGRN is another lysosomal protein and the precursor of granulins (A-G), and PGRN haploinsufficiency causes frontotemporal dementia (FTD)[16,17]. Lentiviral delivery of the PGRN gene protects dopaminergic neurons and counteracts motor dysfunctions in 1-methyl-4-phenyl-1,2,3,6-tetrahydropyridine (MPTP)-induced parkinsonism[18]. Humans and mice with PSAP gene deficiency die prenatally or early postnatally and do not reproduce[19,20]. Interestingly, the first genome-wide CRISPR interference/activation screens to discover susceptibility genes of oxidative stress in induced pluripotent stem cell (iPSC)-derived human neurons identified PSAP. They showed that PSAP loss triggers global lipid alterations and induces ferroptosis selectively in neurons[21].

To improve our understanding of PSAP and PGRN in PD, we analyzed these proteins in SNc, cerebrospinal fluid (CSF), plasma, and leucocytes from PD patients and matched controls. To decipher the role of PSAP specifically in the dopaminergic system, we generated mice with inducible and cell specific PSAP deletion in DA neurons (cPSAP$^{DAT}$). DA neurons degenerate most prominently in PD, while serotonin neurons are relatively preserved. Therefore, we generated mice with PSAP deletion in serotonin neurons (cPSAP$^{SERT}$) to further examine the role of PSAP in a cell population distinct from DA neurons. The mouse models were examined with regards to behavioral, electrophysiological, and neurochemical alterations. In particular, we utilized matrix-assisted laser desorption/ionization-mass spectrometry imaging (MALDI-MSI) to spatially characterize lipidome changes. The vulnerability to AAV-α-syn-induced dopaminergic degeneration was investigated in cPSAP$^{DAT}$ mice. Finally, we examined the therapeutic potential of PSAP delivered by AAV or encapsulated cell biodelivery (ECB) devices against AAV-α-syn-induced parkinsonism in rodents. To examine if PSAP counteracts oxidative stress in DA neurons, we also administered AAV-PSAP to mice exposed to 6-OHDA.

## Results

### PSAP, but not PGRN, is decreased in DA neurons of postmortem SNc from PD patients

In a gross anatomical study, PSAP was previously reported to be not expressed in substantia nigra[22]. However, to our knowledge, PSAP has not been characterized in DA neurons, especially not in those in SNc of PD patients. Here we detected a high level of PSAP in tyrosine hydroxylase (TH) and neuromelanin positive neurons in human SNc (Fig. 1A). We then investigated PSAP and PGRN levels in DA neurons of SNc of PD patients and controls. Immunofluorescent staining on postmortem SNc sections showed prominent co-localization of PSAP and PGRN (Fig. 1A). Importantly, PSAP is reduced, while PGRN is unchanged in individual TH positive neurons in PD compared to controls (Fig. 1A–C). Neuromelanin negatively correlated with TH levels (Supplementary Fig. 1B). PSAP correlated positively with neuromelanin and negatively with TH levels (Supplementary Fig. 1C, D).

### Circulating PSAP correlates with PD motor symptoms, whereas PGRN correlates with non-motor symptoms

Circulating PSAP and PGRN were determined in CSF and plasma from PD patients and controls. PSAP did not differ in CSF but tended to be increased in plasma from PD (Supplementary Fig. 2A, B), while PGRN was decreased in CSF but increased in plasma (Supplementary Fig. 2C, D). Considering the role of PGRN in FTD, PD patients were classified into two groups, PD with mild cognitive impairment (PD-MCI) and PD with normal cognition (PD-NC), according to a MoCA cut-off score of 25[23]. Increased PSAP and PGRN were found in plasma from PD-MCIs compared to controls (Supplementary Fig. 2E, F). Since it is known that PSAP and PGRN form complexes, we determined the PSAP-PGRN complex levels in both CSF and plasma. The levels of plasma complexes were not changed in PD (Supplementary Fig. 2G). However, by

stratifying PD patients, we found significantly increased complex levels in PD-MCI, compared to PD-NC (Supplementary Fig. 2H). The PSAP-PGRN complex is undetectable in CSF. CSF PSAP correlated with plasma PSAP, but no such correlation was found for PGRN (Fig. 2D, E). PSAP positively correlated with PGRN in both CSF and plasma (Supplementary Fig. 2I, J). As a disease control study, we measured CSF PSAP levels in normal pressure hydrocephalus (NPH) patients, which is characterized by cognitive impairment. PSAP levels were not changed in the CSF of NPH patients (Supplementary Fig. 2K). The neurofilament light chain (NfL), a sensitive indicator of neuroaxonal damage, serves as a biomarker of PD. To investigate the relationship of PSAP and PGRN with NfL, we measured NfL in the CSF of PD patients. PSAP did not correlate, while PGRN showed a mild correlation, with NfL (Supplementary Fig. 2L, M). To partially explain the origin of their plasma circulating contents, PSAP and PGRN levels were also determined in peripheral blood mononuclear cells (PBMCs) by flow cytometry (Supplementary Fig. 3). Both PSAP and PGRN were enriched in monocytes (Supplementary Fig. 2N, O). An overall decrease of PSAP, which was significant in CD8$^+$ T cells, was found in PBMCs from PD (Supplementary Fig. 2P), while an increase of PGRN in classical monocytes was found in PD (Supplementary Fig. 2Q).

Regarding clinical symptoms and signs, both CSF and plasma PSAP correlated positively with scores of UPDRS-III, the gold standard rating scale for PD motor symptoms (Fig. 1F, G). However, PSAP levels failed to correlate with non-motor symptom scores obtained with the depression rating scales BDI-II, MADRS-S, HADS-D, the anxiety rating scale HADS-A, and the fatigue rating scale MFS (Fig. 1H, I, Supplementary Table 3). Conversely, CSF and plasma PGRN did not correlate with UPDRS-III scores (Fig. 1J, K) but correlated negatively with the aforementioned non-motor symptom scores (Fig. 1L, M, Supplementary Table 3), especially in PD-MCIs (Supplementary Table 3). These findings suggest that PSAP and PGRN are differently regulated in PD and that PSAP is associated with cardinal motor features of PD.

### Dopaminergic PSAP-deficient mice present reduced levels of dopaminergic markers, dysfunctional striatal synaptic plasticity and behavioral impairments, while serotonergic PSAP-deficient mice behave normally

To study PSAP in the dopaminergic system, we generated a mouse line with inducible PSAP gene deletion specifically in DA neurons (cPSAP$^{DAT}$). We made mice homozygous for floxed PSAP allele (Supplementary Fig. 4A) and crossed them with heterozygous mice expressing CreER$^{T2}$ recombinase under the regulatory sequence of the dopamine transporter (DAT) (Supplementary Fig. 4B). Tamoxifen was administered to 5w-old mice, and deletion of PSAP in DA neurons across the whole SNc was confirmed by fluorescent in situ hybridization (FISH) analysis of PSAP mRNA (Fig. 2A, Supplementary Fig. 4C). To examine neurotransmitter alterations, high-performance liquid chromatography (HPLC) was applied. DA, homovanillic acid (HVA), and 3-methoxytyramine (3-MT) were significantly reduced in the striatum, nucleus accumbens, and hippocampus of 16m-old cPSAP$^{DAT}$ mice (Fig. 2B, Supplementary Fig. 5A, B). Through autoradiographic detection, we observed a reduction of DAT at 4 m but not at later time points (Supplementary Fig. 5C, D). Meanwhile, by densitometry analysis, we noted a mild and progressive TH loss in the striatum of cPSAP$^{DAT}$ mice (Fig. 2C, D, Supplementary Fig. 5E). To study the functional consequences of reduced striatal DA, we performed electrophysiological experiments to measure long-term potentiation (LTP) in the striatum. Striatal LTP involves DA and is an indicator of synaptic plasticity[24]. No LTP could be evoked in striatal slices of cPSAP$^{DAT}$ mice, and LTP could not be restored by the pretreatment with SKF38393, an agonist acting on postsynaptic D1 receptors (Supplementary Fig. 5F–H). This indicates that presynaptic dopaminergic and/or a non-dopaminergic component of striatal LTP is deficient in cPSAP$^{DAT}$ mice. Taken

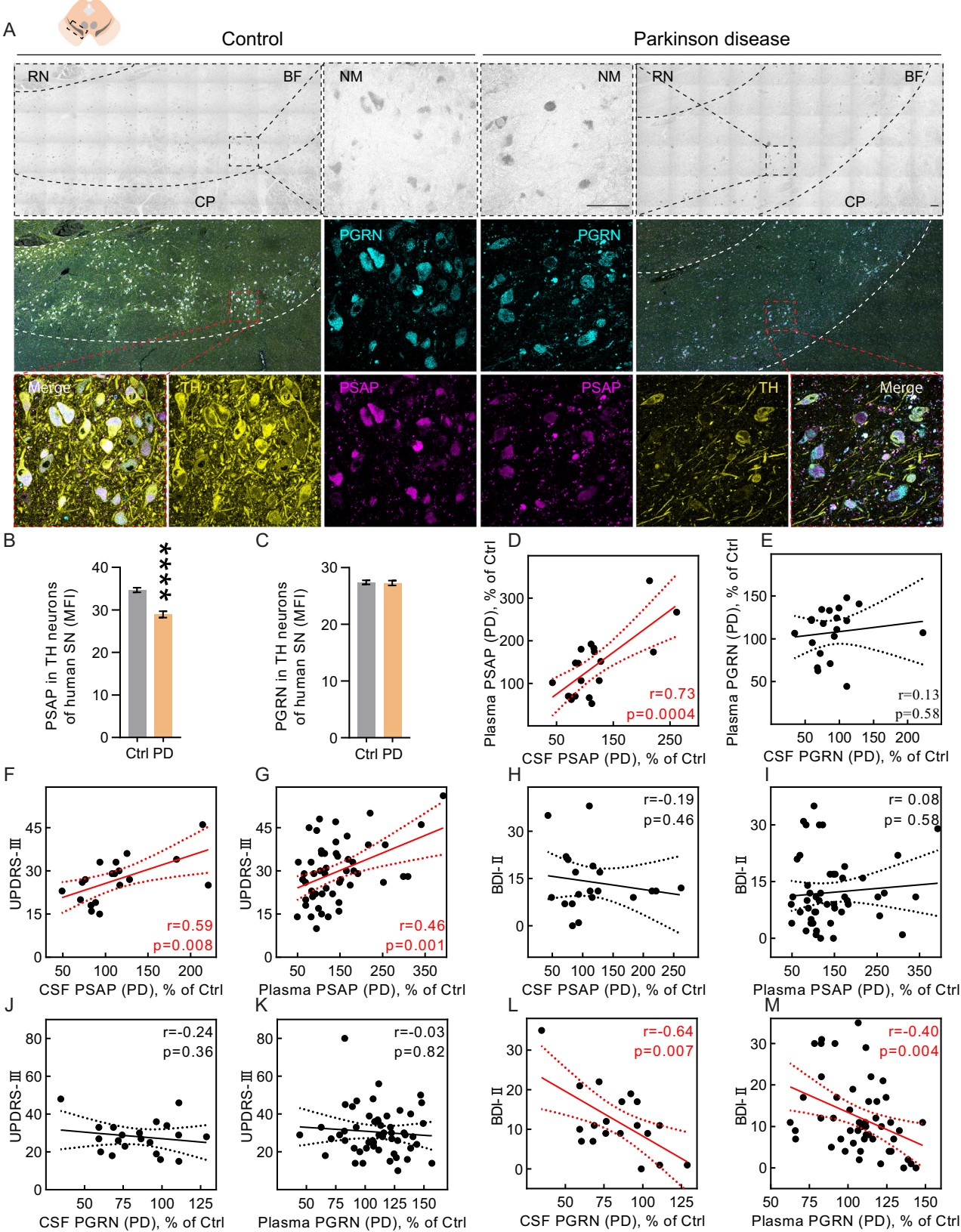

together, our data demonstrate that cPSAP^DAT mice have mild impairments of dopaminergic neurotransmission.

To study whether the harmful role of ablating PSAP in DA neurons is cell-specific, we decided to delete PSAP in another cell type. Immunofluorescent stainings on mouse brain sections showed that PSAP and PGRN strongly co-express in neurons, including DA, serotonin, and

cortical neurons (Supplementary Fig. 6). PSAP is lowly expressed in glia and striatal medium spiny neurons, while PGRN is high in microglia, but low in astrocytes, oligodendrocytes, and striatal medium spiny neurons (Supplementary Fig. 6A–D). Similarly, PSAP expression in human cortex is mainly neuronal (Supplementary Fig. 6E, F). Since serotonin neurons express very high PSAP levels and are relatively preserved at

**Fig. 1 | PSAP and PGRN in PD patients are divergently regulated and associated with different PD symptoms. A** Representative PSAP and PGRN immuno-fluorescent staining in postmortem substantia nigra sections from four PD patients and four controls. BF bright field, RN red nucleus, CP cerebral peduncle, NM neuromelanin. Scale bars, 100 μm. **B, C** Quantification of mean immunofluorescence intensity (MFI) of PSAP and PGRN staining in TH positive neurons (n = 959 and 430 neurons from four controls and four PD patients, respectively). Data are presented as mean ± S.E.M. Student's t-test (**B**), or Welch's t-test (**C**) is applied. Non-significant p value is not labeled, ****p < 0.0001. **D, E** Scatter plots representing the associations of CSF PSAP with plasma PSAP (**D**) and CSF PGRN with plasma PGRN (**E**). Each point depicts a CSF PSAP or PGRN value and the corresponding plasma PSAP or PGRN value of one PD patient, respectively. N = 19, 20 in (**D, E**), respectively.

**F–I** Scatter plots representing associations of CSF or plasma PSAP with scores of UPDRS-III or BDI-II. Each point depicts CSF or plasma PSAP values and the corresponding score of UPDRS-III or BDI-II of a PD patient. N = 19, 54, 20, 50 in (**F–I**), respectively. **J–M** Scatter plots representing associations of CSF or plasma PGRN with scores of UPDRS-III or BDI-II. Each point depicts the CSF or plasma PGRN value and the corresponding score of UPDRS-III or BDI-II of one PD patient. N = 20, 55, 19, 50 in (**J–M**), respectively. Pearson correlation coefficients (r) and p-values are calculated in (**D, E, G, H, J, L**), and nonparametric Spearman correlation r and p-values are calculated in (**F, I, K, M**). The solid and dashed lines indicate the simple linear regression line and the 95% confidential interval (CI), respectively. UPDRS-III Unified Parkinson's disease Rating Scale-III, BDI-II Beck Depression Inventory-II.

early stages of PD[25,26], we explored the role of PSAP in serotonin neurons. For this purpose, mice homozygous for floxed PSAP alleles were crossed with heterozygous mice expressing Cre recombinase under the regulatory sequence of the serotonin transporter (SERT) to generate mice with PSAP gene deletion in serotonin neurons (cPSAP$^{SERT}$) (Supplementary Fig. 4B). Unfortunately, mice with inducible Cre under the SERT promotor were not available to us. It should therefore be noted that the inducible DAT-CreER$^{T2}$ model deleted PSAP during puberty, whereas the SERT-Cre model deleted PSAP during embryonic development and germline deletion may lead to compensations that mask the normal adult role of PSAP. Nevertheless, the FISH analysis confirmed the specific deletion of PSAP mRNA in serotonin neurons of cPSAP$^{SERT}$ mice (Fig. 2E). In contrast to cPSAP$^{DAT}$ mice, no changes in 5-hydroxytryptamine (5-HT) and 5-hydroxyindoleacetic acid (5-HIAA) were detected by HPLC in the hippocampus of cPSAP$^{SERT}$ mice (Fig. 2F, G).

Since PSAP can be taken up by cells[4], we examined PSAP protein with immunofluorescent staining in the two mouse lines. PSAP was dramatically diminished in DA neurons of cPSAP$^{DAT}$ mice at 2 m, which is 2w after tamoxifen treatment, but accumulation occurred over time (Fig. 2H, I, Supplementary Fig. 7A). Being a binding partner of PSAP, PGRN followed a similar trajectory (Fig. 2H, J, Supplementary Fig. 7A). Combining FISH and immunofluorescent staining, we confirmed the presence of PSAP protein in the absence of PSAP mRNA in DA neurons of cPSAP$^{DAT}$ mice (Supplementary Fig. 7B). Similarly, PSAP and PGRN levels were initially reduced and subsequently restored in serotonin neurons of cPSAP$^{SERT}$ mice (Supplementary Fig. 7C–E). It has been reported that both full-length PGRN and cleaved granulins can be detected by the antibody used in these studies[27]. To investigate whether both full-length PSAP and cleaved saposins have been detected, we studied the antibody specificity by knocking-down PSAP using siRNA in mouse N2a cells. Western blot analysis showed different processed forms of PSAP, which were knocked down by PSAP-siRNA (Supplementary Fig. 7F).

Behavioral assessments were performed in cPSAP$^{DAT}$ mice at 4 m, 8 m, and 16 m of age (Fig. 2K). Hypolocomotion in the open field test was observed in 8m- and 16m-old cPSAP$^{DAT}$ mice (Fig. 2L). Impairments of postural control and fine movements were found in the pole test in 4m-old cPSAP$^{DAT}$ mice (Fig. 2M, Supplementary Fig. 8A) and in the beam traversal test in 16m-old cPSAP$^{DAT}$ mice (Fig. 2N, Supplementary Fig. 8B). However, cerebellum-related coordination and balance remained intact in the accelerating rotarod test (Supplementary Fig. 8C). Increased anxiety, measured as time in the center zone of the open field test (Fig. 2O) and distance and time in the light box of a light-dark transition test (Fig. 2P, Q), was found in cPSAP$^{DAT}$ mice. In accordance with a high co-morbidity of anxiety and depression[28], 8m-old cPSAP$^{DAT}$ mice also showed a depressive-like state in the forced swim test (Fig. 2R). On the contrary, 8m-old cPSAP$^{SERT}$ mice were assessed by a battery of behavioral tests but showed no abnormalities (Fig. 2S–W).

We next tested the locomotor responsiveness of cPSAP$^{DAT}$ mice to the treatment of dopaminergic stimulants, including l-3,4-dihydroxyphenylalanine (L-dopa)/benserazide, cocaine, and SKF81297 (Supplementary Fig. 8D). Agreeing with previous studies in wild-type (WT) mice or mice with mild dopaminergic deficiency[29,30], we found hypolocomotion of both WT and cPSAP$^{DAT}$ mice upon L-dopa treatment. In contrast, cocaine induced hyperlocomotion both in WT and cPSAP$^{DAT}$ mice, which was significantly stronger in WT mice. Since cocaine acts at dopaminergic terminals, these experiments, together with the baseline data, provide further evidence that dopamine neurons per se are mildly dysfunctional in cPSAP$^{DAT}$ mice. SKF81297, a D1 agonist stimulating striatal dopaminoceptive postsynaptic neurons, significantly increased locomotion in both genotypes with a similar potency (Supplementary Fig. 8D).

## Spatial lipidomics unveils accumulation of highly unsaturated and shortened lipids along with reduction of sphingolipids throughout the brain of cPSAPDAT mice

Discernibly, the behavioral deficiencies and nigrostriatal electrophysiological malfunction observed in cPSAP$^{DAT}$ mice could not be wholly ascribed to the mild loss of dopaminergic markers. PSAP is cleaved to saposins in the lysosome, which, in turn, regulate a subset of lysosomal sphingolipid related hydrolases[5]. Specifically, saposin A modulates β-galactosylceramidase (GALC); saposin B modulates GM1-β-galactosidase, neuraminidase (Neu), α-galactosidase A (GLA), β-galactosidase (GLB), and arylsulfatase A (ASA); saposin C modulates GLB and glucocerebrosidase (GBA); saposin D modulates ceramidase; saposin D has also been shown to modulate sphingomyelinase (SMase)[31]. Based on this, PSAP loss may trigger accumulation in gangliosides (especially GM1 and GM3), globosides (especially Gb3), lactosylceramide (LacCer), glucosylceramide (GlcCer), galactosylceramide (GalCer), sulfatides, and sphingomyelins (SMs). There may also be a decrease of ceramide due to reduced activity of glycolipids pathways. Because of saposin D deficiency a decrease of sphingosine (Sph) may also be found.

We initially performed a targeted analysis of GlcCer, GalCer, GlcSph, and GalSph levels, but found them to be unchanged in the striatum of cPSAP$^{DAT}$ mice (Supplementary Fig. 9A–D). Considering the important roles of saposins in the metabolism of many glycosphingolipids (GSLs) and since alterations in other lipids than GSLs have been reported in iPSCs lacking PSAP[21], we turned to a high-throughput dual polarity MALDI-MSI method. We conducted an untargeted spatial lipidome analysis in cPSAP$^{DAT}$ and control mouse brains. This investigation uncovered significant differences between 4m-, 8m- and 16m-old cPSAP$^{DAT}$ mice and their controls, as shown by principal component analysis (PCA) in the caudate-putamen, cerebral cortex, and several other brain regions in the striatal and SNc level (Fig. 3A, Supplementary Fig. 9E–G). Volcano plots of all identified lipids, including glycerolipids and sphingolipids, showed similar alteration patterns among brain regions of 4m-, 8m-, and 16m-old cPSAP$^{DAT}$ mice (Fig. 3B, Supplementary Fig. 9H–J, Supplementary data file 1). PCA and subsequent one-way analysis of variance (ANOVA) of principal components revealed longitudinal progression in the magnitude of lipid alterations in all regions (Supplementary Fig. 9K).

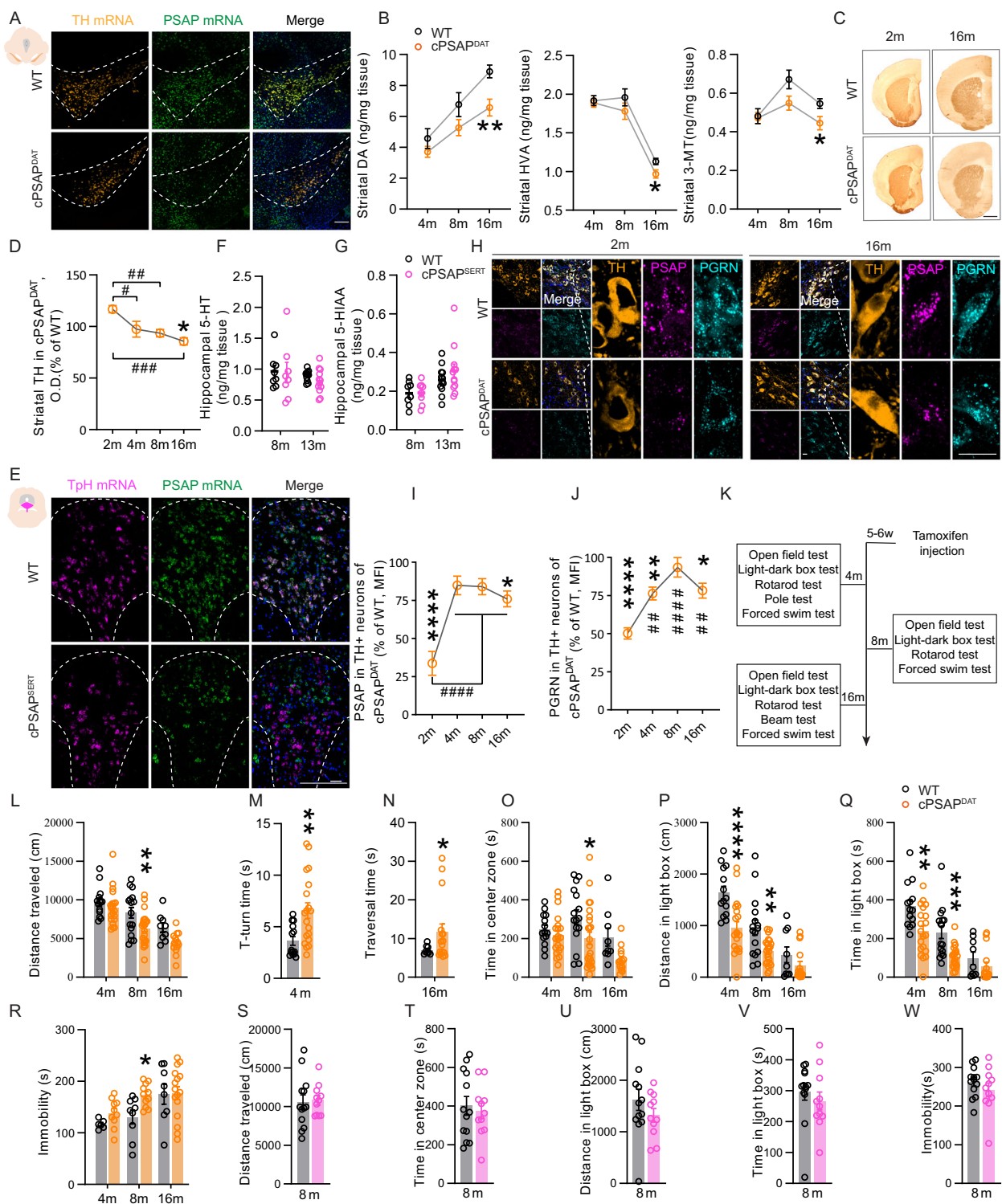

As there were bidirectional changes in glycerolipids, we further analyzed them in the caudate-putamen of 16 m-old mice (Fig. 3C). Cardiolipins (CLs) and phosphatidylcholines (PCs) were chosen for the analysis as they were the most frequently identified. By sorting CLs according to their double bonds and chain length, two patterns were captured in cPSAP^DAT mouse brains: (i) Increases of highly unsaturated CLs coincided with decreases of lowly unsaturated CLs, and (ii) Decreases of long-chain CLs coincided with increases of short-chain CLs (Fig. 4D–G). Similar patterns of change were found in the PCs (Supplementary Fig. 10A–D). The first pattern indicates hyperactive desaturation of fatty acid and resembles the pattern identified in cells

treated with rotenone[32], while the second pattern suggests a hyperactive peroxisomal β-oxidation. It has been reported that desaturation of poly unsaturated fatty acids (PUFA) to highly unsaturated fatty acid (HUFA) is involved in the glycolytic NAD$^+$ recycling (Supplementary Fig. 10E)[32]. Thus, we measured the NAD$^+$/NADH ratio in striatal tissue of cPSAP^DAT and control mice and found the ratio to be significantly increased in the caudate-putamen of cPSAP^DAT mice (Supplementary Fig. 10F). To better understand molecular alterations underlying the global lipid changes, we measured the levels of several key enzymes. FADS-1 is one of the key desaturases for PUFAs and was upregulated in DA neurons of cPSAP^DAT mice (Supplementary Fig. 10G, H). By

**Fig. 2 | cPSAP^DAT mice show reduced levels of dopaminergic markers and behavioral deficiencies, while cPSAP^SERT mice behave normally. A** Fluorescence in situ hybridization (FISH) images of TH (orange) and PSAP (green) mRNA in substantia nigra pars compacta/ventral tegmental area (SNc/VTA) of wild-type (WT) and cPSAP^DAT mice. **B** Line charts showing HPLC measurements of DA, HVA, and 3-MT in the striatum of WT and cPSAP^DAT mice of 4m-, 8m- and 16m-old. $N_{WT}$ = 7, 7, 9, $N_{cPSAP}^{DAT}$ = 10, 10, 15, respectively. **C** TH immunohistochemical staining in striatal sections of 2m- and 16m-old WT and cPSAP^DAT mice. **D** Graph showing densitometry analysis of TH staining in striatal sections of 2m-, 4m-, 8m- and 16m-old WT and cPSAP^DAT mice. $N_{WT}$ = 7, 7, 7, 9, $N_{cPSAP}^{DAT}$ = 6, 10, 10, 16, respectively. **E** Representative FISH images of tryptophan hydroxylase (Tph) (magenta) and PSAP (green) mRNA in dorsal raphe nucleus (DRN) of WT and cPSAP^SERT mice. **F, G** Dot plots showing HPLC measurements of 5-HT (**F**), 5-HIAA (**G**) in the hippocampus of WT and cPSAP^SERT mice of 8m- and 13m-old. $N_{WT}$ = 8, $N_{cPSAP}^{SERT}$ = 12. **H** TH (orange), PSAP (magenta), PGRN (cyan) immunofluorescent staining in substantia nigra of WT and cPSAP^DAT mice of 2m- and 16m-old. Left panels are low-magnification images; right panels are high-magnification images. **I, J** Graphs showing MFI quantification of PSAP (**I**) and PGRN (**J**) in TH positive neurons in the SNc of WT and cPSAP^DAT mice of 2m-, 4m-, 8m-, and 16m-old. N = 48, 70, 46, 51 cells from N = 4, 5, 3, 5 WT mice respectively, and N = 56, 67, 85, 63 cells from N = 4, 5, 5, 5

cPSAP^DAT mice respectively. **K** Timeline of behavioral tests in cPSAP^DAT and control mice. **L** Graph representing distance traveled in open field test by WT and cPSAP^DAT mice of 4m-, 8m-, and 16m-old. $N_{WT}$ = 14, 16, 9, $N_{cPSAP}^{DAT}$ = 20, 25, 15, respectively. **M** Graph showing T-turn time in pole test by WT and cPSAP^DAT mice of 4m-old. $N_{WT}$ = 14 and $N_{cPSAP}^{DAT}$ = 19. **N** Bar graph representing traversal time in beam traversal test by WT and cPSAP^DAT mice of 16m-old. $N_{WT}$ = 8 and $N_{cPSAP}^{DAT}$ = 16. **O–R** Graphs representing time in the center zone in the open field test (**O**), distance and time in light box in light-dark transition test (**P, Q**), and immobility time in forced swim test (**R**) by WT and cPSAP^DAT mice of 4m-, 8m-, and 16m-old. $N_{WT}$ = 14, 16, 9 and $N_{cPSAP}^{DAT}$ = 20, 23-24, 15-16 in (**O–Q**), $N_{WT}$ = 6, 9, 8 and $N_{cPSAP}^{DAT}$ = 10, 11, 16 in (**R**). **S–W** Graphs showing distance traveled and time in the center zone in open field test (**S, T**), distance and time in light box in light-dark transition test (**U**, V), and immobility time in forced swim test (**W**) by WT and cPSAP^SERT mice of 8m-old. $N_{WT}$ = 13, $N_{cPSAP}^{SERT}$ = 11 in (**S–W**). Scale bars, 200 μm (**A, E**), 1 mm (**C**), and 20 μm (**H**). Data are presented as mean ± S.E.M. Student's *t*-test (**B, S–U, W**), Mann–Whitney test (**M, N, V**) or two-way ANOVA with Bonferroni's (**F, G, I, J, L, O–R**) or Fisher's LSD (**D**) post hoc test was applied appropriately. * Compared to WT, # compared to 2 m cPSAP^DAT; non-significant *p* values are not labeled, */#$p$ < 0.05, **/##$p$ < 0.01, ***/###$p$ < 0.001, ****/####$p$ < 0.0001.

definition, the desaturation of saturated fatty acid (SFA) to mono-unsaturated fatty acid (MUFA) could not be ruled out from the causes of accumulation of highly unsaturated lipids in cPSAP^DAT mouse brains. We investigated the levels of SCD-1, a desaturase for SFA to MUFA, in DA neurons of cPSAP^DAT mouse brains, and found it to be unchanged (Supplementary Fig. 10I, J). As for the hyperactive peroxisomal β-oxidation in cPSAP^DAT mice, we measured levels of PEX14 and ACOX-1, key marker of peroxisomes and enzyme involved in peroxisomal β-oxidation respectively, in DA neurons. However, both of them remained unchanged (Supplementary Fig. 10K).

We detected several other sphingolipids than the GSLs mentioned above in the spatial lipidome analysis and found a ubiquitous reduction in cPSAP^DAT mouse brains (Fig. 3H). Specifically, palmitoylcarnitine, originated from the precursor of ceramide-palmitic acid, was reduced (Fig. 3I). Phosphosphingolipids, phosphorylated derivatives of ceramide, including ceramide-1-phosphate (CerP), sphingomyelins (SMs), and Na + /K+ adducts of SMs, were also reduced (Fig. 3I, Supplementary Fig. 10L). Likewise, GM1s, GM2s, GM3s (Fig. 3I), and GDs (Supplementary data file 1) were generally reduced. However, regarding sulfatides, some polyunsaturated ones (SHexCer(t42:2), SHexCer(t43:2), SHexCer(t44:2)) were increased, while some monounsaturated ones (SHexCer(d40:1), SHexCer(d36:1)) were decreased, especially in the cortex, which were consistent with the above identified desaturation pattern (Supplementary data file 1). To see if PSAP deletion impaired lysosomal function, we measured LAMP-1 and cathepsin D (CTSD) levels in DA neurons, and found that LAMP1 was unchanged, while CTSD was reduced in cPSAP^DAT mice, suggestive of malfunction of lysosomes (Supplementary Fig. 10K).

To examine whether levels of key sphingolipid metabolism-related enzymes (Supplementary Fig. 10M) in DA neurons underlie the abovementioned changes in sphingolipids, their levels were measured. We found that UDP-glucose ceramide glucosyltransferase-1 (UGCG-1) was dramatically decreased, while GALC and GLB-1 were mildly increased (Supplementary Fig. 10N). However, GBA, beta-hexosaminidase A (HEX-A), sphingomyelin synthase 1 (SGMS-1), and sphingomyelin phosphodiesterase 1 (SMPD-1) were unchanged in DA neurons of cPSAP^DAT mice (Supplementary Fig. 10N). Since iPSCs lacking PSAP have increased lipofuscin-like granules, we examined such species but found no changes in cPSAP^DAT mice (Supplementary Fig. 10O, P).

To better understand the widespread lipid changes in cPSAP^DAT mouse brains, we also examined lipids using MALDI-MSI in the cerebellum, a region less innervated by DA neurons, of 4m-old cPSAP^DAT mice. FISH analysis confirmed that PSAP was not deleted in this

region (Supplementary Fig. 11A). The lipids changed in the cerebellum (26 out of 136 detected lipids) was much fewer than those in the striatum (105 out of 182 detected lipids), as shown by PCA analysis (Supplementary Fig. 11B) and in the volcano plot (Supplementary Fig. 11C, Supplementary data file 2). Most of the significantly changed lipids in the striatum were also identified in the cerebellum (88 out of 105), but only around a quarter of them (24 out of 88) were significantly changed (Supplementary data file 2). Specifically, lipids changed in the striatum, e.g., SM(36:1), CerP(36:1), CL(74:7), and CL(74:8) were changed in the same direction (Supplementary Fig. 11D, E). However, other lipids, e.g. CL(72:4) and CL(72:9), which were changed in striatum, were unaltered in cerebellum (Supplementary Fig. 11F).

To explore if acute DA neuronal loss can trigger similar lipid alterations as found in the cPSAP^DAT mouse brains, we performed unilateral 6-hydroxydopamine (6-OHDA) lesion in the medial forebrain bundle (MFB) of WT mice (Supplementary Fig. 12A) and examined several lipids in the caudate-putamen. We specifically identified and analyzed several lipids that were either increased (CL(72:9)), decreased (CL(76:9), GM1(36:1), GM1(38:1)), or unchanged (CL(74:9)) in cPSAP^DAT mouse brains. Only CL(74:9) and CL(76:9) displayed a mild decrease in 6-OHDA lesioned mouse brains (Supplementary Fig. 12B, C), which was different from lipid alterations in cPSAP^DAT mice. These data suggests that the lipid alterations found in the cPSAP^DAT mouse brains are not secondary to a DA neuronal loss.

## Spatial lipidomics uncovers confined accumulation of ganglio-sides and increased tryptophan metabolism in the dorsal raphe nucleus of cPSAP^SERT mouse brains

Since cPSAP^SERT mice showed a contrasting lack of behavioral abnormalities and neurotransmitter alterations compared to cPSAP^DAT mice, we examined if the lipid changes would display corresponding cell type-specific differences. The lipidome analysis revealed no differences in the hippocampus, caudate-putamen, and corpus callosum between cPSAP^SERT mice and controls, as shown by PCA (Supplementary Fig. 13A). However, in the cortex and dorsal raphe nucleus (DRN), cPSAP^SERT mice and controls could be separated by principal component 2 (Supplementary Fig. 13A). Volcano plots of all identified lipids found few alterations in the aforementioned brain regions of cPSAP^SERT mice, except some potential differences in the DRN (Supplementary Fig. 13B, Supplementary data file 3). Thus, we performed non-corrected multiple *t*-tests to examine the lipid changes specifically in the DRN (Fig. 3J). In contrast to cPSAP^DAT mice, a general increase of GSLs, including GM1s, GM2s, and GM3s, was unveiled in the DRN of

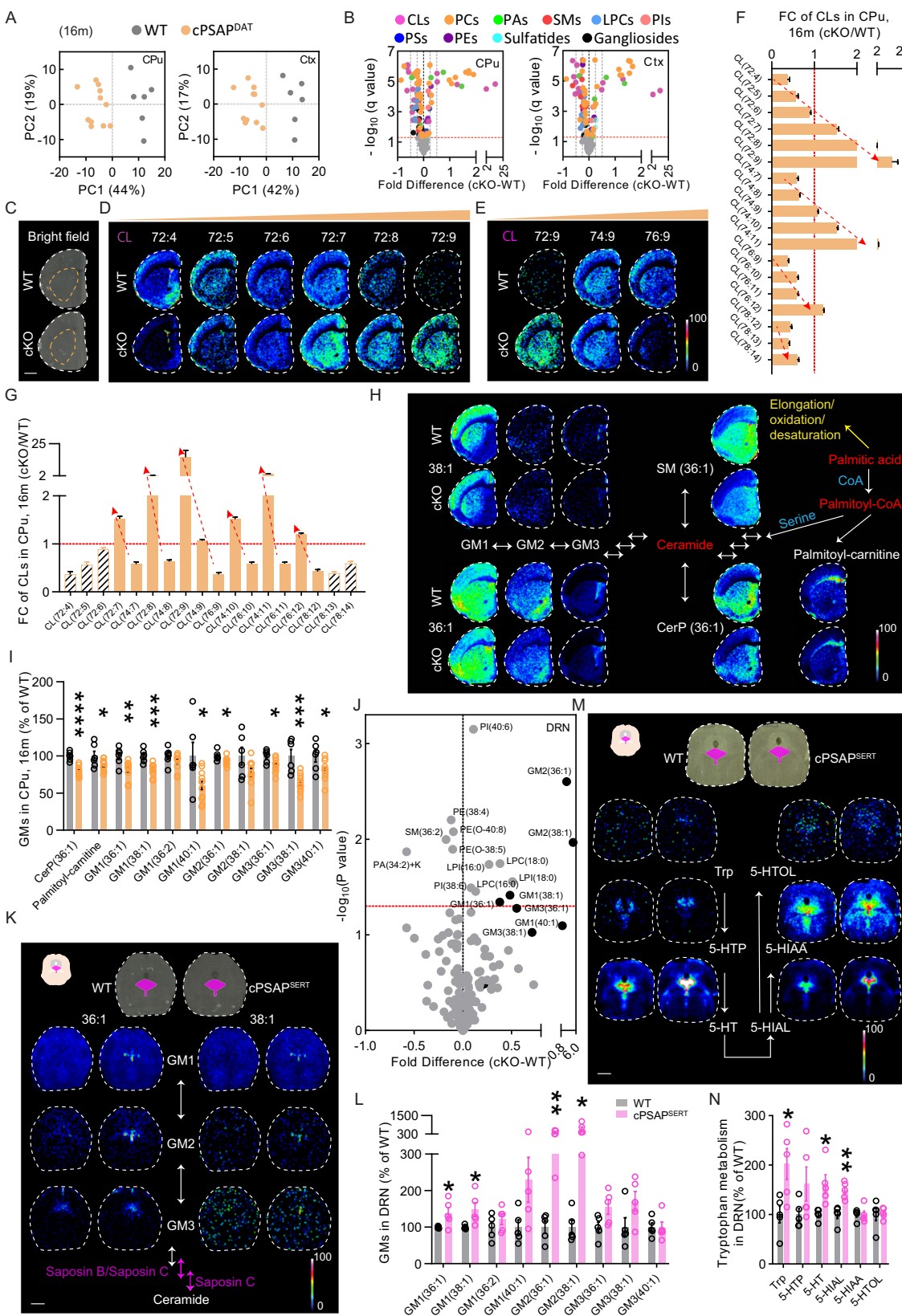

cPSAP[SERT] mice, which agreed with the canonical role of PSAP (Fig. 3K, L).

Although 5-HT and 5-HIAA were unchanged in the hippocampus of cPSAP[SERT] mice as determined by HPLC (Fig. 2F, G), while considering the selective accumulation of GMs in DRN, we wondered how neurotransmitters, especially 5-HT and their metabolites, were

affected in the DRN. As MALDI-MSI can also spatially profile neurotransmitters in a relatively small region, we used this technique to profile neurotransmitters in the DRN. We detected 5-HT and its metabolites and several other neurotransmitters and molecules in the DRN, cortex, striatum, and hippocampus of cPSAP[SERT] mice and controls (Supplementary data file 4). Neurotransmitters, including 5-HT

**Fig. 3 | cPSAP$^{DAT}$ mice display increased highly unsaturated and shortened lipids along with reduction of sphingolipids throughout the brain, while cPSAP$^{SERT}$ mice present confined accumulation of gangliosides and increased tryptophan metabolism in the dorsal raphe nucleus. A** Score plots presenting the first and second principal components (PC1 and PC2) generated by the principal component analysis (PCA) of all annotated lipids in the caudate-putamen (CPu) and cortex (Ctx) of 16m-old WT (gray) and cPSAP$^{DAT}$ (orange) mice. Each point depicts one biological replicate. **B** Volcano plots showing fold differences and the minus logarithm of $q$ value ($-\log_{10}$ ($q$ value)) of all detected lipids in the CPu and Ctx of two genotypes. The red dash line represents FDR ($q$) = 5%. Lipids regulated with a false discovery rate (FDR, $q$) <5% are highlighted with colors. Each dot depicts one lipid. **C** Images showing the analyzed striatal sections in bright field with dash lines delineating CPu. **D**, **E** Ion images (16 m) of cardiolipin (CL) peaks, sorted by numbers of double bonds (**D**) or chain length (**E**). **F**, **G** Graphs showing CLs arranged by double bonds (**F**) or chain length (**G**) with fold changes (FC) The red dashed arrows denote highly unsaturated (**F**) and shortened CLs (**G**)., N$_{cPSAP}^{DAT}$ = 11 for 16m-old mice. **H** Ion images of sphingolipids and their precursors. The white arrows denote the metabolic pathways. **I** Graph showing ceramide-1-phosphate (CerP), palmitoyl-carnitine, and gangliosides in the CPu of 16m-old WT and cPSAP$^{DAT}$ mice. N$_{WT}$ = 6,

N$_{cPSAP}^{DAT}$ = 11. **J** Volcano plot showing the indicated fold differences and -log$_{10}$ ($p$-value) of all annotated lipids in dorsal raphe nucleus (DRN) of 8m-old WT and cPSAP$^{SERT}$ mice. Non-corrected multiple $t$ tests. The red dash line represents $p$ = 0.05. Lipids regulated with $p < 0.05$ are labeled. All gangliosides are in black. **K** Ion images of ganglioside peaks in the DRN level of the midbrain of two genotypes. The top two images (left, WT; right, cPSAP$^{SERT}$) show the analyzed sections in bright field with DRN marked in magenta. The white arrows denote the metabolic pathways. Magenta arrows are metabolic pathways modulated by saposins. **L** Graph showing gangliosides in DRN of two genotypes. Student's t-test. Each circle represents one mouse. N$_{WT}$ = 5, N$_{cPSAP}^{SERT}$ = 5. **M** Ion images of tryptophan and metabolite peaks in DRN level of midbrain of two genotypes. **N** Graph showing tryptophan and metabolites in DRN of two genotypes. N$_{WT}$ = 5, N$_{cPSAP}^{SERT}$ = 5. Scale bars, 1 mm; MALDI-MS ion images are shown using rainbow scale (scaled to 100% of max ion intensity scale) for visualization; data are presented as mean ± S.E.M. Student's $t$-test (**I**, **L**, **N**). *$p < 0.05$, **$p < 0.01$, ***$p < 0.001$, ****$p < 0.0001$. PCs phosphatidylcholines, PAs phosphatidic acids, SMs sphingomyelins, LPCs lyso-PCs, PIs phosphatidylinositols, PSs phosphatidylserines, PEs phosphatidylethanolamines, GMs monosialogangliosides.

and other molecules, were unchanged in the cortex and hippocampus of cPSAP$^{SERT}$ mice, which agreed with our HPLC data (Supplementary Fig. 13C). However, tryptophan, 5-HT, and 5-HIAL were significantly upregulated in the DRN of cPSAP$^{SERT}$ mice, indicating increased tryptophan metabolism (Fig. 3M, N)[33]. In addition, spermidine was significantly increased, while dihydroxyphenylethylene glycol (DOPEG), a metabolite of norepinephrine (NE), was significantly decreased (Supplementary Fig. 13C).

It is known that, apart from protein synthesis, a majority of tryptophan is directed to the kynurenine pathway, the de novo pathway of NAD$^+$ synthesis (Supplementary Fig. 14A)[34]. Since tryptophan metabolism was increased in the DRN of cPSAP$^{SERT}$ mice, we wondered if de novo synthesis of NAD$^+$ was upregulated. IDO-1 and TDO-2 are two key rate-limiting enzymes of the de novo pathway. By immunofluorescent staining, we revealed that both enzymes were upregulated in TPH$^+$ neurons in the DRN of cPSAP$^{SERT}$ mice (Supplementary Fig. 14B–E), indicating a boosted de novo synthesis of NAD$^+$. To examine the de novo pathway in cPSAP$^{DAT}$ mice, we also measured IDO-1 and TDO-2 levels in DA neurons. Intriguingly, IDO-1 is not expressed in DA neurons (Supplementary Fig. 14F), and TDO-2 is expressed in DA neurons but was not changed in cPSAP$^{DAT}$ mice (Supplementary Fig. 14G, H).

### cPSAP$^{DAT}$ mice show elevated vulnerability to α-syn-induced dopaminergic degeneration, while viral overexpression of PSAP protects mice from α-syn toxicity

Accumulating data has shown that lipids are critically involved in α-syn aggregation and neurotoxicity[35] and the formation of Lewy bodies[36]. Particularly, it has recently been reported that unsaturated fatty acids increase α-syn toxicity[37]. Since we found major increases in unsaturated fatty acids in cPSAP$^{DAT}$ mice, we wanted to investigate the role of PSAP in α-syn toxicity. We, therefore, examined whether PSAP deletion affects the vulnerability of DA neurons to α-syn overexpression and whether any α-syn toxicity could be counteracted by PSAP overexpression. To achieve this, 4m-old cPSAP$^{DAT}$ and WT control mice received unilateral stereotaxic intranigral injections of AAV-α-syn or AAV-GFP with or without AAV-PSAP (Supplementary Fig. 15A).

To evaluate striatal DA receptor sensitization caused by α-syn-mediated DA loss, apomorphine rotation tests were conducted at 6w, 10w, and 14w after surgery (Fig. 4A). Repeated measures (RM) two-way ANOVA of net contralateral rotations revealed that cPSAP$^{DAT}$ mice deteriorated more than WT mice when injected with AAV-α-syn but not with AAV-GFP (Fig. 4B, Supplementary Fig. 15B). Replenishing PSAP by injecting AAV-PSAP prevented the worsening effect of AAV-α-syn on rotational behavior in cPSAP$^{DAT}$ mice (Fig. 4B). Furthermore, AAV-PSAP injection rendered AAV-GFP-injected cPSAP$^{DAT}$ mice to rotate

ipsilaterally, reflecting a rescue effect of PSAP overexpression on DA neurons lacking PSAP (Supplementary Fig. 15B).

Immunofluorescent and immunohistochemical stainings of post-mortem mouse brains confirmed robust overexpression of α-syn and GFP in striatum and substantia nigra (Fig. 4C, E, Supplementary Fig. 15C, 15E). Immunofluorescent staining also showed that AAV-α-syn diminished PSAP in TH-positive neurons (Fig. 4C, D), while AAV-GFP did not (Supplementary Fig. 14C, D), which was in accordance with our result in DA neurons of PD patients (Fig. 1A, B) and PSAP downregulation in PD iPSC-derived DA neurons presenting α-syn accumulation[6]. PSAP level in TH-positive neurons of cPSAP$^{DAT}$ mice was lowered but not absent, replicating our observation that PSAP protein from neighboring cells is taken up by DA neurons in cPSAP$^{DAT}$ mice (Fig. 2H, I, Supplementary Fig. 7A). Densitometry analysis of immunohistochemical staining showed that ipsilateral striatal TH was reduced by AAV-α-syn, but not by AAV-GFP, in both genotypes (Fig. 4E, F). However, AAV-α-syn caused a significantly more prominent striatal and nigral TH loss in cPSAP$^{DAT}$ mice than in WT mice (Fig. 4E, F). Furthermore, in agreement with the behavioral data, significant restoration of striatal and nigral TH by AAV-PSAP overexpression was revealed by comparing AAV-α-syn-injected with AAV-α-syn+AAV-PSAP-injected cPSAP$^{DAT}$ mice (Fig. 4E, F). DAT not only manifested similar restorative changes as TH but was also normalized by AAV-PSAP overexpression in AAV-α-syn-injected WT mice (Fig. 4E, F). Phosphorylated α-syn Ser129 (p-Ser129 α-syn) is considered as the pathological form of α-syn and prone to aggregate[38]. To investigate if PSAP affects levels of p-Ser129 α-syn in α-syn-overexpressed mouse brains, we stained p-Ser129 α-syn in these mouse brain sections. AAV-α-syn overexpression leads to mild p-Ser129 α-syn accumulations in WT mice but could not be reversed by AAV-PSAP overexpression (Fig. 4G, Supplementary Fig. 15F). While in cPSAP$^{DAT}$ mouse brains, p-Ser129 α-syn accumulations were dramatically increased and reversed to similar level as WT mice by AAV-PSAP overexpression (Fig. 4G, Supplementary Fig. 15F). Only taking p-Ser129 α-syn accumulations with an area larger than 0.64 μm$^2$ into account[39], similar results were obtained (Fig. 4H, Supplementary Fig. 15F). To examine the resistance of p-Ser129 α-syn accumulation to proteinase K (PK) treatment in AAV-α-syn over-expressing cPSAP$^{DAT}$ mice, we pretreated striatal sections with PK solution (10 μg/ml) and performed immunostaining. PK digested most of the small accumulations but spared some large accumulations, indicating possible aggregation of p-Ser129 α-syn in these mice (Supplementary Fig. 15G).

### Viral overexpression of PSAP protects mice from 6-OHDA induced dopaminergic degeneration

PSAP has been identified as susceptibility gene of oxidative stress in iPSC-derived human neurons[21]. We investigated if viral overexpression

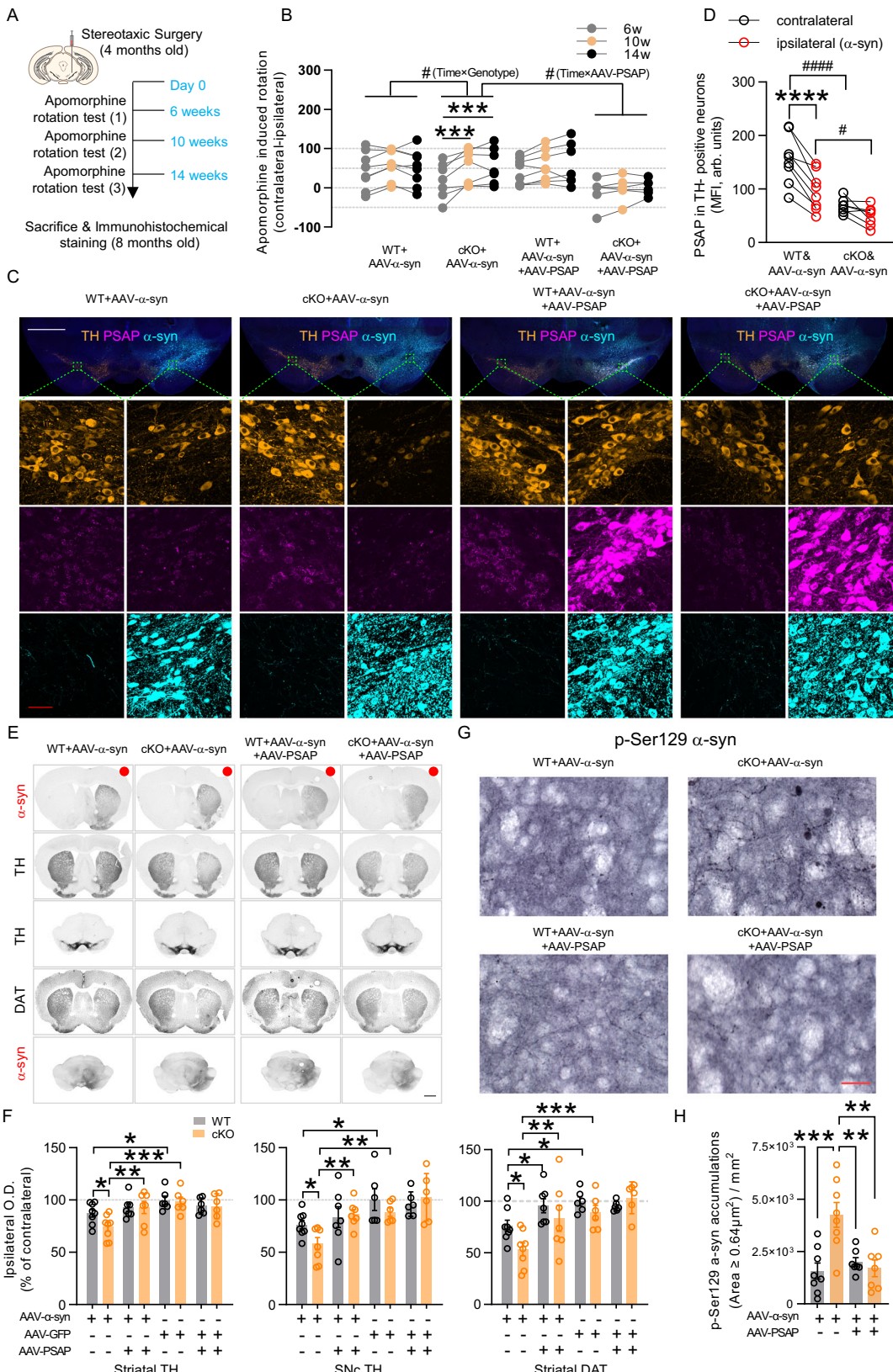

of PSAP could protect DA neurons from 6-OHDA-induced oxidative stress. Six weeks before the 6-OHDA striatal partial lesion, WT mice were injected with AAV-PSAP or AAV-GFP in the ipsilateral SNc, followed by apomorphine rotation test two weeks after the lesion to monitor the protective effect of PSAP (Fig. 5A). AAV-PSAP-injected mice rotated to the contralateral side significantly less than AAV-GFP-

injected mice, indicative of protected striatal dopamine innervation from 6-OHDA lesion by PSAP overexpression (Fig. 5B). Immuno-fluorescent staining confirmed overexpression of GFP or PSAP in mice and revealed partially preserved DA neurons in the SNc (Fig. 5C). Densitometry analysis of immunohistochemical staining showed significantly reduced striatal and nigral TH and striatal DAT

**Fig. 4 | cPSAP<sup>DAT</sup> mice are more vulnerable to AAV-α-synuclein-induced toxicity, while AAV-PSAP intranigral injection counteracts the toxicity by reducing p-Ser129 α-syn levels. A** Schematic representation of unilateral stereotaxic surgery (upper left) and timeline of experiments. **B** Quantification of apomorphine-induced net contralateral rotation (contralateral-ipsilateral) of AAV-α-syn injected mice with or without AAV-PSAP at 6w (gray), 10w (orange), and 14w (black). Each dot depicts one mouse. $N = 8, 7, 8, 6$, mice in four groups, respectively. Repeated measures (RM) two-way ANOVA with Bonferroni's post hoc test was applied; *compared to 6w, # interactions. **C** Representative images of TH (orange), PSAP (magenta), and α-syn (cyan) immunofluorescent staining on postmortem substantia nigra sections of AAV-α-syn-injected mice with or without AAV-PSAP. Top panel, low-magnification (scale bar=1000 μm) images of the whole substantia nigra. Bottom panels, high-magnification (scale bar=50 μm) images of TH neurons. **D** Quantification of mean fluorescence intensity (MFI) of PSAP staining in TH-positive neurons of AAV-α-syn-injected WT and cPSAP<sup>DAT</sup> mice. Black and red circles depict contralateral and ipsilateral PSAP immunoreactivity, respectively. $N_{WT} = 8$, $N_{cPSAP^{DAT}} = 7$. RM two-way

ANOVA with Bonferroni's post hoc test; *compared to contralateral, # compared to cPSAP<sup>DAT</sup>. **E** Representative images of α-syn (red dots indicate the injection side) and TH immunohistochemical staining in striatal and substantia nigra sections, and DAT staining in striatal sections of AAV-α-syn-injected mice. Scale bar, 1 mm. **F** Densitometry analysis of ipsilateral TH immunoreactivity in the striatum and SNc, and ipsilateral DAT immunoreactivity in the striatum of all groups of mice. Values are normalized to the mean of their corresponding contralateral immunoreactivity. Each circle represents one mouse. $N = 8, 8, 7, 6, 6, 6, 6$ mice in eightgroups, respectively. Two-way ANOVA with Fisher's LSD post hoc test. **G** Representative images of p-Ser129 α-syn immunohistochemical staining (enhanced by nickel) in striatal sections of AAV-α-syn-injected mice. Scale bar, 25 μm. **H** Quantification of number of p-Ser129 α-syn accumulations (area ≥ 0.64 μm²) in striatal sections of AAV-α-syn-injected mice; $N = 8, 8, 7, 7$mice in four groups, respectively; One-way ANOVA with Bonferroni's post hoc test. Data are presented as mean ± S.E.M. */#$p < 0.05$, **$p < 0.01$, ***$p < 0.001$, ****/####$p < 0.0001$.

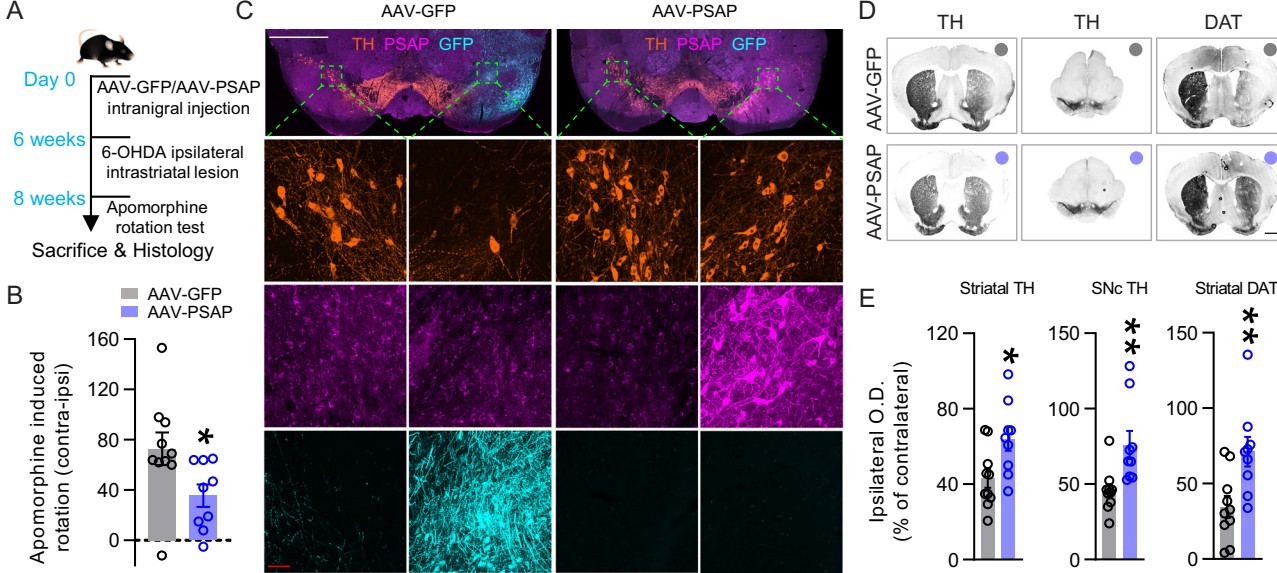

**Fig. 5 | Viral overexpression of PSAP protects mice from 6-OHDA induced dopaminergic degeneration. A** Timeline of AAV injection, 6-OHDA striatal lesion, and apomorphine rotation test. **B** Quantification of apomorphine-induced net contralateral rotation of AAV-GFP or AAV-PSAP injected WT mice. $N = 10$, 99-10 mice in two groups, respectively. **C** Representative images of TH (orange), PSAP (magenta), and GFP (cyan) immunofluorescent staining on postmortem substantia nigra sections of 6-OHDA-lesioned mice injected with AAV-GFP or AAV-PSAP. Top panel, low-magnification (scale bar = 1000 μm) images of the whole substantia nigra. Bottom panels, high-magnification (scale bar = 50 μm) images of TH neurons.

**D** Representative images of TH immunohistochemical staining in striatal and substantia nigra sections, and DAT staining in striatal sections of 6-OHDA-lesioned mice injected with AAV-GFP or AAV-PSAP. Scale bar, 1 mm. **E** Densitometry analysis of ipsilateral TH immunoreactivity in the striatum and SNc, and ipsilateral DAT immunoreactivity in the striatum of all groups of mice; Values are normalized to their corresponding contralateral immunoreactivity; Each circle represents one mouse; $N = 10$, 9 mice in two groups, respectively. Student's $t$-test. Data are presented as mean ± S.E.M. *$p < 0.05$, **$p < 0.01$.

immunoreactivity by 6-OHDA in AAV-GFP-injected mice, which were partially protected in AAV-PSAP-injected mice (Fig. 5D, E).

### Encapsulated cell biodelivery of PSAP in the striatum protects against α-syn-induced dopaminergic loss in rats

Since PSAP and PGRN are secreted proteins[4,40], and both were taken up in DA neurons in cPSAP<sup>DAT</sup> mice, we reasoned that extracellular delivery of PSAP may counteract α-syn-induced toxicity. Encapsulated cell biodelivery (ECB) provides an effective and reversible means to deliver therapeutic agents to a targeted area of brain parenchyma by grafting an ECB device[41]. Using this technique, we unilaterally delivered PSAP to the striatum of rats injected with AAV-α-syn in SNc of the same hemisphere (Fig. 6A). Since neuronal uptake of PSAP is facilitated by PGRN[13,14], we examined PSAP alone but also in complex with PGRN and also included a group of PGRN alone in this experiment.

Devices were generated with clonal ARPE-19 cell lines over-expressing PSAP, PGRN, and PSAP-PGRN (Supplementary Fig. 16A).

Normal ARPE-19 cells were used to generate control devices (Supplementary Fig. 16A). The secretion capabilities of the devices were determined by ELISAs (Supplementary Fig. 16B–D). An ELISA method for monitoring PGRN-PSAP complexes was developed, and ECB-PGRN and ECB-PGRN-PSAP were found to secrete PGRN-PSAP complexes. In contrast, ECB-PSAP devices secreted only free PSAP (Supplementary Fig. 16B–D). Size exclusion chromatography of culture medium from ECB-PGRN and ECB-PGRN-PSAP cells (Supplementary Fig. 16E) and subsequent ELISAs of the sampled protein-containing fractions showed a high level of free PGRN and low level of PGRN-PSAP complexes secreted by ECB-PGRN cells and exclusive high level of PGRN-PSAP complexes secreted by ECB-PGRN-PSAP cells (Supplementary Fig. 16F, G). After the stereotaxic surgeries with simultaneous intranigral AAV-α-syn injection and striatal ECB device implantation, locomotive behaviors were assessed in the open field test at 2w, 8w, and 12w, followed by an apomorphine rotation test at 12w (Fig. 6B).

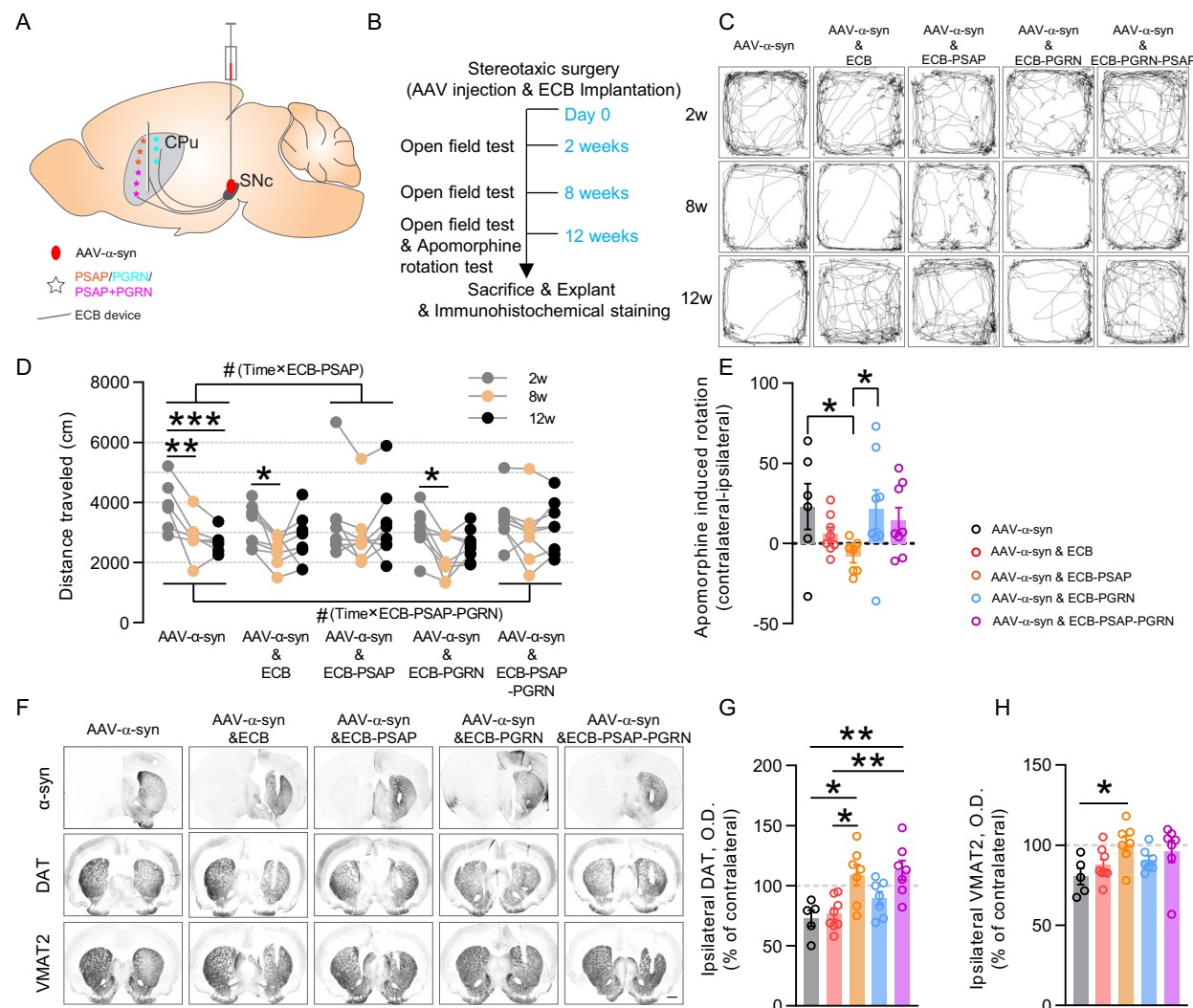

**Fig. 6 | Encapsulated-cell biodelivery of PSAP protects against AAV-α-synuclein-induced parkinsonism in rats. A** Schematic representation of simultaneous unilateral intranigral AAV-α-syn injection and ipsilateral striatal ECB device implantation. **B** Timeline of experiments. **C** Representative tracking images of all groups of rats (AAV-α-syn, AAV-α-syn & ECB, AAV-α-syn & ECB-PSAP, AAV-α-syn & ECB-PGRN, and AAV-α-syn & ECB-PSAP-PGRN) in the open field test at 2w, 8w, and 12w. **D** Quantification of distance traveled in the open field test of all groups of rats at 2w (gray), 8w (orange), and 12w (black). $N = 6, 8, 8, 8, 8$ rats in five groups, respectively. Repeated measures two-way ANOVA with Bonferroni's post hoc test was applied; *compared to 2w, # interactions. **E** Quantification of apomorphine-induced net contralateral rotation of all groups of rats. $N = 6, 8, 7, 8, 8$ rats in five groups, respectively. Kruskal–Wallis test followed by Dunn's *post-hoc* test was applied. **F** Representative images of α-syn (top), DAT (middle), and VMAT2 (bottom) immunohistochemical staining in striatal sections of all groups of rats. Scale bar, 1 mm. **G, H** Densitometry analysis of ipsilateral DAT (**G**) and VMAT2 (**H**) immunoreactivity in the striatum of all groups of rats. Values are normalized to the mean value of their corresponding contralateral immunoreactivity. $N = 6, 8, 7, 7, 7$ rats in five groups, respectively. One-way ANOVA with Bonferroni's post hoc test was applied in (**G**), and Kruskal–Wallis test followed by Dunn's post hoc test was applied in (**H**). Data in (**E, G, H**) are presented as mean ± S.E.M. */#$p < 0.05$, **$p < 0.01$, ***$p < 0.001$.

RM two-way ANOVA analysis of distance traveled in an open field revealed that rats without devices suffered from AAV-α-syn-induced progressive hypolocomotion, and rats with control ECB or ECB-PGRN showed hypolocomotion at 8w. In contrast, rats with ECB-PSAP or ECB-PSAP-PGRN displayed intact locomotion (Fig. 6C, D). The significant interactions of time×ECB-PSAP and time×ECB-PSAP-PGRN demonstrated that PSAP and PSAP-PGRN complex replenishment protected rats from α-syn-induced hypolocomotion (Fig. 6C, D). No differences in thigmotaxis, an anxiety indicator, were found among groups (Supplementary Fig. 16H). In the apomorphine rotation test, rats with ECB-PSAP rotated significantly less to the contralateral side than rats without devices and rats with ECB-PGRN, providing further evidence for a protective effect of PSAP against α-syn-induced locomotor disorder (Fig. 6E).

Postmortem immunohistochemical staining validated overexpression of α-syn in the striatum and substantia nigra (Fig. 6F, Supplementary Fig. 16I). To examine α-syn-induced effects on

dopaminergic markers, immunohistochemical staining of TH, DAT, and vesicle monoamine transporter-2 (VMAT2) were conducted. In accordance with the fact that RPE cells secrete neurotrophic factors and L-dopa[42], control ECB devices and other devices significantly counteracted striatal TH loss (Supplementary Fig. 16I, J). However, ECB-PSAP and ECB-PSAP-PGRN devices also displayed a protective effect on DAT levels compared to rats without devices and rats with control ECB (Fig. 6F, G). Moreover, rats with ECB-PSAP devices showed preserved VMAT2 compared to rats without devices (Fig. 6F, H).

## Discussion

Human genetic studies have highlighted critical functions of PGRN and PSAP in neurodegeneration[7,40]. Discrepancies about the linkage of PSAP variants with PD risk exist and may be caused by ethnic heterogeneity and/or age of the studied populations[8–12]. The evidence of saposin D-related PD in Japanese population and the reduction of PSAP

revealed in iPSC-derived DA neurons with α-syn accumulation from PD patients indicate a potential role of PSAP in PD pathogenesis[6,7]. Accumulating evidence from studies points towards a functional co-dependence of PSAP and PGRN, especially under stressful conditions[14]. However, the roles of PSAP and PGRN and their relationship in DA neurons and PD remain poorly investigated. Unexpectedly, we found differential expression and regulation of PSAP and PGRN in samples from PD patients and controls. PSAP is reduced, while PGRN is unchanged in surviving DA neurons of PD patients. The peripheral levels of PSAP, but not PGRN, correlate with its central levels. Furthermore, when correlated to clinical symptoms of PD, circulating PSAP differs from PGRN. Circulating PSAP correlates mainly with motor symptoms. Conversely, circulating PGRN correlates exclusively with non-motor symptoms, especially depression, of which the mechanism warrants further study. The neuron-dominant expression pattern of PSAP and microglia-dominant expression pattern of PGRN may contribute to their different associations with PD. Although CSF or plasma PSAP does not distinguish PD from healthy controls, they may indicate PD disease progression. Therefore, these results suggest a more prominent role of PSAP than PGRN in modulating core motor symptoms of PD.

Our studies in cPSAP^DAT and cPSAP^SERT mice revealed significant differences between these two mouse lines both in behavioral repertoires and neurotransmitter levels, with cPSAP^SERT mice being relatively unaffected. However, it should be emphasized that a direct comparison of the data between these mouse lines should be done cautiously as the inducible DAT-CreER^T2 model involves PSAP deletion in puberty, whereas the SERT-Cre model deletes PSAP during embryonic development enabling compensations. Even though the dopaminergic degeneration is relatively mild, cPSAP^DAT mice still show significant behavioral deficiencies and impaired striatal neuroplasticity. DAT levels were initially decreased at 4 m but got restored at later timepoints. This could have been caused by accumulation of PSAP and saposins in DA neurons over time, Moreover, we believe that part of these functional deficiencies in the cPSAP^DAT mice may relate to lipid metabolism alterations. Spatial lipidomics in cPSAP^DAT mouse brains revealed increased highly unsaturated lipids containing poly/highly unsaturated fatty acids (PUFAs/HUFAs), indicative of hyperactive fatty acid desaturation, and increased short-chain lipids containing short fatty acyl chains, suggesting excessive fatty acid peroxisomal β-oxidation. Indeed, the composition of lipids with fatty acyl chains, including but not limited to CLs, PCs, PAs, PIs, PSs, PEs, and their lyso-species, is dramatically changed in cPSAP^DAT mouse brains. Despite that PSAP was only deleted in DA neurons, these lipid changes occurred in many brain regions. However, in the cerebellum, which is less innervated by DA neurons, these changes were less pronounced. The phenomenon of widespread lipid changes needs further investigation but might involve a direct "lipid horizontal flux" between cells or be indirectly mediated by for example "diffusible metabolic factors" predisposing cells to changes in lipid composition. Similar mechanisms have been suggested for the pathogenic spread of tau protein aggregation[43]. Surprisingly, sphingolipids underwent a universal decrease in cPSAP^DAT mouse brains. We found increased levels of enzymes (GALC and GLB-1) for degradation and decreased levels of enzyme (UGCG-1) for synthesis of glycosphingolipids, which may partially explain general decrease of glycosphingolipids. The hyperactive desaturation and peroxisomal oxidation patterns may also apply to palmitic acid and thereby reduce the capacity of ceramide synthesis, contributing to the general decrease of sphingolipids[44]. However, the levels of GlcCer, GalCer, GlcSph, and GalSph, which are intimately modulated by saposins[4], are unchanged. It indicates that their reduction by decreased de novo synthesis and increased degradation might be counteracted by their accumulation due to the lack of saposins. Acute DA neuronal loss induced by 6-OHDA did not trigger similar lipid alterations as in cPSAP^DAT mice, indicating a minor contribution of loss

of dopaminergic neurons to the changes of lipids caused by PSAP deletion. It should be noted that although MALDI-MSI provided spatial information of lipid changes, liquid chromatography (LC)-MS-based lipidomics would be complementary for the analysis of a more comprehensive and high-throughput lipidomics analysis of mouse brain tissue[45,46]. Further, when coupled with LC and ion mobility or applied with chemical derivatization strategies[46,47], electrospray ionization (ESI) can provide specific information on the isomeric and isobaric species, which would enhance the biological interpretation of the lipidomics analysis.

In accordance with the behavioral data, the spatial lipidomic study in cPSAP^SERT mouse brains revealed much fewer and completely different alterations than in cPSAP^DAT mouse brains. Consistent with the canonical role of PSAP, a general increase of gangliosides was found in cPSAP^SERT mice. Moreover, these mice displayed increased tryptophan metabolism and serotonin-synthesis function, specifically in the DRN. As a majority of tryptophan is dedicated to the de novo pathway of NAD^+ synthesis, we investigated this pathway and found that the two rate-limiting enzymes, IDO-1 and TDO-2, were increased in serotonin neurons of cPSAP^SERT mice. However, IDO-1 is not expressed, while TDO-2 stayed unchanged, in DA neurons of cPSAP^DAT mice. Despite of this, NAD^+/NADH ratio was still increased in the striatum of cPSAP^DAT mice, which can be explained by the glycolytic NAD^+ recycling mediated by hyperactive PUFA desaturation[32]. It indicates that desaturation-mediated NAD^+ recycling compensated for the shortage of de novo synthesis of NAD^+ in DA neurons of cPSAP^DAT mice. Consistently, we found increased levels of FADS-1, key enzyme in PUFA desaturation, in DA neurons of cPSAP^DAT mice. It has also been reported that the activity of PUFA desaturases can be increased by inhibition of aerobic respiration without increasing protein levels of these desaturases[32]. Meanwhile, PSAP deficiency suppresses glycolysis and oxidative phosphorylation[48], and PSAP overexpression is sufficient to boost oxidative metabolism[49]. Thus, PSAP deficiency causes hyperactive PUFA desaturation in cPSAP^DAT mice probably by impairing mitochondria function. To this end, the difference of de novo pathway of NAD^+ synthesis between DA neurons and serotonin neurons contributes to the difference in lipidomes of cPSAP^DAT and cPSAP^SERT mice.

The lipidome data in the cPSAP^DAT mice are interesting in relation to PD. PUFAs have been reported to be increased in PD[50] and show high binding affinity to α-syn[35,51]. The exposure to free or phospholipid-bound PUFAs causes structural changes of α-syn and facilitates its oligomerization[52–55]. PUFAs are particularly vulnerable to oxidation, and the likelihood of peroxidation of a PUFA is proportional to its number of double bonds[56,57]. PUFA peroxidation products also promote the production of toxic oligomers of α-syn[58]. Although SCD-1 levels were unaltered, it could not be ruled out that its activity was increased, in DA neurons of cPSAP^DAT mice. A previous study has shown that, by inhibiting SCD-1, lipid desaturation is reduced, and consequently the toxicity of α-syn is alleviated in human-derived neurons[37]. When applied to 3 K (E35K + E46K + E61K) α-syn mice, an SCD1 inhibitor reduces α-syn hyperphosphorylation and improves behavior performance[59]. Based on these data, an SCD1 inhibitor, YTX-7739, is currently being tested in a clinical trial with PD patients[60]. Iron-dependent lipid peroxidation, a feature of ferroptosis, has been reported in iPSC-derived neurons specifically sensitized by PSAP deletion[21]. Lipofuscin-like granules were observed in these PSAP-deficient neurons[21]. Considering the vulnerability of PUFAs to peroxidation, we examined the levels of lipofuscin-like granules in cPSAP^DAT mice but found no significant change. It is noteworthy that increases in iron levels and lipid peroxidation, essential for lipofuscin formation, were observed under antioxidant-free conditions in PSAP-deficient neurons[21]. However, in vivo environment is different from the antioxidant-free culture medium, which might be the reason why we could not detect lipofuscin formation. Agreeing with the dysregulated peroxisomal activity in cPSAP^DAT mice, ether lipids, the

peroxisome-derived glycerophospholipids, were found to be enriched in iPSC-derived neurons lacking PSAP[21]. Canonically, peroxisome shortens very long-chain fatty acids (>22). With mitochondria impaired or overloaded, peroxisome can also oxidize medium- and long-chain fatty acids[61]. A previous study has shown that PSAP knockdown dampens mitochondria respiratory function[48]. The levels of the peroxisomal proteins PEX14 and ACOX-1 were unchanged in DA neurons of cPSAP$^{DAT}$ mice. However, the excessive fatty acid peroxisomal β-oxidation could be a result of increased enzyme activities, which requires additional analyses. Intriguingly, short-chain lipids have also been reported to promote α-syn aggregation[62,63]. Hence, PSAP may play a role in the pathogenesis of PD.

Experiments using intranigral overexpression of AAV-α-syn demonstrated that PSAP is not only involved in the basal physiological functions of DA neurons but also increases the DA neuron vulnerability towards α-syn toxicity when deleted. Indeed, cPSAP$^{DAT}$ mice injected with AAV-α-syn showed significantly more severe impairments in rotational behavior and reduced dopamine marker staining compared to control mice. Moreover, the downregulation of PSAP levels by α-syn overexpression implies a vicious circle that might aggravate the disease severity. Neuroprotective effects of PSAP-derived peptides (prosaptides) have been reported in MPTP-induced acute PD models that do not mimic the progressive feature of PD[64,65]. Here we extend these studies by showing that AAV-mediated overexpression of full-length human PSAP protects DA neurons from α-syn toxicity and rescues behavioral deficits of cPSAP$^{DAT}$ mice. Consistent with aforementioned relationship of lipids and PD, we revealed higher levels of p-Ser129 α-syn in the striatum of AAV-α-syn-injected cPSAP$^{DAT}$ mice, compared to WT mice, which was alleviated by the overexpression of PSAP. In WT mice, PSAP overexpression did not reverse the levels of p-Ser129 α-syn, which might relate to that p-Ser129 α-syn induced by α-syn overexpression in WT mice was mild. However, reduced DAT levels by α-syn overexpression in WT mice were rescued by PSAP overexpression, indicating that other factors involved in the toxicity induced by α-syn overexpression have been reversed. It is well documented that α-syn induces oxidative stress[66,67]. PSAP-deficient iPSC-derived human neurons are more susceptible to oxidative stress[21]. Here we extend these findings by showing that PSAP overexpression partially protects mice from 6-OHDA-induced dopaminergic degeneration. Although the 6-OHDA-induced model does not mimic the progressive nature of PD and does not recapitulate several molecular and cellular aspects of PD pathogenesis, it is a potential tool to study oxidative stress in DA neurons. Taken together, PSAP may offer protection both by reducing p-Ser129 α-syn and counteracting oxidative stress.

The re-uptake of PSAP by DA neurons in the cPSAP$^{DAT}$ mice motivated us also to deliver PSAP via ECB in the striatum. Studies have shown that extracellular PSAP can be transported to the endosome/lysosome through the mannose-6-phosphate receptor or through sortilin-1 by binding to PGRN[14]. We, therefore, included PGRN and the PGRN-PSAP complex in this set of experiments. The ECB technique is a therapeutic platform being developed to treat neurodegenerative diseases[41]. Interestingly, RPE cell grafting was intensively investigated for the treatment of PD, which failed probably due to the lack of biological activity of grafted RPE[68–70]. The encapsulation of cells facilitates continuous secretion of therapeutic factors in a prolonged duration[41]. While ECB-based delivery of PGRN did not result in any behavioral improvement or neuroprotective effect, we found that the ECB delivery of PSAP and the PSAP-PGRN complex both counteracted behavior impairments and dopaminergic loss induced by intranigral AAV-α-syn overexpression. Due to the multifaceted nature of cell-based delivery systems, it is not certain whether the protective effect of PSAP or PSAP-PGRN complex is direct or depending on factors released by RPE cells. Indeed, an interplay between PSAP and neurotrophic factors and/or L-dopa secreted from the ECB system could also take part in the neuroprotection. In any case, PSAP-PGRN complex did not display any advantage over PSAP alone. Future studies are needed to elucidate the half-life of PSAP protein. It should be noted that the cleavage of PSAP into saposins varies between cell types and more studies are needed to specifically determine the processing of PSAP in DA neurons[71].

Mutations in GBA1, which encodes β-glucocerebrosidase, are the most common genetic risk factor for PD, but gene carriers still show a low penetrance to develop PD[72]. PSAP mutations have been reported to modify the risk of GBA-induced PD[73]. It has been proposed that the three lysosomal proteins: PSAP, PGRN, and Pro-cathepsin D (Pro-CTSD), form a lysosomal network contributing to GBA-induced PD[14]. Hence, therapeutic studies of PSAP in PD models with GBA deficiency will be interesting to perform in the future.

In conclusion, our experiments demonstrate that PSAP levels are altered in PD patients reflecting motor symptoms. Studies with PSAP deletion in DA neurons reveal the role of PSAP in physiological homeostasis of DA neurons, especially in lipid metabolism. PSAP deletion in DA neurons increases the vulnerability of mice to α-syn overexpression. Notably, α-syn-induced parkinsonism could be rescued by intranigral AAV-PSAP injection. Likewise, AAV-PSAP protected against 6-OHDA-induced oxidative stress and DA neurotoxicity. Furthermore, striatal ECB-PSAP implantation shows therapeutic effects in treating α-syn-induced parkinsonism. Thus, PSAP is a potential modifier of pathogenesis of PD and its replenishment may be beneficial in halting PD progression.

## Methods

### Human samples

All human experiments were conducted according to the Declaration of Helsinki. Formalin-fixed paraffin-embedded (FFPE) human nigra sections of 5 μm thick were obtained from the brain bank of Karolinska Institutet. Pre-mortem informed consents were signed by all donors (four PD patients and four controls without neurological disorders), with approval from the regional ethics review board of Stockholm (2014/1366-31). Demographic and clinical information are provided in Supplementary Table 1.

Clinical ratings, cerebrospinal fluid (CSF), plasma, and peripheral blood mononuclear cells (PBMCs) were from participants of both genders enrolled in the Neurology Clinic at the Karolinska University Hospital. The study was approved by the regional ethics review board of Stockholm (2016/19-31/2; 2019-04967). Signed informed consents were received from all participants. All PD patients were diagnosed by specialists following the Movement Disorder Society PD Criteria. Disease severity of PD patients was assessed by rating scales, including the Unified Parkinson's Disease Rating Scale-III (UPDRS-III) ($N = 55$), the Hoehn & Yahn (H&Y) scale ($N = 55$), the Montreal Cognitive Assessment (MoCA) ($N = 62$), the Montgomery-Asberg Depression Rating Scale-Self (MADRS-S) ($N = 54$), the Hospital Anxiety and Depression Scale ($N = 51$) including HADS-Anxiety (HADS-A) and HADS-Depression (HADS-D), the Mental Fatigue Scale (MFS) ($N = 51$), and the Beck Depression Inventory-II (BDI-II) ($N = 54$). By setting the MoCA cut-off value at 25, PD patients were classified as PD with mild cognitive impairment (PD-MCI, $N = 30$; MoCA ‹ 25) and PD with normal cognition (PD-NC, $N = 30$; MoCA ≥ 25). Levodopa equivalent daily dose (LEDD) was calculated until the sample date for all patients. Family members of patients, hospital/research staff, or patients without significant neurological disorders, depression, and immunological diseases were recruited as healthy controls. Demographic and clinical characteristics can be found in Supplementary Table 2.

Standardized lumber puncture procedures were carried out to obtain CSF samples. The first 2 ml of CSF was abandoned, and about 10 ml to 12 ml of CSF from the first portion was collected in a sterile polypropylene tube and routinely assayed for cell counting. CSF was centrifuged at 1800 × $g$, 4 °C, for 10 min to remove cells, then aliquoted and stored at −80 °C.

Venipuncture was administered using EDTA tubes to obtain whole blood samples, which were processed within 4 h. Plasma samples were collected during the isolation of PBMCs. PBMC suspensions for flow cytometry analysis were prepared with density gradient centrifugation, using Lymphoprep (Axis-Shield) and following the instructions. PBMC suspensions in freezing medium (10% DMSO, 90% FBS) and plasma samples were stored at −80 °C.

Twenty out of twenty-two CSF samples of PD patients were coupled with plasma samples, which facilitated the correlation analysis of CSF PSAP or PGRN and plasma PSAP or PGRN, respectively. Very few control subjects had both CSF and plasma samples, and thus control subjects were not included in the correlation analysis.

## Immunofluorescence on human FFPE sections

Human FFPE 5-μm sections were deparaffinized and rehydrated, and antigen retrieval (Tris/EDTA buffer: 10 mM Tris-base, 1 mM EDTA solution, 0.5% Tween 20, pH 9.0) was performed for 20 min in a water bath at 95 °C followed by 10 min rinse in cold water. Unspecific bindings were blocked by 1 h incubation with 5% normal donkey serum followed by overnight incubation with mouse anti-TH (1:100; 22941, Nordic Biosite), rabbit anti-PSAP (1:100; 10801-1-AP, Proteintech), and goat anti-PGRN (1:100; AF2420, R&D Systems) in 2.5% normal donkey serum. On the second day, upon washing, sections were incubated for 10 min with 2.5% normal donkey serum followed by 2 h incubation with secondary antibodies in 1% normal donkey serum: donkey anti-mouse IgG Alexa Fluor 488 (1:500; A21202, Invitrogen), donkey anti-goat IgG Alexa Fluor 568 (1:500; A11057, Invitrogen), and donkey anti-rabbit IgG Alexa Fluor 647 (1:500; 711-605-152, Jackson Immuno-Research). After washing, sections were covered with coverslips using a fluorescent mounting medium (S3023, Dako). Fluorescent images were acquired using a Carl Zeiss LSM 880 confocal microscope. Tile scan and z-stack were applied accordingly. Region of interest (ROI) was defined by TH staining. Mean fluorescent intensity (MFI) of PSAP and PGRN were measured in the ROIs. The optical density (O.D.) of neuromelanin was also quantified in the same ROIs. All values were corrected for non-specific backgrounds by subtracting values acquired from unstained areas. Autofluorescence influence from human sections was ruled out by pre-setting signal readouts based on a negative control staining without primary antibodies (Supplementary Fig. 1A).

## Enzyme-linked immunosorbent assay (ELISA) in CSF and plasma

PSAP levels in CSF and plasma were measured by the Human PSAP ELISA kit (LS-F6339, Life-Span Biosciences), of which the assay range is 0.78-50 ng/ml and suitable for both CSF and plasma regarding their content of PSAP. The Human PGRN ELISA kit from Adipogen (AG-45A-0018YEK-KI01) with an assay range of 0.063-4 ng/ml was applied to measure PGRN in CSF. Considering the abundance of PGRN in plasma, the Quantikine Human PGRN Immunoassay Kit (DPGRN0, R&D Systems) with an assay range of 1.6–100 ng/ml was utilized instead. The NF-light ELISA RUO kit (10-7002, Quanterix) was used for NfL measurement in CSF. All ELISA procedures were implemented according to the manufacturer's instructions. Samples were diluted according to the requirements of the corresponding kits. Duplications were made for each sample.

## Flow cytometry

Frozen PBMCs were processed as described before. In brief, frozen aliquots were quickly thawed and washed with PBS. Viability staining was performed with a near-IR dead cell marker (Invitrogen, L10119), followed by washing and incubation in blocking buffer (1% mouse serum and 1% FBS in PBS). Cells were incubated in a mixture of antibodies to identify classically activated monocytes (CL-Mo) ($CD14^+CD16^-$) and non-classically activated monocytes (NC-Mo) ($CD14^-CD16^+$): CD3 FITC clone UCHT1 (1:50; 11-0038-42, Thermo Fisher

Scientific), CD16 PerCP clone 3G8 (1:30; MHCD1631, Thermo Fisher Scientific), and CD14 BV510 clone M5E2 (1:30; 348807, Biolegend); or cytotoxic $CD8^+$ T cells and $CD4^+$ T helper cells: CD4 FITC clone OKT4 (1:50; A27064, Thermo Fisher Scientific), CD3 PerCP/Cy5.5 clone UCHT1 (1:30; 300430, Biolegend), and CD8 BV510 clone SK1 (1:30; 563919, BD biosciences). Upon washing, cells were fixed and permeabilized with Cytofix/Cytoperm solution (BD, #554714) for 20 min, followed by subsequent intracellular PGRN staining using a monoclonal rabbit anti-human PGRN antibody (1:50; 0.6 μg/μl, ab187070, Abcam), PSAP antibody (1:50; 0.7 μg/μl, 10801-1-AP, Proteintech) or an isotype control rabbit IgG antibody (5 μg/μl; Thermo Fisher Scientific) for 1 h at RT. Secondary staining was performed with APC-conjugated goat anti-rabbit antibody (1:100; R&D systems, F0111) for 30 min at RT. Stained cells were analyzed by multicolor flow cytometer Gallios (Beckman Coulter). Mean fluorescence intensity (MFI) was used to represent PSAP and PGRN levels. Dead cell and doublets removal and cell subsets identification were performed using a hierarchical gating strategy for monocyte and T-lymphocyte, respectively (Supplementary Fig. 3A, B). PSAP and PGRN antibodies are specific in detecting PSAP and PGRN in monocytes and T lymphocytes (Supplementary Fig. 3C, D). Data were acquired and analyzed with Kaluza v2.1.1 and FlowJo 10.4.2 (Tree Star) respectively.

## Animals

Conditional PSAP gene-targeted mice (C57BL/6J) were purchased from Cyagen and generated by flanking exon 2-4 of the PSAP gene with loxp sites (Floxed) (Supplementary Fig. 4A). The following primers were used: F1: GCAGAAGATGCAGGACCGTGTG and R1: ATCAC TGGGTCTCCCTAGCCCAAG; F2: ATTCCTGACTCCTGCATCCTG and R2: CACCTTCCTCACAAACCCCTG. Floxed PSAP mice were bred with mice expressing $CreER^{T2}$ enzyme under the dopamine transporter (DAT) gene regulatory sequence in a bacterial artificial chromosome (BAC-DAT-$CreER^{T2}$)[74] or with mice expressing Cre enzyme under the serotonin transporter (SERT) gene regulatory sequence in a bacterial artificial chromosome (BAC-SERT-Cre)[75]. The crosses facilitate inducible PSAP gene deletion specifically in dopamine neurons by generating mice homozygous for the conditional targeted PSAP allele and heterozygous for the BAC-DAT-$CreER^{T2}$ allele ($cPSAP^{DAT}$) and PSAP gene deletion specifically in serotonin neurons by generating mice homozygous for the conditional targeted PSAP allele and heterozygous for the BAC-SERT-Cre allele ($cPSAP^{SERT}$). Littermates homozygous for floxed PSAP carrying no copy of the transgene BAC-DAT-$CreER^{T2}$ or BAC-SERT-Cre served as wild-type controls for $cPSAP^{DAT}$ or $cPSAP^{SERT}$, respectively (Supplementary Fig. 4B). Mice for 6-OHDA experiment were 4m-old male and purchased from Charles River Laboratories. Mice were accommodated in rooms with 12-h light/dark cycles and controlled temperature/humidity (20 °C/53%) and provided with food pellets and water on an ad libitum basis. These experiments were approved by the local ethical committee at Karolinska Institute (N1525-2017, N105/16, N3218-2022) and conducted in accordance with the European Communities Council Directive of 24 November 1986 (86/609/EEC).

Ten-week-old wild-type adult male Sprague Dawley rats, 300–390 g at the time of surgery, were purchased from Charles River and housed four per cage with ad libitum to food and water in 12-hour light/dark cycles. All animal experiments were approved by the local Animal Ethics Committee at Karolinska Institutet (5018-2018).

## Tamoxifen administration

To induce PSAP knock-out in dopamine neurons in $cPSAP^{DAT}$ mice, tamoxifen was administered in $cPSAP^{DAT}$ and their littermate controls at five weeks. Tamoxifen (T5648, Sigma) was dissolved in a mixture of sunflower seed oil (S5007, Sigma) and ethanol (9:1), resulting in a final concentration of 10 mg/ml. 100 mg/kg body weight of tamoxifen was injected (i.p.) daily for five consecutive days.

## Behavior tests on mice

Animals were brought to the testing room for habituation 30 min prior to the starting of tests.

**Open field test (OFT).** The test was administered for 30 min in a 46 × 46 cm arena, in which mice were allowed to move freely. The arena was illuminated with a 30-lux indirect light. A ceiling camera coupled to the EthoVision XT11.5 (Noldus) software was used for video tracking. Distance traveled in the arena and time spent in the center zone were measured. In the locomotor responsiveness experiments, immediately prior to the test, mice were treated (i.p.) with saline or dopaminergic stimulants, including L-dopa/benserazide (10/7.5 mg/kg; D1507/B7283, Sigma-Aldrich), cocaine (30 mg/kg; C5776, Sigma-Aldrich), or SKF81297 (2.5 mg/kg; Cat.1447, Tocris), all in saline. Upon treatment, mice were subjected to the open field test for 30 min.

**Light-dark transition test (LDT).** The test was performed in a box (45 × 30 × 30 cm) comprised of a small (one-third) dark chamber and a large (two-thirds) illuminated (200 lux indirect light) chamber. Mice were allowed to explore freely between two chambers for 15 min while video-tracked by a ceiling camera coupled to the EthoVision XT11.5 (Noldus) software. Distance traveled and time spent in the light chamber were calculated.

**Pole test (PT).** The test was conducted on a vertical pole (diameter: 9 mm; height: 75 cm) above the home cage. Mice were placed facing up on the top of the pole and trained for two days to turn and climb down the pole to their home cage. On testing day, five trials of each animal were recorded. The time to orient downward (T turn) and total time to turn and descend the pole were counted with a maximum duration of 60 s. An average of T-turn time and total time were calculated. Mice fell easily from the pole when they became old (8 and 16 m); thus, results from these two time points were not displayed. Instead, we performed a beam walking test to replace PT at 16 m, which is introduced later in this part.

**Forced swim test (FST).** The forced swim test was based on a revised version of Porsolt's protocol. Mice were placed in a vertical Plexiglas cylinder (diameter: 20 cm) with a 15 cm water depth (23–25 °C) and allowed to swim undisturbed for 6 min and dried before returning to their home cage. The procedure was videotaped, and the last 4 min of the test was analyzed automatically by the EthoVision XT11.5 (Noldus) software.

**Accelerating rotarod test.** The test was performed on a rotarod (47650, Ugo Basile) with falling sensors and automatic timers. Mice were put on a rotating rod that accelerated from 4 rpm to 40 rpm over five minutes. The latency to fall was recorded. Before testing, mice have trained three trials per day, with a half-hour gap between trials, for two consecutive days. All mice received three trials during the testing day, and an average latency to fall was calculated for each mouse.

**Beam traversal test.** The beam was constructed from Plexiglas as described. Mice have trained five trials per day for two consecutive days to traverse the beam. On the test day, a mesh grid (1 cm squares) of corresponding width was mounted over the beam leaving a 1 cm gap between the grid and the beam, mice were videotaped for five trials, and the traversal time and steps needed were counted.

**Apomorphine-induced rotation.** The test was revised from the previous description. Mice were treated with 3 mg/kg apomorphine (i.p; Sigma-Aldrich) and videotaped in a 20 × 20 cm arena by a ceiling camera for 48 min. Total net contralateral rotation (total left-total right 360° turns) in 48 min was quantified.

## Behavior tests on rats

**Open field test.** The test was administered for 10 min in a 60 × 60 cm arena, in which rats were allowed to move freely. The arena was illuminated with a 30-lux indirect light. A ceiling camera coupled to the EthoVision XT11.5 (Noduls) software was used for video tracking. Distance traveled in the arena and time spent in the center zone were measured.

**Apomorphine-induced rotation.** Rats were treated with 1 mg/kg apomorphine (i.p.; Sigma-Aldrich) and videotaped in a 60 × 60 cm arena by a ceiling camera for 30 min. Total net contralateral rotation (total left-total right 360° turns) in 30 min was quantified.

## Experiment design of behavior tests on cPSAP^DAT and cPSAP^SERT mice

Aiming to characterize behaviors and obtain mouse brains of different ages (4 m, 8 m, 16 m), three cohorts of cPSAP^DAT and age and sex-matched WT mice were enrolled in behavior tests. We pooled data from both sexes. The first cohort was assessed with OFT, LDT, PT, rotarod test, and FST at 4 m, then sacrificed with brains taken. The second cohort was assessed with OFT, LDT, rotarod test, and FST at 8 m, then terminated with brains collected. The last cohort was assessed three times; at 4 m, they were assessed with OFT, LDT, PT, and rotarod test; at 8 m, they were assessed with OFT, LDT, and rotarod test; at 16 m, mice were assessed with OFT, LDT, rotarod test, beam traversal test, and FST, then sacrificed with brains dissected.

The cPSAP^SERT and WT mice were assessed by OFT, LDT, and FST at 8 m, then terminated with brains collected.

## Immunofluorescence and lipofuscin imaging

Animals were anesthetized with isofluorane, then perfused with cold phosphate buffer solution (PBS) and 4% paraformaldehyde (PFA). Mouse brains were collected and placed in 4% PFA for 24 h, then dehydrated in 30% sucrose until the tissue sank. Processed brains were embedded in OCT and cryopreserved at −80 °C. Frozen brains were sliced with a cryostat into either 12 μm coronal sections mounted on a Superfrost slide or 30 μm (35 μm for rat brain) free-floating coronal sections preserved in antifreeze solution. Before immunostaining, sections were washed in PBS for 5 min and subjected to antigen retrieval (Tris/EDTA, pH 9.0) for 10 min in a water bath at 80 °C. After antigen retrieval, sections were washed in PBS for 5 min, then incubated in blocking buffer (5% normal donkey serum, 0.25% Triton-X in PBS) for 1 h at room temperature (RT). Subsequently, they were incubated overnight at 4 °C with corresponding primary antibodies: chicken anti-TH (1:500; ab76442, Abcam), chicken anti-GFAP (1:100; ab4674, Abcam), mouse anti-TPH (1:100; T0678, Sigma), mouse anti-NeuN (1:100; MAB377, Chemicon), mouse anti-Darrp32 (1:100; 611520, BD Biosciences), goat anti-IBA1 (1:100; ab5076, Abcam), rabbit anti-IBA1 (1:100; 019-19741, Wako), mouse anti-Olig2 (1:100; MABN50, Merck), rabbit anti-PSAP (1:100; 10801-1-AP, Proteintech), sheep anti-PGRN (1:100; AF2557, R&D systems), mouse anti-human α-synuclein (1:100; sc12767, Santa Cruz), rabbit anti-IDO-1 (1:100; ab106134, Abcam), rabbit anti-TDO-2 (1:100; 15880-1-AP, Proteintech), rabbit anti-FADS-1 (1:100; 10627-1-AP, Proteintech), rabbit anti-SCD-1 (1:100; ab19862, Abcam), rabbit anti-PEX14 (1:100; 10594-1-AP, Proteintech), rabbit anti-ACOX-1 (1:100; 10957-1-AP, Proteintech), rat anti-LAMP-1 (1:100; sc-19992, Santa Cruz Biotechnology), goat anti-CTSD (1:100; AF1029, R&D systems), rabbit anti-GBA (1:100; ab128879, Abcam), rabbit anti-UGCG-1 (1:100, 12869-1-AP, Proteintech), rabbit anti-GALC (1:100, 11991-1-AP, Proteintech), rabbit anti-GLB-1 (1:100, 15518-1-AP, Proteintech), rabbit anti-HEX-A (1:100; 11317-1-AP, Proteintech), rabbit anti-SGMS-1 (1:100; 19050-1-AP, Proteintech), and rabbit anti-SMPD-1 (1:100; 14609-1-AP, Proteintech). The following day, sections were washed in PBS for 3 × 5 min, then incubated for 1 h at RT with corresponding fluorophore-conjugated secondary antibodies: donkey

anti-mouse IgG Alexa Fluor 488 (1:500; A21202, Thermo Fisher Scientific), donkey anti-mouse IgG Alexa Fluor 647 (1:500; A32787, Thermo Fisher Scientific), donkey anti-rabbit IgG Alexa Fluor 568 (1:500; A10042, Thermo Fisher Scientific), donkey anti-sheep IgG Alexa Fluor 488 (1:500; ab150177, Abcam). After $3 \times 5$ min washing in PBS, the sections were counterstained with DAPI for 1 min and covered with coverslips using fluorescent mounting medium (S3023, Dako) (free-floating sections were mounted on a microscope slide before covering). Fluorescent images were acquired with a Carl Zeiss LSM 880 confocal microscope. Tile scan and z-stack were applied appropriately.

For lipofuscin imaging, mouse brain sections were incubated with chicken anti-TH (1:500; ab76442, Abcam) primary antibody and goat anti-chicken IgY (1:500; ab175674, Abcam) secondary antibody, then imaged with a Carl Zeiss LSM 880 confocal microscope. TH staining signal in the 405 channel and autofluorescent signal in the 594-channel were quantified[76]. Notably, a much higher gain was used to excite the autofluorescence than the gain used in antibody staining. MFI was measured in the same way as aforementioned.

## Cell culture, siRNA transfection, and Western blot
N2a mouse neuroblastoma cells (ATCC) were maintained in Dulbecco's Modified Eagle's Medium (DMEM) supplemented with high glucose, 10% Fetal Bovine Serum (FBS), 1% penicillin/streptomycin (final concentration 100 U/ml of penicillin and 100 μg/ml of streptomycin), 10 mM HEPES, 1 mM sodium pyruvate, 2 mM l-GlutaMAX and 1 × Non-Essential Amino Acids (NEAA) (all from Thermo Fisher Scientific, Waltham, MA, USA). Cells were transfected with PSAP-siRNA (AM16708, Thermo Fisher Scientific) or scramble siRNA with Lipofectamine RNAiMax (13778030, Thermo Fisher Scientific) according to the manufacturer's instructions.

Cells lysates were extracted in RIPA buffer (150 mM sodium chloride, 1.0% Triton X-100, 0.5% sodium deoxycholate, 0.1% SDS, 50 mM Tris, pH 8.0) with protease inhibitor cocktail tablet (#A32963, Thermo Fisher Scientific, Waltham, MA, USA) and phosphatase inhibitor (PhosSTOP, #4906837001, Merck, NJ, USA). Concentrations of all samples were measured with the BCA Protein Assay Kit (#23225, Thermo Fisher Scientific, MA, USA) according to the protocol provided by the manufacturer. For Western blotting, samples were mixed with 4×Laemmli protein sample buffer (#1610747, Bio-Rad, CA, USA) and denatured by heating for 10 min at 95 °C. Ten microliters of each sample was loaded in precast 12% Mini-PROTEAN TGX Stain-Free Protein Gels (#4568046, Bio-Rad, CA, USA) and resolved by electrophoresis in a Bio-Rad mini-PROTEAN System. Protein was then transferred to a 0.2 μm nitrocellulose membrane with a Trans-Blot Turbo Transfer System (Bio-Rad). Upon blocking with buffer (#927-60001, LI-COR, NE, USA), membrane was incubated with primary antibodies, including rabbit anti-PSAP (1:100; 10801-1-AP, Proteintech) and mouse anti-β-actin (1:6000, #A5441-100UL, Sigma-Aldrich, MO, USA). Primary antibodies were detected using fluorophore-conjugated goat anti-rabbit (1:10,000, IRDye 680CW) and goat anti-mouse (1:10,000, IRDye 800CW) secondary antibody. The membrane was scanned on an LI-COR Imaging System (Odyssey DLx, LI-COR, NE, USA).

## Fluorescence in situ hybridization (FISH)
RNAscope Fluorescent Multiplex Assay (Advanced Cell Diagnostics) was applied for FISH. Mouse brain was dissected out and snap-frozen in 2-methylbutane on dry ice. Frozen brains were sliced with a cryostat (Leica CM 3050S). Fresh frozen sections were mounted onto Superfrost slides and kept at −80 °C until use. On the day of FISH, fresh frozen sections were first fixed in 4% PFA for 15 min at 4 °C. Next, sections were dehydrated with graded ethanol, followed by incubation with protease IV for 30 min at RT. Then, sections were hybridized with appropriate probes: Th (Mm-Th-C2, cat. 317621-C2), Tph (Mm-Tph2-C2, cat. 318691-C2), and PSAP (Mm-Psap-O1, cat. 529201) for 2 h at 40 °C, followed by four amplification steps (AMP-1FL, 30 min; AMP-

2FL, 15 min; AMP-3FL, 30 min; AMP-4FL, 15 min) at 40 °C. Last, sections were counterstained with DAPI and covered with coverslips using a fluorescent mounting medium (S3023, Dako). Fluorescent images were acquired with a Carl Zeiss LSM 880 confocal microscope, and tile scan and z-stack were applied accordingly.

## Immunohistochemistry
Immunohistochemical staining was mainly performed on free-floating sections. Except staining for TH, GFP, α-synuclein, and p-Ser129 α-synuclein, staining for other proteins requires antigen retrieval (Tris/EDTA, pH 9.0) for 30 min in a water bath at 80 °C. Sections were washed in PBS for $4 \times 5$ min, then quenched for 15 min with a 3% $H_2O_2$ solution. After washing in PBS for $4 \times 5$ min, the sections were pre-incubated with blocking buffer (5% normal goat serum or normal horse serum, 0.25% Triton-X in PBS) for 1 h at RT. Afterwards, they were incubated overnight at 4 °C with corresponding primary antibodies: rabbit anti-TH (1:500; AB152, Merck), rabbit anti-DAT (1:500; AB1591P, Merck), rabbit anti-vesicular monoamine transporter 2 (VMAT2) (1:1000, 20042, ImmunoStar), chicken anti-GFP (1:500; ab13970, Abcam), mouse anti-human α-synuclein (1:500; sc12767, Santa Cruz), and rabbit anti-p-Ser129 α-synuclein (1:250, ab51253, Abcam). On the second day, after washing in PBS for $4 \times 5$ min, sections were incubated for 2 h at RT with corresponding biotinylated secondary antibodies: goat anti-rabbit (1:300, BA1000, Vector Laboratories), goat anti-chicken (1:300; SAB3700204, Merck), and horse anti-mouse (1:300; BA2001, Vector Laboratories). After washing in PBS for $4 \times 5$ min, streptavidin-HRP complex (Vectastain Elite ABC kit; Vector Laboratories) was applied to the sections for 1 h at RT. After washing in PBS for $4 \times 5$ min, sections were exposed to diaminobenzidine (DAB) under monitoring. For p-Ser129 α-synuclein staining, DAB development was enhanced by nickel solution (SK4100, Vector Laboratories). After development, sections were washed in PBS for $4 \times 5$ min and mounted on microscopic slides. Sections were subjected to serial dehydration and covered with coverslips with DPX mounting medium (Sigma-Aldrich). Images were obtained using high-resolution scanners (Epson Perfection V750 PRO, NanoZoomer S360MD). Optical densitometry analysis was performed with Image J in grayscale. p-Ser129 α-synuclein images were transformed into binary format, and then particles were analyzed. Rat brain coronal sections containing the surgical lesion within the coordinate range of +0.2 to +1.0 mm (relative to bregma) were used to bilaterally measure the TH-, DAT-, and VMAT2-positive fiber density of the whole caudate-putamen and the percentage of ipsilateral versus contralateral density was calculated. Mouse brain coronal sections within the coordination range of +0.6 to +1.0 mm (relative to bregma) were used to bilaterally measure the TH- and DAT-positive fiber density of the whole caudate-putamen. Sections within the coordination range of −2.9 to −3.4 mm (relative to bregma) were used to measure TH-positive neuron density of the whole substantia nigra pars compacta. The percentage of ipsilateral versus contralateral density was calculated, respectively. All densitometry values were corrected for non-specific background staining by subtracting values from the corresponding cortex in striatal sections or unstained areas in nigral sections.

## Proteinase K digestion
Proteinase K (PK) digestion was performed according to previous studies[77,78]. Tissue sections were washed $3 \times 5$ min in PBS and wet mounted on a slide. Upon mounting, sections were incubated with 10 μg/ml PK (#19133, Qiagen) in 0.05% SDS/PBS for 5 min at 60 °C in a humidified dark chamber on a hot plate. After PK treatment, sections were washed $3 \times 5$ min in PBS and subjected to p-Ser129 α-syn immunohistochemical staining.

## High-performance liquid chromatography (HPLC)
Sample preparation and HPLC with electrochemical detection (ECD) were done as previously described[79].

Chemicals and reagents, including dopamine hydrochloride (DA), homovanillic acid (HVA), 3-methoxytyramine (3-MT), 3,4-Dihydroxyphenylacetic acid (DOPAC), serotonin hydrochloride (5-HT), 5-hydroxyindole-3-acetic acid (5-HIAA), dihydroxyphenylalanine (DOPA), epinephrine (EPI), l-Noradrenaline hydrochloride (NA), vanillylmandelic acid (VMA), 3-methoxy-4-hydroxyphenylglycol (MHPG), acetonitrile (Chromasolv Plus), monobasic sodium phosphate, ethylene-diamine-tetra-acetic acid (EDTA) disodium salt, 1-octanesulfonic acid (OSA) sodium salt, triethylamine (TEA), 70% perchloric acid (PCA), 85% phosphoric acid and sodium bisulfite were purchased from Sigma Aldrich. HPLC-grade water was generated by a Milli-Q Ultra-Pure water system (Merck Millipore).

Tissue samples were pre-weighted and mixed with ice-cold 0.1 M PCA. Samples were then homogenized using an ultrasonic processor (EpiShear Probe Sonicator; Active Motif) at 20% amplitude for 6 s and incubated on ice for 10 min. After vortexing, homogenized samples were centrifuged at $16,000 \times g$ for 15 min at 4 °C. Eluents were obtained by transferring and centrifuging supernatants in filter tubes with 0.2 μm nylon membrane inserts at $5000 \times g$ for 3 min and kept at −80 °C.

On the day of analysis, standard solutions with concentrations of 200, 100, 50, 10, 5, 2, and 1 ng/ml were prepared with 0.1 M PCA for DA, HVA, 3-MT, DOPAC, 5-HT, 5-HIAA, DOPA, EPI, NA, VMA, and MHPG. Later, calibration curves were calculated using linear regression of peak area versus concentration ($r = 0.999$) with Chromeleon software. Samples were subjected to HPLC-ECD analysis (Dionex Ultimate 3000 series, ThermoFisher Scientific). Separation of analytes was performed on a Dionex C18 reversed-phase MD-150 3.2 mm × 150 mm column (3 μm particle size). Column and analytical cell were kept at 45 °C. The mobile phase (75 mM monobasic sodium phosphate, 3.1 mM OSA, 100 μl/l TEA, 25 μM EDTA, and 10% acetonitrile, pH 3.0 adjusted by 85% phosphoric acid, degassed) was pumped at a flow rate of 0.5 ml/min. For detecting neurotransmitters and metabolites, the first and second analytical cells were set to −100 mV and +300 mV, respectively. Samples were thawed on ice, avoiding light, loaded in the autosampler, and kept at 5 °C for injection. Chromatograms were acquired with Dionex Chromeleon 7 software. Concentrations of analytes were calculated and showed as ng/mg.

## Autoradiographic detection of DAT

Fresh frozen sections mounted on Superfrost slides were used for autoradiographic detection of DAT. Sections were pre-incubated in binding buffer (50 mM Tris−HCl/120 mM NaCl, pH 7.5) for 20 min. They were incubated in binding buffer with 50 pM [$^{125}$I] RTI-55 (Perkin-Elmer Life Sciences) and 1 μM fluoxetine (Tocris Bioscience) for 60 min. For non-specific binding, 100 μM nomifensine (Sigma-Aldrich) was added to the assay. Lastly, sections were washed in an ice-cold binding buffer for 2 × 10 s and rapidly dipped in deionized water. When dried, sections were exposed to Kodak Biomax MR Film (Sigma-Aldrich) in a dark room. After 24 h, autoradiograms were digitized using a high-resolution scanner (Epson Perfection V750 PRO). Optical densitometry was done with Image J in grayscale in the same way as aforementioned.

## Electrophysiology

Adult cPSAP$^{DAT}$ ($N = 5$) and WT mice ($N = 6$) (4−5 m of age) underwent cervical dislocation and decapitation. Their brains were immediately removed, and coronal brain slices (400 μm thick) of striatal level were prepared with a microslicer (VT 1000 S; Leica Microsystem, Heppenheim, Germany). Slices were incubated in oxygenated (95% $O_2$ + 5% $CO_2$) artificial cerebrospinal fluid (aCSF) (126 NaCl, 2.5 KCl, 1.2 $NaH_2PO_4$, 1.3 $MgCl_2$, 2.4 $CaCl_2$, 10 glucose, and 26 $NaHCO_3$, in mmol/L, pH 7.4), for 1 h at 32 °C. Slices were moved to a recording chamber and were continuously supplied with oxygenated aCSF at 28 °C.

Extracellular field potentials were recorded using a glass micropipette filled with aCSF positioned on the surface of the dorsolateral region of the striatum. Synaptic excitatory postsynaptic potentials/population spikes (fEPSP/PSs) were evoked by stimulation pulses delivered every 15 s to the striatum by a concentric bipolar stimulating electrode (FHC, Bowdoinham, ME, United States) placed near the recording electrode. Single stimuli (0.1 ms duration) were applied at an intensity that yields 50−60% maximal fEPSP/PSs determined by a stimulus/response curve established for each slice at the beginning of the recording session. High-frequency stimulation (HFS) (100−Hz train of 1-s duration repeated four times with a 10-s inter-train interval) was used to induce long-term potentiation (LTP) of the fEPSP/PS. Signals were amplified 500 or 1000 times via an Axopatch 200B or a GeneClamp 500B amplifier (Axon Instruments), acquired at 10 kHz, and filtered at 2 kHz. For the D1 agonist study ($N = 5$ mice), slices were pretreated with SKF38393 (5 μM) in the bath before recording. Data were acquired and analyzed with the pClamp 10 software (Axon Instruments, Foster City, CA, United States).

## MALDI-MSI analysis of lipids

**Sample preparation.** Fresh frozen mouse brains were cut at a thickness of 12 μm using a cryostat microtome (Leica CM, Leica Microsystems). Tissue sections were thaw-mounted onto conductive indium tin oxide (ITO) glass slides (Bruker Daltonics) and stored at −80 °C. Sections were desiccated at room temperature for 15 min before spray coating of norharmane matrix solution. Prior to matrix coating, the slide was scanned on a flatbed scanner (Epson perfection V500). The matrix solutions were prepared by dissolving the norharmane matrix powder in 80% MeOH (7.5 mg/ml) solution in a glass vial and sonicated briefly. An automated pneumatic sprayer (HTX-Technologies LLC, Chapel Hill, NC, USA) was used, which was combined with a pump (AKTA FPLC P-905 pump, Cytiva, Uppsala, Sweden) to spray heated matrix solution over the tissue sections. The pump was kept running at 100 μL/min using a 50% acetonitrile pushing solvent before the experiments to ensure a stable flow of the solvent with isocratic pressure. The matrix solution was sprayed using instrumental parameters of a solvent flow rate of 70 μL/min at isocratic pressure, a nitrogen flow of 6 psi, spray temperature of 60 °C, 15 passes with offsets and rotations, a nozzle head velocity of 1200 mm/min, and track spacing of 2.0 mm.

**MALDI-MSI analysis.** All MALDI-MSI experiments for lipid imaging were performed in both negative and positive ionization modes on the same tissue sections using a MALDI-FTICR (Solarix XR 7T-2ω, Bruker Daltonics) mass spectrometer equipped with a Smartbeam II 2 kHz laser. The size of laser was chosen to give a lateral resolution of 100 μm in both polarities with an offset value of 50 μm to ensure no laser ablation overlaps between the polarity switch of the analysis. The instrument was tuned for optimal detection of lipid molecules ($m/z$ 200−2000) in both polarities using the quadrature phase detection (QPD) (2ω) mode. The time-of-flight (TOF) values were set at 0.8 ms for positive and 1.0 ms for negative ion mode analysis and the transfer optics frequency was kept at 4 MHz for both polarity analyses. The quadrupole isolation $m/z$ value (Q1 mass) was set at $m/z$ 220.00 for both polarity analysis. Spectra were collected by summing 100 laser shots per pixel in both polarities. Both methods were calibrated externally with red phosphorus over an appropriate mass range. Ion signals of $m/z$ 885.549853 (monoisotopic peak of [PI(38:4)-H]$^-$) and $m/z$ 798.540963 (monoisotopic peak of [PC(34:1) + K]$^+$) were used for internal calibration for negative and positive polarity analysis, respectively. The laser power was optimized at the start of each analysis and then held constant during the MALDI-MSI experiment. Any possible bias due to factors such as matrix degradation or variation in mass spectrometer response was minimized by randomized analysis of the tissue sections.

The lipid ions were primarily identified by database searches (LIPID MAPS, Nature Lipidomics Gateway, www.lipidmaps.org) based on the high mass accuracy and isotopic distribution provided by the Fourier-transform ion cyclotron resonance (FTICR)-MS analysis. After the MALDI-MSI experiments, MALDI-tandem MS (MS/MS) was performed on tissues by collecting spectra from the brain regions where the target ion is abundant using MALDI-(collision-induced dissociation) CID-FTICR, and the product ions were compared to product ion spectra of standards obtained from LIPID MAPS database (Nature Lipidomics Gateway, www.lipidmaps.org) and/or previously published data from the literature. In the case of MALDI-MS/MS imaging analysis, freshly prepared matrix-coated brain tissue samples were imaged at 100 μm resolution for the target ions, and brain tissue distributions of the product ions were compared to the distribution of the precursor ion. In case of sodium, potassium and matrix adduct of the same lipid species, brain tissue distribution of the adduct ions were compared to the $[M + H]^+$ ions (Supplementary data file 6).

**Image analysis.** MSI data were visualized in FlexImaging (v.5.0, Bruker Daltonics). For further analysis, data were imported to SCiLS Lab (v.2019a Pro, Bruker Daltonics), and brain regions were annotated according to Paxinos and Franklin's stereotaxic atlas. All individual spectra were normalized to the root-mean-square (RMS) of all data points. The average peak areas of the list of annotated-lipid species from each brain region in the mass range $m/z$ 400–2000 were exported in both polarities (Supplementary Figs. 17–31) from SCiLS for statistical analysis.

### MALDI-MSI analysis of neurotransmitters

Fresh-frozen mouse brains were cut at a thickness of 12 μm using a cryostat microtome (Leica CM, Leica Microsystems). Tissue sections were thaw-mounted onto conductive indium tin oxide (ITO) glass slides (Bruker Daltonics), and stored at −80 °C. To minimize enzymatic degradation, sections were thawed and dried in a vacuum desiccator for 20 min. Reactive matrix, FMP-10, was dissolved in 5.5 ml of 70% acetonitrile (4.4 mM) and sprayed over the tissues using an automated pneumatic sprayer (TM-Sprayer, HTX Technologies). The nozzle temperature of the sprayer was set at 90 °C, and the reagent was sprayed pneumatically (6 psi of N2) onto the sample in twenty horizontal passes at a linear nozzle velocity of 110 cm/min with 2 mm track spacing and a flow rate of 80 μl/min. Prior to MSI analysis, the slide was scanned on a flatbed scanner (Epson perfection V500). All MALDI-MSI and experiments were performed in positive ionization mode using a MALDI-FTICR (solariX 7 T 2ω, Bruker Daltonics) mass spectrometer equipped with a Smartbeam II 2 kHz laser. Data were acquired within the range of $m/z$ 150–1000 by firing 100 laser shots per raster position and Q1 mass was set to $m/z$ 379. The lateral resolution was 100 μm. The method was externally calibrated using red phosphorus and internally calibrated using the FMP-10 cluster ion signal ($m/z$ 555.2231) as lock mass. Identification of neurotransmitters and metabolites has been described elsewhere[80,81]. All chemicals and solvents were purchased from Sigma-Aldrich and were used without further purification. Reactive matrix for detection of neurotransmitters (FMP-10) was purchased from Tag-ON AB (Uppsala, Sweden).

### Ultra-high performance liquid chromatography coupled to tandem mass spectrometry (UHPLC-MS/MS)

Glucosyl (β) ceramide (d18:1/18:0), galactosyl (β) ceramide (d18:1/18:0), glucosyl (β) ceramide-d5, glucosyl (β) sphingosine (d18:1), galactosyl (β) sphingosine (d18:1), glucosyl (β) sphingosine-d5, and galactosyl (β) sphingosine-d5 were obtained from Avanti Polar Lipids (Sigma-Aldrich, St. Louis, MO, USA). Acetonitrile (ACN), methanol (MeOH), formic acid, ammonium formate, all LC-MS grade, were purchased from Fisher Scientific (Pittsburgh, PA, USA), dimethyl sulfoxide (DMSO) and all other chemicals were purchased from Sigma-

Aldrich. A Direct-Q-3 UV water purification system (Merck Millipore, Darmstadt, Germany) was used to acquire deionized water (>18 MΩ). Levels of glucosylceramide (GlcCer), galactosylceramide (GalCer), glucosylsphingosine (GlcSph), and galactosylsphingosine (GalSph) in the extracts from the cPSAP^DAT mouse striatum were determined by UHPLC-MS/MS following the protocol as described elsewhere[82].

### NAD⁺/NADH measurement

A Biovision colorimetric kit (K337-100) was used to measure the NAD⁺/NADH ratio in the striatum of cPSAP^DAT mouse brains. To remove NADH-consuming enzymes, mouse brain tissue lysates were filtered through 10 kDa molecular weight cut-off filters (Amicon Ultra, EMD Millipore). The filtrate was subsequently analyzed by using the kit. All procedures of the analysis were conducted according to the manufacturer's instructions.

### Adeno-associated viruses (AAVs)

All AAVs were AAV6 with a promoter of synapsin. The production of AAV-GFP and AAV-α-synuclein was previously described[83]. Stocks of AAV-GFP and AAV-α-synuclein (AAV-α-syn) were diluted in PBS to a final concentration of $7 \times 10^{13}$ gc/ml before injection. AAV-PSAP was customized and purchased from Vector Biolabs. The stock was diluted in PBS to a final concentration of $1 \times 10^{12}$ gc/ml before injection.

### Stereotaxic surgery on mice

For AAV injections, all procedures were conducted under general anesthesia induced by isoflurane (3% for induction, 1% for maintenance) with 0.5 lpm air flow. Mice were placed on a stereotaxic frame (Kopf) with thermal support, and their heads were fixed and aligned horizontally and vertically to the frame. Eye lubricant was used to protect mouse eyes from drying. Lidocaine was applied for local skin anesthesia. Coordinates for the right substantial nigra (SN) used for AAVs injection were anterior-posterior: −3.1 mm, medial-lateral: −1.2 mm, dorsal-ventral: −4.2 mm as calculated relative to bregma and dural surface according to the stereotaxic atlas (Paxinos and Franklin, 2001). Hamilton syringes (5 μl) fitted with a 33 G needle were used for AAV injections. One microliter of AAV solution was injected at a rate of 0.2 μl/min for 5 min. After the injection, the needle was left in the position for 5 min, then slowly retracted. Incisions were stitched and cleaned. Mice recovered with thermal support. Mice were treated with Temgesic (i.p.) three times after surgery. AAV-α-syn/AAV-GFP was injected alone or together with AAV-PSAP in the SN of cPSAP^DAT and WT mice (Supplementary Fig. 10A).

For striatal and medial forebrain bundle (MFB) 6-OHDA lesion, half an hour before surgery, mice were pretreated with desipramine/pargyline (25 mg/kg, 5 mg/kg), i.p. Other surgical procedures are similar as mentioned above. Mice were injected with 1.5 μl 6-OHDA (2 μg/μl in 0.02% ascorbic acid in saline) or 1 μl 6-OHDA (3 μg/μl in 0.02% ascorbic acid in saline) in the striatum (anterior-posterior: +0.5 mm, medial-lateral: −2.1 mm, dorsal-ventral: −3.0 mm) or the MFB (anterior-posterior: −1.1 mm, medial-lateral: −1.1 mm, dorsal-ventral: −4.8 mm), respectively, at the rate of 0.3 μl/min. Postsurgical care was performed as aforementioned.

### Stereotaxic surgery on rats

Rats received surgical procedures under general anesthesia induced by isoflurane (5% for induction, 2% for maintenance) with 1 lpm air flow. Rats were mounted on a stereotaxic frame (Kopf) with thermal support, with their heads fixed and aligned horizontally and vertically to the frame. Eye lubricant was used to protect rat eyes from drying. Lidocaine was applied for local skin anesthesia. Coordinates for the right substantial nigra of rats used for injection were anterior-posterior: −5.3 mm, medial-lateral: −1.7 mm, dorsal-ventral: −7.2 mm relative to bregma and dural surface. Hamilton syringes (5 μl) fitted with a 33 G needle were used for AAV injections. Three μl of AAV solution were

injected at a rate of 0.3 µl/min for 10 min. After the injection, the needle was left in the position for 5 min, then slowly retracted.

Following AAV injection, ECB devices were implanted in the striatum of rat brains. The coordinates for the right striatum of rats were anterior-posterior: +0.5 mm, medial-lateral: −2.8 mm, dorsal-ventral: −7.0 mm relative to bregma and dural surface. The devices were guided into the striatum by a cannula attached to a Microdrive. ECB device was loaded into the cannula and pushed through the cannula by a plunger until the glue-end of the device became just visible. Then the plunger was fixed, and the cannula was slowly retracted out of the rat brain with the ECB device left in the striatum. Subsequently, the plunger was moved out of the brain slowly. Incisions were stitched and cleaned. Rats recovered with thermal support. Rats were treated with Temgesic (i.p.) three times after surgery.

Five groups of rats were included in the experiment and injected with AAV-α-syn in the right substantia nigra. Meanwhile, four different ECB devices (ECB, ECB-PSAP, ECB-PGRN, and ECB-PSAP-PGRN) were implanted in the right striatum of rats from four of the five groups, respectively.

### ARPE-19 cell cultures
Adherent parental ARPE-19 cells and clonal ARPE-19/PSAP, ARPE-19/PGRN, and ARPE-19/PSAP + PGRN co-expressing cell lines were maintained and cultured in F12/DMEM media (Cat. 31331028, Thermo Fisher Scientific) supplemented with 10% FCS and penicillin+streptomycin (PEST, Thermo Fisher Scientific). Each cell line was split once per week.

### Generation of PSAP and PSAP-PGRN overexpressing ARPE-19 cell lines
ARPE-19 cells were split and seeded in 35 mm (diameter) cell culture dishes the day before transfection. ARPE-19 and ARPE-19/PGRN cells were co-transfected with a plasmid encoding the human PSAP-encoding cDNA and an expression construct encoding the sleeping beauty transposase using the Promega Fugene 6 transfection kit, according to previously described protocols[84]. Selection of cells that were stably transfected with the PSAP encoding cDNA was achieved using the addition of hygromycin (200 mg/ml) to the culture medium, and individual colonies were recovered 2–3 weeks post-transfection and expanded for analysis of PSAP, PGRN and PSAP-PGRN complex expression levels. PGRN levels were determined using the human PGRN DuoSet ELISA kit (DY2420, R&D system), and PSAP and PSAP-PGRN complex levels were determined using custom-made ELISAs.

### Human PSAP ELISA
An ELISA for secreted human PSAP was generated based on the commercially available anti-PSAP antibodies H00005660-M01 (mouse monoclonal antibody) and HPA004426 (rabbit polyclonal antibody). The mouse monoclonal anti-PSAP antibody M01 (0.43 mg/ml) was diluted at 1:500 in PBS, and 50 ml of this solution was added to each well of 96-well Maxisorp plates. After overnight incubation at room temp, the antibody solution was discarded, and the plates were washed three times in PBS. Subsequently, each well of the plates was incubated with 150 ml of blocking solution (PBS/Tween-20 (0.1%)/BSA (2%)), and the plates were sealed and left for 1–2 h at room temperature or overnight at +4 °C. After discarding the blocking solution, the plates were washed twice in PBS/Tween-20 (0.1%). Fifty microliters of each sample, including C-terminally His-tagged human PSAP (Abcam, cat. 203534) as standard, diluted in PBS/ Tween-20 (0.1%)/BSA (1%) were then added to the blocked and washed ELISA plate. After incubation at room temp for 2 h or overnight at 4 °C, the analytes were discarded, and unbound and non-specifically bound proteins were removed by washing the plate three times in PBS/Tween-20 (0.1%). Each well was then incubated with 50 ml of the rabbit polyclonal anti-PSAP antibody (HPA00046, 0.1 mg/ml) diluted 1:300 in PBS/Tween-20 (0.1%)/BSA (1%). After 1 h of incubation at room temp, the detection antibody

solution was discarded, and the plate was subjected to three washes in PBS/ Tween-20 (0.1%) solution. Finally, the wells were exposed to 50 ml Ultra-TMB substrate to monitor the presence of HRP activity. The reactions were stopped with 50 ml 2 M $H_2SO_4$, and the plates were immediately read for absorbance of 450 nm (Molecular devices).

### Human PGRN-PSAP complex ELISA assay
Maxisorp plates (96-well) were coated overnight at room temp with 50 ml/well of the anti-PGRN capture antibody provided in the anti-human PGRN Duo-set DY2420 kit (R&D systems). After 3 washes in PBS/Tween-20 (0.1%) solution, the wells were incubated with 150 ml PBS/Tween-20 (0.1%)/BSA (2%) solution for 1-2 h at room temp or overnight at +4 °C to block nonspecific binding of the analyte to the plates. Analytes and standard samples (PGRN/PSAP complexes generated and purified from conditioned media from cell cultures over-expressing PGRN and PSAP) were diluted in PBS/ Tween-20 (0.1%)/BSA (1%) and 50 ml/well were incubated over-night at +4 °C. The reactions were discarded, and the wells were washed three times with PBS/Tween-20 (0.1%) solution. The rabbit polyclonal anti-PSAP antibody (HPA000426) was diluted 1:300 in PBS/Tween-20 (0.1%)/BSA (1%) and used as a detection antibody. After 1 h of incubation at room temp, the secondary antibody solution was removed, and the wells were washed three times in PBS/Tween-20 (0.1%) solution. Bound anti-PSAP antibodies were then targeted using HRP-conjugated anti-rabbit antibodies (Life Technologies), diluted 1:1000 in PBS/Tween-20 (0.1%)/BSA (1%), and incubated for 1 h at room temperature. Finally, the plate was washed, and HRP activity was monitored in the PSAP ELISA.

### Cell encapsulation
The procedure of cell encapsulation has been described previously[85]. Briefly, after trypsinization, the ARPE-19 (control) and different ARPE-19 /factor cell lines were re-suspended in human endothelial (HE)-SFM media (cat. 11111-044, Thermo Fisher Scientific) and then encapsulated in a poly-sulfone hollow fiber membrane containing a polyester terephthalate filament matrix using a custom-manufactured automated cell-loading system. Each end of the membrane was capped with photopolymerized acrylate to retain the cells and scaffolding within the device. The final cylinder-shaped devices, 4 mm long and ~1 mm in diameter, were stored in (HE)-SFM media at 37 °C, 5% $CO_2$ for 2–3 weeks before implantation in rats.

### Analyses of secreted PGRN, PSAP, and PGRN/PSAP complexes in conditioned media from ECB-PGRN and ECB-PGRN-PSAP cells
Frozen batches of cell culture media were slowly thawed overnight at 4 °C. The media was then centrifuged at 7200 rcf for 20 min to pellet dead cells and debris. To secure the complete removal of particles, the supernatant was sterile filtered/degassed using Sarstedt Filtration Units (0.22 um filters) prior to further processing. Subsequently, the filtered conditioned media was concentrated 25x using Amicon centrifugal filters with a 30 kDa cut-off (Millipore Sigma).

Size exclusion chromatography (SEC) was performed on a HiLoad 16/600 Superdex200PG column using an ÄKTA basic 10 FPLC (Cytiva, Sweden). The column was equilibrated and operated in degassed buffer (20 mM sodium phosphate, pH 7.5). The concentrated sterile filtered media was injected from a 3 ml loop, and chromatograms were recorded by monitoring the absorbance at 280 nm; 1 ml fractions were collected and analyzed. Fractions containing PGRN/PSAP complexes were identified using three different ELISA assays, PSAP ELISA, PGRN ELISA (hPGRN ELISA DuoSet) (R&D Systems, USA)) and the PGRN-PSAP complex assay as described before.

### Statistics
All statistical analyses were performed with Prism 9.3.1 (GraphPad, San Diego, USA) or SPSS statistics 25 (IBM). Gaussian distribution (Shapiro-Wilk test) and standard deviation (SD) equality have been tested

before selecting appropriate parametric or nonparametric tests. Outliers were detected by Grubb's test and removed. Student's unpaired *t*-tests with or without Welch's correction and Mann–Whitney *U* tests were used for two-group comparisons; paired *t*-test was used for paired data. One-way analysis of variance (ANOVA) followed by Bonferroni's post hoc test and Kruskal–Wallis's test followed by Dunn's post hoc test were used for multiple-group comparisons. Two-way ANOVA followed by Bonferroni's or Fisher's LSD *post hoc* test has been used in multiple-group comparisons with two independent factors. For matched data, repeated measures (RM) two-way ANOVA followed by Bonferroni's post hoc test was used, assuming sphericity. For principal component analysis (PCA), data were standardized by scaling data to have a mean of zero and SD of one; principal components (PCs) were selected based on parallel analysis. Multiple *t*-tests were done in the volcano plots with a false discovery rate (FDR) set at 5% using the two-stage step-up method of Benjamini, Krieger, and Yekutieli. In association studies, Pearson correlation co-efficiency (r) and *p*-value were calculated, adjusting for age, gender, disease duration, and LEDD score; bivariate normal distribution was tested by Shapiro-Wilk test in residuals before the analysis; regression outliers were detected by residual analysis and removed. For data sets that failed to fulfill bivariate normal distribution, non-parametric Spearman correlation r and *p*-value were calculated. All tests were two-tailed. Data are presented as mean ± S.E.M. Numbers are indicated in each figure legend. A *p*-value less than 0.05 was considered significant. *$p < 0.05$, **$p < 0.01$, ***$p < 0.001$, ****$p < 0.0001$. A statistics checklist is provided (Supplementary data file 5).

### Reporting summary

Further information on research design is available in the Nature Portfolio Reporting Summary linked to this article.

## Data availability

All data associated with this study are presented in the main manuscript, Supplementary data files and Supplementary Information. A source data file containing data underlying all figures and tables are provided with this paper. Lipids and neurotransmitters MALDI-MS raw data in imzML format from dopaminergic PSAP deficient (cPSAP$^{DAT}$), serotonergic PSAP deficient (cPSAP$^{SERT}$), and 6-OHDA-lesioned wildtype (lipids only) mouse brain tissue sections are available at Figshare with the following https://doi.org/10.17044/scilifelab.23856609 (https://doi.org/10.17044/scilifelab.23856609.v1). To respect the data protection and privacy of participants, detailed clinical information can be shared on request from qualified investigators within the limits of participants' consent and according to ethics and material transfer agreements. Source data are provided with this paper.

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

## Acknowledgements

We would like to thank Cyagen for generating the floxed PSAP mice, Shervin Khosousi for assistance in sorting human samples, Tianyi Li, Ioannis Mantas and Yunting Yang for technical support, Marcus Saarinen and Congcong Yuan for helpful discussions. This work was supported by a Wallenberg Clinical Scholarship, the Swedish Brain Foundation, Swedish Research Council, grants 2018-05501, 2018-0550, 2019-0142, 2020-02089, 2021-03293, 2022-04198, the Swedish Parkinson Foundation, Van Geest Foundation, Lexa International/Nordstjernan and the Science for Life Laboratory.

## Author contributions

Study conceptualization and design: P.S., Y.H., X.Z., and P.E.A. Data collection and analysis: Y.H., I.K., R.S., X.Z., J.L., L.U.W., A.N., D.M., J.Z.P., J.K., and H.B. Supervision: P.S., P.E.A., X.Z., and K.C. Y.H. and P.S. wrote the manuscript. All authors contributed to the revision of the manuscript and approved the final draft.

## Funding

## Competing interests

J.L., L.U.W., and H.B. are employees of Sinfonia Biotherapeutics AB, focusing on encapsulated-cell biodelivery therapy. All other authors declare no competing interests.
