## [Peer Review File · Nature Communications]

Reviewers' comments:

Reviewer #1 (Remarks to the Author):

This manuscript by He et al., describes a role for prosaposin (PSAP) in protecting neurons from alpha-synuclein damage in mouse and rat models whilst also showing evidence of subtle alterations of PSAP in Parkinson's disease (PD) patients. The authors also show some subtle effects when PSAP is knocked down only in dopaminergic neurons in mice, which do not occur when knocked down in serotonergic neurons. Most experiments are comprehensive with appropriate controls but the following points should also be addressed before publication:

1. When comparing PSAP in fluids between controls and PD there are no differences between groups but correlations are then presented for PSAP in plasma and CSF of PD patients are expressed as % control. Why have the authors presented the data in this way rather pg/mL? The correlation of levels in control subjects should also be shown – if this is a true relationship disease status should not matter.
2. This data would be strengthened if another dementia cohort with predominantly less dopamine neuron loss had been examined e.g. AD.
3. Similarly, how does PSAP link to NFL levels – is PSAP changes mainly just linked to neuron loss?
4. PSAP RNA levels are mainly microglial and astrocytic (4 fold) according to publicly available data in humans, but mainly neuronal in mice. Although specific cell population staining data of PSAP is presented for mice then this should be presented for humans and ideally for PD/controls. A lot of the data is reliant on PSAP being neuronal but based on this publicly available data it is likely most PSAP is from glia in humans.
5. The authors conduct a lot of lipidomics work to suggest these changes may affect lysosomal function. Have the authors measured lysosomal function directly? This mechanistic information would be more convincing then.
6. The authors linking of PSAP to PGRN could be made clearer in both results and discussion, similarly knowing the levels of PSAP-PGRN complex in PD v control would also be useful.
7. Can the authors describe why DAT would be reduced at 4 months but not at later ages in the cPSAPDAT mice, surely this is critical to explain the link between PSAP and dopaminergic neuron loss?
8. Previous groups using AAV-WT-synuclein show substantial neuronal loss in WT mice but this is not as robust here. The synuclein also induces PSAP loss so can the authors describe/demonstrate how synuclein may drive PSAP loss? This is quite critical because even if increasing PSAP levels is protective if synuclein still remains then this may not be as translational as projected by the authors.
9. Overall, a lot of the findings are subtle but are over-interpreted by the authors, some of this should be toned down. E.g. Behavioral effects in cPSAP-DAT mice, ECB data, biofluids comparing PD to controls.

Reviewer #2 (Remarks to the Author):

The manuscript by He et al. describes an interesting series of experiment examining the potential role of prosaposin (PSAP) in PD pathogenesis. Given the broad interest in lysosomal dysfunction in PD, this line of study should be of considerable interest to the community. After demonstrating that there is some correlation between PSAP and motor disability in PD patients, the author focus on the examination of two mouse models of PSAP deficiency. One is a conditional, postnatal deletion of PSAP in dopaminergic (DA) neurons; the other is a germline deletion in 5-HT neurons. Behavioral, biochemical, electrophysiological and lipidomic approaches are used to characterize these mice; the latter approach is the most interesting of the group given the role of PSAP in lipid metabolism. Lastly, the authors examine the sensitivity of the DA neurons lacking PSAP to viral delivery of an alpha-synuclein expression construct. Although the data make the case that postnatal loss of PSAP in dopaminergic neurons has a wide range of deleterious consequences, including increased sensitivity to the stress created by alpha-synuclein over-expression, the study has major flaws that undermine the central conclusions advanced.

Major concerns:

- The linkage between PSAP mutations and PD risk has been challenged, making statements about the ability of saposin D variant to cause autosomal PD suspect. The authors need to acknowledge this fact. Moreover, why should PSAP levels be altered in idiopathic PD cases in which there aren't PSAP mutations? Were the changes in PSAP MFI correlated with the TH MFI? In the analysis presented in Figure 1, TH appears to have been used as simply a binary marker, rather than a continuous variable.
- The extent of PSAP recombination achieved in the DAT-CreERT2 mouse following tamoxifen treatment needs to be more rigorously documented. There is one image. The mediolateral and rostrocaudal extent of the deletion is important, particularly given the 'global' disruption of lipid metabolism. Do mice with a single copy of the PSAP gene have a phenotype?
- What is the half-life of the PSAP protein? This is important assessing whether the temporal pattern of PSAP levels makes sense or not.
- The effects of PSAP deletion in DA neurons on histochemical properties and transmitter release at relevant time points should be presented before the behavioral data. This is important for many reasons, not the least of which is that the extent of DA depletion seen in the PSAP model is very modest and significant only at the 18 month time point; these changes do not explain the behavioral deficits reported. It is also worth noting that loss of TH staining (or that of DAT) is not equivalent to degeneration. Were stereological methods used for cell counting?
- The synaptic plasticity experiments lack appropriate controls or evidence of rescue with the addition of DA; field potential recordings were used for these experiments, which have limited interpretability in

the striatum because it is not a laminated structure. Furthermore, the sample sizes (N=3) are too small to draw conclusions.

- One of the major strengths of the paper is the use of MALDI-MSI. However, the widespread alterations in lipid metabolism in response to a limited, cell-specific deletion of PSAP is unexplained. If, as suggested by the authors, there is a strong 'horizontal flux' of appropriately processed lipids, why should there be any deficit at all in the mice with selective deletion of a group of neurons that constitute less than 0.1% of the total? Where are there changes in lipid metabolism in regions not innervated by DA neurons?
- As the changes in dopaminergic signaling do not explain the behavioral deficits, the authors turned to the lipidomics. But these data do not explain the evolution of behavioral deficits either as best as I can tell.
- The comparison between the inducible DAT-CreERT2 model and the SERT-Cre model is problematic: one involves gene deletion in puberty, the other during embryonic development. As consequence, the conclusions about the cellular specificity of PSAP function are not compelling. Had the authors used the DAT-Cre line to delete PSAP earlier in development, would they have seen the same thing they saw in the SERT-Cre model?
- What is the nature of the protection afforded by PSAP expression in the PSAP null DA neurons? The only outcome measures are related to phenotypic markers and not to synuclein pathology. Does the impact of PSAP 'rescue' have anything specifically to do with alpha-synuclein or would the same outcome have been seen with any stressor?
- The use of encapsulated cell bio-delivery devices to normalize regional PSAP levels is interesting. However, it is complicated by the fact that the ERG cells used co-release neurotrophic factors and levodopa. Moreover, the evidence that the ECBs alter the impact of aSYN OE is not very convincing. The aggregate distance traveled data is not different among treatment groups at the longest time point if the one outlier for the ECB-PSAP data is set aside. The data on DAT and VMAT2 should not have been analyzed with parametric statistics given its distribution and sample sizes.

Minor concerns:

- While it might be argued there was a trend for the plasma PSAP, there was no trend for CSF.
- 5-HT neurons are lost in PD patient, particularly in the caudal raphe.
- The title of the paper is misleading and should be edited.
- The repeated digression into progranulin biology does little to serve the accessibility or impact of the paper.

Reviewer #3 (Remarks to the Author):

This is a very interesting manuscript entitled “Prosaposin maintains dopaminergic neuronal function and counteracts α -synuclein-induced parkinsonism” by He et al. Recent studies have highlighted the role of these two lysosomal proteins PSAP and PGRN in neurodegeneration broadly and lipid metabolism specifically. Both precursor proteins are cleaved into several individual peptides inside the lysosomes. Here, the authors address the roles of cell-type specific full-length PSAP and PGRN and their relationship in DA neurons and PD.

Using a combination of studies in two different mice models with selective PSAP deletion in either dopamine neurons (cPSAPDAT) or in serotonin neurons (cPSAPSERT), the authors carefully examined these mice for behavioral, electrophysiological, and neurochemical alterations. Critically, only the mice with selective PSAP deletion in dopamine neurons show behavioral symptoms. Additionally, the authors convincingly show that either AAV-PSAP injection or striatal ECB-PSAP implantation show therapeutic effects after insult with AAV delivered α -synuclein transduction, hence offering some sort of protection. Clearly the cell-specific deletion of PSAP and all the phenotypic description are an enormous strength of this manuscript.

Additionally, the discovery of dysregulated lipid metabolism specifically in mice with PSAP deletion in dopamine neurons represents in my opinion an important step in the field. However, specifically these findings need more explanation.

Key points:

1. Given the dramatic changes in the cardiolipin composition, addressing following points would be beneficial: The figure panels 4G,H show relative changes but does not account for absolute amounts. Often low abundant lipids show the most dramatic effect, but overall changes are neglectable. How do these changes add up to the total lipid class? This should help to make a more sophisticated assessment of lipid changes. It should also be discussed how the authors think these changes contribute to disease (fluidity of membranes, etc.).

The authors also mention changes SCD-1 activity in respect to parkinsonism: Have the authors tested these mice (or isolated cells) for SCD-1 activities?

2. The authors mention peroxisomal degradation and use NAD⁺/NADH ratio as a proxy to measure activity. This is a very rough approximation and alternative experiments should be considered to confirm these findings.

3. BMP has recently re-emerged as a key component of glycosphingolipid degradation in FTD and other neurodegenerative diseases. Given the striking losses of GSL in the mouse model it would be important

to examine PG/BMP levels to possibly provide more mechanistic insights into this regulation? BMP also has emerged as a central biomarker to these diseases so it would be interesting to test the available CSF and plasma samples.

This is particularly IMPORTANT since CL is considered as a source of BMP conversion/biosynthesis and BMP is extremely rich in PUFAs.

4. The authors report ganglioside reduction in the brains and show GM levels in the figure panel 4J. GMs are relatively low abundant in comparison to more complex gangliosides (GDs, GTs, GQs, and GPs) in the brain. It would be helpful to see all classes to gain a more complete picture. In addition, the authors speculate that these changes might be due to reduced biosynthesis in the sphingolipid pathway rather than degradation. It would be valuable information to see key enzymes in sphingolipid pathway tested for abundance and activity.

5. Minor point:

While the PCA data nicely separates mice with PSAP-deficient dopamine neurons but not mice with PSAP-deficient serotonin neurons from control mice (at least not in the glycerophospholipid pathway) these graphs don't add much. Hence, I would suggest to combine the key lipidomics/metabolomics findings of both mouse models.

Other minor points:

6. How do the authors explain the discrepancy between their expression pattern of PSAP in the SNc and the literature that PSAP has previously been reported not to be expressed in SNc?

7. The correlation of circulating PSAP with clinical symptoms is intriguing. Have the authors considered exosomes as a source of circulating PSAP and PGRN? Maybe this could be discussed?

8. In some behavioral assays only selected time points are displayed. Is there a reason for this? See for example figure panel 2D, E among others.

9. In EDF 6J PGRN levels don't reach WT levels in but PSAP do. Do the author have an idea/an explanation for this observation?

Considering all these comments, the paper should include more data explaining the potential relationship between PSAP, lipids and susceptibility of DA neurons in parkinsonism which in part can be achieved by addressing the points. Especially a more detailed explanation between peroxisomal turnover, mitochondrial lipid accumulation (cardiolipins) and lysosomal PSAP would help understanding the molecular basis of parkinsonism.

Point-to-point responses to the Reviewers

Point-to-point responses to the comments from Reviewer 1:

Reviewer 1:

This manuscript by He et al., describes a role for prosaposin (PSAP) in protecting neurons from alpha-synuclein damage in mouse and rat models whilst also showing evidence of subtle alterations of PSAP in Parkinson's disease (PD) patients. The authors also show some subtle effects when PSAP is knocked down only in dopaminergic neurons in mice, which do not occur when knocked down in serotonergic neurons. Most experiments are comprehensive with appropriate controls but the following points should also be addressed before publication:

Response: We would like to thank the reviewer for her/his recognition of our extensive investigation of the role of PSAP in relation to dopamine/serotonin neurotransmission and PD.

1. When comparing PSAP in fluids between controls and PD there are no differences between groups but correlations are then presented for PSAP in plasma and CSF of PD patients are expressed as % control. Why have the authors presented the data in this way rather pg/mL? The correlation of levels in control subjects should also be shown – if this is a true relationship disease status should not matter.

Response: According to the manufacturer's instruction of PSAP ELISA kit (<https://www.lsbio.com/elisakits/human-psap-prosaposin-sandwich-elisa-elisa-kit-ls-f6339/6339>), there is an inter-assay CV (<12%) while using the kit. We had 79 plasma samples and could not finish all samples in one ELISA plate. In order to avoid interassay variance, we decided to normalize PD values to the corresponding controls in the same plate. Therefore, the values we present here is percentage instead of absolute value (pg/ml). However, we have now double-checked the results with the absolute values and the main findings, i.e. plasma PSAP/PGRN correlations with UPDRS-3 and BDI-2 (see graphs below).

To make all data consistent, we also transformed CSF data into percentages. However, we have provided average absolute values in **Extended Data Table 2**.

In the correlation of levels between CSF PSAP/PGRN and plasma PSAP/PGRN, we did not include control subjects because only very few of them have both CSF and plasma samples. In other words, the controls we used were largely different for the studies of CSF versus plasma. However, many PD patients had both CSF and plasma taken at the same time and analyzed here. This is stated in the **Methods** section.

2. This data would be strengthened if another dementia cohort with predominantly less dopamine neuron loss had been examined e.g. AD.

Response: We have analyzed CSF samples from normal pressure hydrocephalus (NPH) patients, which is characterized by cognitive impairment but no significant DA neuron loss. As shown in **Extended Data Fig. 2K**, there is no difference in CSF PSAP levels between NPH and control subjects.

3. Similarly, how does PSAP link to NFL levels – is PSAP changes mainly just linked to neuron loss?

Response: Very good point. We have now measured CSF NfL levels in the same cohort of PD and controls as previously used for measurements of PSAP and PGRN. Interestingly, PSAP levels did not correlate with NfL levels in the CSF, while PGRN levels significantly associated with NfL levels (**Extended Data Fig. 2L-M**). It indicates that PSAP changes is not linked to neuroaxonal loss detected by NfL. We find this interesting since NfL is not a strong biomarker for PD. The correlation analysis of NfL with PSAP/PGRN has now been discussed in the **Supplementary Discussion**.

4. PSAP RNA levels are mainly microglial and astrocytic (4 fold) according to publicly available data in humans, but mainly neuronal in mice. Although specific cell population staining data of PSAP is presented for mice then this should be presented for humans and ideally for PD/controls. A lot of the data is reliant on PSAP being neuronal but based on this publicly available data it is likely most PSAP is from glia in humans.

Response: Since we are performing a translational study, we find this point important. At protein levels, immunohistochemical staining from the human protein atlas has shown that PSAP is mainly expressed in neurons of human brain, rather than glial cells (<https://www.proteinatlas.org/ENSG00000197746-PSAP/brain/cerebral+cortex#img>). We have now also performed immunofluorescent staining of PSAP in FFPE cortical sections of healthy human brain (**Extended Data Fig. 6E**) and confirmed a neuronal enrichment of PSAP as shown by the protein atlas (**Extended Data Fig. 6F**). Moreover, a recent study also showed that PSAP is mainly expressed in neurons of human brain (PMID: 31864418). In this context, we would also like to point out that we clearly demonstrate the presence of PSAP protein in DA neurons from brain samples of PD patients and controls. In many cases, there is a discrepancy between mRNA and protein levels of a gene. As shown in the same database (<https://www.proteinatlas.org/ENSG00000197746-PSAP/single+cell+type>), PSAP transcription levels are similar across neuronal and glial cells.

5. The authors conduct a lot of lipidomics work to suggest these changes may affect lysosomal function. Have the authors measured lysosomal function directly? This mechanistic information would be more convincing then.

Response: We now have investigated proteins important for lysosomal function and lipid metabolism. It turned out LAMP1 is not changed, but cathepsin D (CTSD) is

reduced, indicating a general impairment in lysosomes (**Extended Data Fig. 10K**). Moreover, we also measured several lysosomal enzymes related to sphingolipid degradation, including GBA, GALC, GLB-1, HEX-A, and identified increased levels of GALC and GLB-1, further suggesting a disturbed lysosomal function (**Extended Data Fig. 10M-N**). Therefore, there is a lysosomal component in the lipid dyshomeostasis of these mice. Some sentences on this topic have also been added to the **Discussion**.

6. The authors linking of PSAP to PGRN could be made clearer in both results and discussion, similarly knowing the levels of PSAP-PGRN complex in PD v control would also be useful.

Response: We now have analyzed the relationship of PSAP and PGRN levels in both CSF and plasma. As shown by the data (**Extended Data Fig. 2I-J**), PSAP and PGRN correlate in both biofluids, confirming an interdependence of these two proteins at the individual level. Moreover, we also measured the suggested PSAP-PGRN complex levels in both CSF and plasma. Interestingly, there is no change in plasma PSAP-PGRN complex levels between PD and controls (**Extended Data Fig. 2G**). However, when divided into subgroups based on cognitive function, the complex significantly differs between PD-NC and PD-MCI subgroups (**Extended Data Fig. 2H**), indicating a contribution of PSAP-PGRN complex in cognitive impairment of PD. Meanwhile, the complex level in CSF is out of the detection range, indicating a low level of PSAP-PGRN levels in the CSF. We also added more text on the linking of PSAP and PGRN in the **Supplementary Discussion**.

7. Can the authors describe why DAT would be reduced at 4 months but not at later ages in the cPSAPDAT mice, surely this is critical to explain the link between PSAP and dopaminergic neuron loss?

Response: This is a good question. We think that the reuptake of PSAP into neurons somehow contributed to this restoration. However, the reuptake did not solve all dysfunctions caused by PSAP deletion, for example, lipid alterations appear to be more difficult to reverse. This is now added to the **Supplementary Discussion**.

8. Previous groups using AAV-WT-synuclein show substantial neuronal loss in WT mice but this is not as robust here. The synuclein also induces PSAP loss so can the authors describe/demonstrate how synuclein may drive PSAP loss? This is quite critical because even if increasing PSAP levels is protective if synuclein still remains then this may not be as translational as projected by the authors.

Response: In general, dopaminergic loss induced by AAV- α -synuclein overexpression in mice is mild, but stronger in rats, as also found here.¹ The loss of PSAP could be a multifactorial consequence induced by α -synuclein overexpression. It is indeed known that α -synuclein overexpression have different impact on the cellular system, including the synaptic nerve terminals, nucleus, ER (ER stress), lysosomes, and mitochondria (oxidative stress)². In terms of the protective effect of PSAP, our newly added data on p-Ser129 α -synuclein and 6-OHDA toxicity show that PSAP offers protection both by alleviating p-Ser129 α -synuclein aggregation and counteracting oxidative stress. Text on these new data has been added the **Discussion**.

9. Overall, a lot of the findings are subtle but are over-interpreted by the authors, some of this should be toned down. E.g. Behavioral effects in cPSAP-DAT mice, ECB data, biofluids comparing PD to controls.

Response: We are now presenting the data in a more balanced manner. In particular, we adjusted the language concerning the results from the experiments on behaviors

in cPSAP^{DA1} mice, ECB implantation, and biofluid changes of PSAP/PGRN levels. However, the conclusions still stands significant and becomes even firmer with our newly added data.

References in response to Reviewer 1:

1. Ulusoy, A., Decressac, M., Kirik, D. & Bjorklund, A. Viral vector-mediated overexpression of alpha-synuclein as a progressive model of Parkinson's disease. *Progress in brain research* **184**, 89-111 (2010).
2. Wong, Y.C. & Krainc, D. alpha-synuclein toxicity in neurodegeneration: mechanism and therapeutic strategies. *Nature medicine* **23**, 1-13 (2017).

Point-to-point responses to the comments from Reviewer 2:

Reviewer 2:

The manuscript by He et al. describes an interesting series of experiment examining the potential role of prosaposin (PSAP) in PD pathogenesis. Given the broad interest in lysosomal dysfunction in PD, this line of study should be of considerable interest to the community. After demonstrating that there is some correlation between PSAP and motor disability in PD patients, the author focus on the examination of two mouse models of PSAP deficiency. One is a conditional, postnatal deletion of PSAP in dopaminergic (DA) neurons; the other is a germline deletion in 5-HT neurons. Behavioral, biochemical, electrophysiological and lipidomic approaches are used to characterize these mice; the latter approach is the most interesting of the group given the role of PSAP in lipid metabolism. Lastly, the authors examine the sensitivity of the DA neurons lacking PSAP to viral delivery of an alpha-synuclein expression construct. Although the data make the case that postnatal loss of PSAP in dopaminergic neurons has a wide range of deleterious consequences, including increased sensitivity to the stress created by alpha-synuclein over-expression, the study has major flaws that undermine the central conclusions advanced.

Response: We are grateful for the reviewer's careful reading of the manuscript and appreciation of the timeliness of our comprehensive investigation in relation to PD. We have made several new experiments to strengthen the conclusions of this work.

Major concerns:

1. The linkage between PSAP mutations and PD risk has been challenged, making statements about the ability of saposin D variant to cause autosomal PD suspect. The authors need to acknowledge this fact. Moreover, why should PSAP levels be altered in idiopathic PD cases in which there aren't PSAP mutations? Were the changes in PSAP MFI correlated with the TH MFI? In the analysis presented in Figure 1, TH appears to have been used as simply a binary marker, rather than a continuous variable.

Response: We agree with the reviewer and apologize for not adding references that failed to detect saposin D variants in autosomal PD. We have now corrected this in the **Introduction** and **Discussion**. According to published data, it seems that the ethnic heterogeneity and/or age of the studied populations caused discrepancies in results among studies¹. Thus, PSAP mutation-related PD may not be a common genetic form of PD in the whole population. However, since the Japanese study has shown robust

evidence of Saposin D-related PD in their population, the importance of PSAP in the dopaminergic system has at least been demonstrated. Corroborating this, we now citing a recent study presenting several decreased lysosomal-associated proteins, including PSAP, in iPSC-derived DA neurons of young onset sporadic PD patients². Taken together with our data, it is suggested that PSAP could be a critical modifier of the pathogenesis of PD.

Regarding PSAP decrease in idiopathic PD patients, we found that AAV- α -synuclein overexpression induces downregulation of PSAP in DA neurons providing some mechanistic insights. Since α -synuclein has different impacts on the cellular system, including the synaptic nerve terminals, nucleus, ER (ER stress), lysosomes, and mitochondria (oxidative stress), the regulation of PSAP could be multifactorial.

We have now added data on the correlation between PSAP and TH when measured as continuous variables and found that PSAP MFI correlated negatively to TH MFI (**Extended Data Figure 1D**). Moreover, there is a negative correlation between TH and neuromelanin levels (**Extended Data Figure 1B**). Accordingly, PSAP significantly correlates with both TH and neuromelanin levels in human DA neurons.

2. The extent of PSAP recombination achieved in the DAT-CreERT2 mouse following tamoxifen treatment needs to be more rigorously documented. There is one image. The mediolateral and rostrocaudal extent of the deletion is important, particularly given the 'global' disruption of lipid metabolism. Do mice with a single copy of the PSAP gene have a phenotype?

Response: We now have analyzed PSAP transcripts in DA neurons of a series of sections covering the whole substantia nigra of cPSAP^{DAT} mice. Consistent with global disruption of lipid metabolism, PSAP has been successfully deleted in DA neurons of the whole substantia nigra pars compacta (**Extended Data Figure 4C**).

Regarding the influence of PSAP haploinsufficiency on mice, it would be interesting to study. However, this would require the establishment of a new mouse line heterozygous for the conditional targeted PSAP allele and the BAC-DAT-CreER^{T2} allele and extensive studies. This will take several years to accomplish and is not reasonable for this study.

3. What is the half-life of the PSAP protein? This is important assessing whether the temporal pattern of PSAP levels makes sense or not.

Response: This is a good point. We have now added text on this topic in the **Supplementary Discussion**. The turnover of PSAP in neurons has been reported in the rat dentate gyrus³. According to the study, after P14, immature neurons in the SGZ of the hippocampus show decreased PSAP immunoreactivity; after P21, immature neurons do not produce PSAP by themselves, instead, they uptake PSAP secreted by mature neurons and cleave PSAP into saposins. At P28, full-length PSAP is undetectable in immature neurons, which means, in a maximum of 7 days, PSAP has been cleaved into saposins or degraded. In our case, we presented a decrease in PSAP two weeks after the tamoxifen injection, then we observed PSAP recovery at later time points. Therefore, in our study, PSAP uptake is activated earliest two weeks after PSAP genetic deletion and the detected PSAP in later time points could be instantly uptaken PSAP. Thus, the temporal changes in PSAP level found in our study are consistent with the aforementioned study.

4. The effects of PSAP deletion in DA neurons on histochemical properties and transmitter release at relevant time points should be presented before the behavioral

data. This is important for many reasons, not the least of which is that the extent of DA depletion seen in the PSAP model is very modest and significant only at the 18 month time point; these changes do not explain the behavioral deficits reported. It is also worth noting that loss of TH staining (or that of DAT) is not equivalent to degeneration. Were stereological methods used for cell counting?

Response: This is a well-taken point. We have now switched the order of these two parts of data and agree that the mild dopamine deficits cannot fully explain the behavioral data. The new order of presenting the data better connects the behavioral defects with lipid dyshomeostasis.

We agree that reductions in TH/DAT stainings are not equivalent to neuronal loss. Actually, in the manuscript, regarding this, we now used the term “reduced levels of dopaminergic markers” instead of “neurodegeneration”.

The reason why we did not count the cell numbers was the mild loss of TH staining. If there is a significant change in cell numbers, we would probably have detected a more severe loss of TH staining in the striatum⁴. Therefore, we did not count the cell numbers since it is not likely that we would gain more knowledge from it.

5. The synaptic plasticity experiments lack appropriate controls or evidence of rescue with the addition of DA; field potential recordings were used for these experiments, which have limited interpretability in the striatum because it is not a laminated structure. Furthermore, the sample sizes (N=3) are too small to draw conclusions.

Response: LTP measurement in the striatum has been widely used to determine synaptic plasticity although there is a limitation indicated by the reviewer.⁵⁻⁷ We have now added another group with D1 agonist (SKF38393) treatment (**Figure. 2E-G**), which has been shown to restore LTP in PINK1-deficient mice⁸. However, SKF38393 did not restore LTP in the cPSAP^{DAT} mice. Some sentences on these new data have now been added to the **Supplementary Discussion**. Moreover, for all groups, we have now increased the sample sizes to N=5-6 in our study.

6. One of the major strengths of the paper is the use of MALDI-MSI. However, the widespread alterations in lipid metabolism in response to a limited, cell-specific deletion of PSAP is unexplained. If, as suggested by the authors, there is a strong ‘horizontal flux’ of appropriately processed lipids, why should there be any deficit at all in the mice with selective deletion of a group of neurons that constitute less than 0.1% of the total? Where there changes in lipid metabolism in regions not innervated by DA neurons?

Response: From what we observed, we think the horizontal flux is a consequence of lipid dyshomeostasis initiated from the DA system rather than a way to alleviate the dopaminergic dysfunction. Moreover, considering the minor proportion (<0.1%) of DA neurons in the brain, our revealed global lipid dyshomeostasis highlights the importance of PSAP in modulating the lipid metabolism of DA neurons and subsequently many brain regions. Vice versa, lipids from other regions do not counteract the changes seen in cPSAP^{DAT} mice. We have now added this to the **Supplementary Discussion**.

To further examine the horizontal flux, we have now also profiled lipids in the cerebellum that is regionally isolated from the SN and sparsely innervated by DA neurons. We found fewer lipid changes in the cerebellum than striatum (**Extended Data Figure 11**). However, the significantly altered lipids in the cerebellum showed similar changing direction and amplitude as in the striatum. Combining these data, we

think that DA-innervated areas have stronger alterations in lipid composition in cPSAP^{DAT} mice supporting a horizontal flux hypothesis.

7. As the changes in dopaminergic signaling do not explain the behavioral deficits, the authors turned to the lipidomics. But these data do not explain the evolution of behavioral deficits either as best as I can tell.

Response: Very good point. We are excited by the massive changes in several classes of lipids in the cPSAP^{DAT} mice. We present longitudinal changes in both behaviors and lipids in our study. The behavior defects are similar across different time points, and the lipid changes gradually exacerbate. In the revised manuscript, we have now examined levels of proteins/enzymes that regulate lipid metabolism and found changes in the cPSAP^{DAT} mice. As expected, several different proteins/enzymes are altered. Future works are needed to target them individually to better understand the role of each lipid class for the behavioral phenotype in the cPSAP^{DAT} mice.

It should be remembered that changes in markers of the dopamine system does not correlate so well with the symptoms of PD⁹. For instance, in PD patients, some of them will only show motor symptoms until they lose 50% of DA neurons in the SN, indicative of a non-linear relationship between pathological changes and behavioral changes.¹⁰ Similarly, the progressive lipid changes may not necessarily linearly present as progressive behavior defects. Moreover, the existing behavior tests for PD study are not very sensitive and thus sometimes do not reflect minor differences among time points. Some sentences on this topic have now been added to the **Supplementary Discussion**.

8. The comparison between the inducible DAT-CreERT2 model and the SERT-Cre model is problematic: one involves gene deletion in puberty, the other during embryonic development. As consequence, the conclusions about the cellular specificity of PSAP function are not compelling. Had the authors used the DAT-Cre line to delete PSAP earlier in development, would they have seen the same thing they saw in the SERT-Cre model?

Response: To avoid developmental influence and compensations in the dopamine system, we decided to use the DAT-CreERT2 model. We believe that this system is more relevant for studies of the physiology of the mature dopamine system particularly in relation to parkinsonism that initiates in adulthood. Indeed, our cPSAP^{DAT} mice, to some extent, reflects idiopathic PD since these patients show PSAP decrease in adult age². If the PSAP gene was deleted already in the embryonic stage using a DAT-Cre line, we believe that we would have detected an even stronger phenotype both behaviorally and pathologically in these mice. A DAT-Cre line may present more new changes, including neurogenesis of DA neurons, which were not detected in our present study and not particularly relevant to parkinsonism.

We would have preferred to perform the experiments using SERT-Cre under an inducible promotor. However, such mice are not available from Jackson Laboratories, and we cannot find any citation in PubMed using SERT-CreERT2 mice. We still think that the dramatic differences between the two mouse lines are important to report. Since the changes in lipids are so dramatic in the cPSAP^{DAT} mice, it is justified to ask if this happens when PSAP is deleted in any other neuronal population. By including the data with the SERT cre mice, we provide data indicating that this is not the case. We believe that our comparison using these two mouse lines did not compromise the

conclusion we made, instead, it indirectly strengthened our conclusion. The good point raised by the reviewer is now added to the **Supplementary Discussion**.

9. What is the nature of the protection afforded by PSAP expression in the PSAP null DA neurons? The only outcome measures are related to phenotypic markers and not to synuclein pathology. Does the impact of PSAP 'rescue' have anything specifically to do with alpha-synuclein or would the same outcome have been seen with any stressor?

Response: We have now measured the pathological form of α -synuclein, p-Ser129 α -synuclein, in the studied mice (**Figure 5G-H; Extended Data Figure 14F**). It turned out that the aggregation of p-Ser129 α -synuclein is dramatically exacerbated in AAV- α -synuclein overexpressing cPSAP^{DAT} mice, compared to AAV- α -synuclein overexpressing WT mice. Moreover, AAV-PSAP overexpression reversed this situation in cPSAP^{DAT} mice. These facts indicate that p-Ser129 α -synuclein aggregation plays an important role in the vulnerability of cPSAP^{DAT} mice to AAV- α -synuclein overexpression.

To investigate if PSAP also protects mice against oxidative stress, we performed striatal 6-OHDA partial lesion and tested if AAV-PSAP could protect WT mice against this stress. To our excitement, AAV-PSAP protects WT mice against 6-OHDA-induced toxicity both behaviorally and pathologically (**Figure 6**). Therefore, as discussed in the revised manuscript, the rescue effects of PSAP involves alleviation of p-Ser129 α -synuclein aggregation and oxidative stress.

10. The use of encapsulated cell bio-delivery devices to normalize regional PSAP levels is interesting. However, it is complicated by the fact that the ERG cells used co-release neurotrophic factors and levodopa. Moreover, the evidence that the ECBs alter the impact of aSYN OE is not very convincing. The aggregate distance traveled data is not different among treatment groups at the longest time point if the one outlier for the ECB-PSAP data is set aside. The data on DAT and VMAT2 should not have been analyzed with parametric statistics given its distribution and sample sizes.

Response: The secreted contents of RPE cells did confound the study, especially the rescue effect of PSAP, PGRN, or PSAP-PGRN on TH (**Extended Data Figure 15J**). However, comparisons among ECB-PSAP, ECB-PGRN, ECB-PSAP-PGRN, and control ECB still showed a protective effect of PSAP or PSAP-PGRN, especially on DAT levels (**Figure 6G**). Moreover, ECB-PSAP showed a significant protective effect on AAV- α -synuclein overexpressing mice both behaviorally and pathologically, while control ECB failed both behaviorally and on DAT and VMAT2 levels. Therefore, the confounding factors did not hinder our demonstrating the protective effect of ECB-PSAP and PSAP *per se*.

Regarding the presumable outlier in aggregate distance traveled data, this rat was an outlier if we only look at the first timepoint. If we consider also the second and third timepoint, it is not an outlier. Moreover, even if we remove this presumable outlier, the difference is still significant, and the conclusion does not change (see attached graph below). However, in this study, since we focused more on the progression of the rotation behavior and it is a repeated measure (RM) study, this time point is not an outlier in terms of three timepoints of the same rat. Thus, we prefer to keep data from this rat, which does not affect the conclusion.

We double-checked the statistics of DAT and VMAT2. There is no problem to do parametric statistics for DAT, since the data sets are normally distributed (Shapiro-Wilk test) and have equal variances. While indeed, for VMAT2, we neglected that the data sets were not normally distributed, so we now have reanalyzed this set of data. However, the difference between AAV-α-synuclein + ECB-PSAP and AAV-α-synuclein only group is still significant, which means the conclusion stays unchanged. All statistics have been updated in corresponding figure legends and **Extended Data File 5**.

Minor concerns:

11. While it might be argued there was a trend for the plasma PSAP, there was no trend for CSF.

Response: Sorry for neglecting this, we now have changed the text.

12. 5-HT neurons are lost in PD patient, particularly in the caudal raphe.

Response: Sorry for not indicating this properly. We have nuanced the text on 5-HT loss in PD. The serotonin system is rather well preserved at early stages of PD.^{11, 12} While at more advanced stages, there is 30% decrease of SERT in various brain regions.^{13, 14} However, in general, 5-HT neurons are less affected in PD, compared to DA neurons. Moreover, our pathological studies were focused on the dorsal raphe nucleus. Therefore, this factor does not undermine our conclusion. This point has been added to **Supplementary Discussion**.

13. The title of the paper is misleading and should be edited.

Response: We now have changed the title of our paper and focus more on the lipid homeostasis maintenance.

14. The repeated digression into progranulin biology does little to serve the accessibility or impact of the paper.

Response: To some extent, yes. However, since PSAP and PGRN are close partners and show strong interdependence, in the beginning, we wondered if we could benefit from studying PSAP together with PGRN. Unexpectedly, in the context of PD, these two proteins are not so synchronized. However, our study has proven that PGRN is related to non-motor symptoms of PD, which may provide some information for the studying of prodromal PD. Moreover, our investigation of PGRN's protective effect on the PD model also solved that question that may arise after our publication of this study,

i.e., does PGRN has a similar protective effect on PD, and will there be a synergistic effect if PSAP and PGRN are provided together? Therefore, in terms of the paper itself, the PGRN part is not as essential. However, in a larger context, we provided valuable information for the field.

References in response to Reviewer 2:

1. Oji, Y., Hatano, T., Funayama, M. & Hattori, N. Reply: PSAP intronic variants around saposin D domain and Parkinson's disease. *Brain : a journal of neurology* **144**, e4 (2021).
2. Laperle, A.H., *et al.* iPSC modeling of young-onset Parkinson's disease reveals a molecular signature of disease and novel therapeutic candidates. *Nature medicine* **26**, 289-299 (2020).
3. Morishita, M., *et al.* Temporal changes in prosaposin expression in the rat dentate gyrus after birth. *PloS one* **9**, e95883 (2014).
4. Laguna, A., *et al.* Dopaminergic control of autophagic-lysosomal function implicates *Lmx1b* in Parkinson's disease. *Nat Neurosci* **18**, 826-835 (2015).
5. Pascoli, V., Turiault, M. & Lüscher, C. Reversal of cocaine-evoked synaptic potentiation resets drug-induced adaptive behaviour. *Nature* **481**, 71-75 (2011).
6. Calabrese, V., *et al.* A positive allosteric modulator of mGlu4 receptors restores striatal plasticity in an animal model of l-Dopa-induced dyskinesia. *Neuropharmacology* **218**, 109205 (2022).
7. Shen, W., Flajolet, M., Greengard, P. & Surmeier, D.J. Dichotomous dopaminergic control of striatal synaptic plasticity. *Science (New York, N.Y.)* **321**, 848-851 (2008).
8. Kitada, T., *et al.* Impaired dopamine release and synaptic plasticity in the striatum of PINK1-deficient mice. *Proceedings of the National Academy of Sciences of the United States of America* **104**, 11441-11446 (2007).
9. Nandhagopal, R., *et al.* Longitudinal progression of sporadic Parkinson's disease: a multi-tracer positron emission tomography study. *Brain : a journal of neurology* **132**, 2970-2979 (2009).
10. Lang, A.E. & Lozano, A.M. Parkinson's disease. Second of two parts. *The New England journal of medicine* **339**, 1130-1143 (1998).
11. Strecker, K., *et al.* Preserved serotonin transporter binding in de novo Parkinson's disease: negative correlation with the dopamine transporter. *Journal of neurology* **258**, 19-26 (2011).
12. Joutsa, J., Johansson, J., Seppänen, M., Noponen, T. & Kaasinen, V. Dorsal-to-Ventral Shift in Midbrain Dopaminergic Projections and Increased Thalamic/Raphe Serotonergic Function in Early Parkinson Disease. *Journal of nuclear medicine : official publication, Society of Nuclear Medicine* **56**, 1036-1041 (2015).
13. Guttman, M., *et al.* Brain serotonin transporter binding in non-depressed patients with Parkinson's disease. *European journal of neurology* **14**, 523-528 (2007).
14. Kerényi, L., *et al.* Positron emission tomography of striatal serotonin transporters in Parkinson disease. *Archives of neurology* **60**, 1223-1229 (2003).

Point-to-point responses to the comments from Reviewer 3:

Reviewer 3:

This is a very interesting manuscript entitled “Prosaposin maintains dopaminergic neuronal function and counteracts α -synuclein-induced parkinsonism” by He et al. Recent studies have highlighted the role of these two lysosomal proteins PSAP and PGRN in neurodegeneration broadly and lipid metabolism specifically. Both precursor proteins are cleaved into several individual peptides inside the lysosomes. Here, the authors address the roles of cell-type specific full-length PSAP and PGRN and their relationship in DA neurons and PD.

Using a combination of studies in two different mice models with selective PSAP deletion in either dopamine neurons (cPSAP^{DAT}) or in serotonin neurons (cPSAP^{SERT}), the authors carefully examined these mice for behavioral, electrophysiological, and neurochemical alterations. Critically, only the mice with selective PSAP deletion in dopamine neurons show behavioral symptoms. Additionally, the authors convincingly show that either AAV-PSAP injection or striatal ECB-PSAP implantation show therapeutic effects after insult with AAV delivered α -synuclein transduction, hence offering some sort of protection. Clearly the cell-specific deletion of PSAP and all the phenotypic description are an enormous strength of this manuscript.

Additionally, the discovery of dysregulated lipid metabolism specifically in mice with PSAP deletion in dopamine neurons represents in my opinion an important step in the field. However, specifically these findings need more explanation.

Response: We sincerely express our gratitude for the reviewer’s positive comments on our manuscript.

Key points:

1. Given the dramatic changes in the cardiolipin composition, addressing following points would be beneficial: The figure panels 4G,H show relative changes but does not account for absolute amounts. Often low abundant lipids show the most dramatic effect, but overall changes are neglectable. How do these changes add up to the total lipid class? This should help to make a more sophisticated assessment of lipid changes. It should also be discussed how the authors think these changes contribute to disease (fluidity of membranes, etc.).The authors also mention changes SCD-1 activity in respect to parkinsonism: Have the authors tested these mice (or isolated cells) for SCD-1 activities?

Response: The MALDI-MSI technique in our study is semi-quantitative, so we could not offer absolute values. However, as seen in the new **Figure 3**, cardiolipins (72:4/5/7) show strong signals in WT mice and are dramatically altered in cPSAP^{DAT} mice. Moreover, the lipid changes in desaturation and chain-length revealed in our study are two key features of all lipids; thus, most likely, common changes cover most species of lipids, including abundant and low abundant ones. Unfortunately, limited by the detection range of the technique, we now could not calculate changes of total lipid class. This is now mentioned in the **Supplementary Discussion**.

Moreover, regarding the contribution of lipid changes to disease, we now have added more information in the **Supplementary Discussion**.

Since we studied intact brain tissue rather than cells, we measured SCD-1 protein levels instead of activities in DA neurons. We could not find changes in SCD-1 protein levels (**Extended Data Figure 10I-J**). However, we indicate that there could be changes in SCD-1 activities. To study SCD-1 activity, isolated DA neurons will be needed, which is difficult to obtain from mice. A dedicated study using iPSC-induced DA neurons with PSAP gene edited will be interesting to perform in the future. Meanwhile, we also investigated the FADS-1 levels in these mice, and interestingly, FADS-1 is increased, corroborating the increased desaturation of lipids (**Extended Data Figure 10G-H**).

2. The authors mention peroxisomal degradation and use NAD⁺/NADH ratio as a proxy to measure activity. This is a very rough approximation and alternative experiments should be considered to confirm these findings.

Response: The NAD⁺/NADH ratio was measured mainly for the understanding of NAD⁺ recycling in these mice, which, according to literature, is related to glycolysis and mitochondria function. The peroxisome activity may only indirectly take part in the NAD⁺ recycling. To further investigate the peroxisomal function in these mice, we measured PEX14 and ACOX-1, key functional peroxisomal protein and enzyme for peroxisomal β -oxidation, respectively, and found no changes (**Extended Data Figure 10K**). However, like SCD-1, the enzyme activity of ACOX-1, as indicated in the **Discussion**, our data does not preclude that changes in enzyme activity could happen.

3. BMP has recently re-emerged as a key component of glycosphingolipid degradation in FTD and other neurodegenerative diseases. Given the striking losses of GSL in the mouse model it would be important to examine PG/BMP levels to possibly provide more mechanistic insights into this regulation? BMP also has emerged as a central biomarker to these diseases so it would be interesting to test the available CSF and plasma samples. This is particularly IMPORTANT since CL is considered as a source of BMP conversion/biosynthesis and BMP is extremely rich in PUFAs.

Response: BMP is one type of negatively charged lipids and is important for the binding of lipid degradation enzymes and thus affects the metabolism of other lipids. Based on this feature, BMP has been proposed to be responsible for the accumulation of lipids in different studies. However, in our study, we found both increases and decreases of different lipid species, especially the decreased sphingolipids in cPSAP^{DAT} mice. Therefore, BMP may play a complex role in these changes. Moreover, with MALDI-MSI experiments, we cannot separate BMP from PG. This is because BMP and PG are isomers (see attached figure below, https://www.lipidmaps.org/resources/lipidweb/lipidweb_html/lipids/complex/lysobpa/index.htm) and the fragments of BMP and PG are identical as the two lipids are isomers. We have now added this to the **Supplementary Discussion**.

Biosynthesis of bis(monoacylglycero)phosphate

Nonetheless, we detected BMP/PG (44:12) and BMP/PG (36:1) (**Supplementary Information**) since they are the common peaks that are detected as BMP/PG within the brain. These lipid peaks did not show significant changes in the statistics. According to the literature¹, BMP(44:12) is the most abundant BMP species in the mouse brain (50 to 70%), while the most abundant PG species, PG(34:1), account for over 90% of PG species in the mouse brain. Meanwhile, based on the literature, we still cannot identify the BMP/PG species with high accuracy. Likewise, the BMP test in CSF and plasma samples could not be performed by our available MALDI-MSI technique.

4. The authors report ganglioside reduction in the brains and show GM levels in the figure panel 4J. GMs are relatively low abundant in comparison to more complex gangliosides (GDs, GTs, GQs, and GPs) in the brain. It would be helpful to see all classes to gain a more complete picture. In addition, the authors speculate that these changes might be due to reduced biosynthesis in the sphingolipid pathway rather than degradation. It would be valuable information to see key enzymes in sphingolipid pathway tested for abundance and activity.

Response: In addition to the monosialogangliosides (GM1, GM3, GM2), we also detected diasialogangliosides (GD1s and GD3). GMs and some GD1s came up significant in statistical analyses (**Extended Data File 1**). Detection of further complex gangliosides including GTs and GQs require additional methodology to increase signal intensity in the high-mass region in negative ion mode. For this, we searched the literature and applied a protocol using 2,6-DHA matrix that can be helpful for this². While we could detect GT and GQ gangliosides within rodent brain tissue sections (see figure below), the stability of the 2,6-DHA matrix that is used in this experiment was too low (around 3 hours) within our FT-ICR source vacuum. Since we need a run-time of around 15-20 hours, we couldn't run all our samples to perform statistics.

MALDI-FT-ICR MSI of GT and GQ gangliosides within coronal mouse brain tissue sections. Ion images of A) GT1+Na-2H(36:1), B) GT1+Na-2H(38:1), C) GT1+Na-2H(36:1), D) GT1+Na-2H(38:1) are visualized with RMS normalization. Scale bar is 4 mm. Species identified based on their fine isotopic structure.

Additionally, GM1, GD1a, GD1b, and GT1b are the four major brain gangliosides from 5 months of gestation (see figure below)³, of which, GM1 and GD1 have been detected to be changed in our study.

Moreover, we now have measured key sphingolipid related enzymes in DA neurons of cPSAP^{DAT} mice (**Extended Data Figure 10M-N**). Sphingolipid degradation related enzymes, including GBA, GALC, GLB-1, HEX-A, SMPD-1 have been examined. Interestingly, GALC and GLB-1 were increased in these mice. Sphingolipid synthesis related enzymes, including UGCG-1 and SGMS-1, have also been checked, and in accordance with the GSL changes, UGCG-1, an enzyme for GlcCer synthesis, was significantly decreased. However, these enzymes do not fully explain the changes of sphingolipids, for instance the decrease of Phosphosphingolipids. Therefore, we conclude in the **Discussion** that the reduced *de novo* biosynthesis of sphingolipids and disruption of related enzymes contributed together to the changes of sphingolipid changes.

Minor points:

5. While the PCA data nicely separates mice with PSAP-deficient dopamine neurons but not mice with PSAP-deficient serotonin neurons from control mice (at least not in the glycerophospholipid pathway) these graphs don't add much. Hence, I would suggest to combine the key lipidomics/metabolomics findings of both mouse models.

Response: As suggested, we now have integrated these two figures into the new **Figure 3**.

6. How do the authors explain the discrepancy between their expression pattern of PSAP in the SNc and the literature that PSAP has previously been reported not to be expressed in SNc?

Response: This could be the limitation of antibody production in the early years. Our paper is the first one to clearly demonstrate PSAP expression in DA neurons of SNc.

7. The correlation of circulating PSAP with clinical symptoms is intriguing. Have the authors considered exosomes as a source of circulating PSAP and PGRN? Maybe this could be discussed?

Response: We now have added some text on the potential exosome-derived source of circulating PSAP and PGRN in the **Supplementary Discussion**.

8. In some behavioral assays only selected time points are displayed. Is there a reason for this? See for example figure panel 2D, E among others.

Response: This was because the mice could not perform pole test and fell easily from the pole when they become old (8m and 16m). Instead, we did beam walking test to replace this test at 16m. This is now explained in the **Methods** section.

9. In EDF 6J PGRN levels don't reach WT levels in but PSAP do. Do the author have an idea/an explanation for this observation?

Response: This implies that these neurons are still dysfunctional, which affects the trafficking PGRN. Consistent with this, it is known that PGRN get into cells by both PSAP-dependent and independent ways⁴. This is now added to the **Supplementary Discussion**.

References in response to Reviewer 3:

1. Saville, J.T., Lehmann, R.J., Derrick-Roberts, A.L.K. & Fuller, M. Selective normalisation of regional brain bis(monoacylglycero)phosphate in the mucopolysaccharidosis 1 (Hurler) mouse. *Experimental neurology* **277**, 68-75 (2016).
2. Colsch, B., Jackson, S.N., Dutta, S. & Woods, A.S. Molecular microscopy of brain gangliosides: illustrating their distribution in hippocampal cell layers. *ACS chemical neuroscience* **2**, 213-222 (2011).
3. Olsen, A.S.B. & Faergeman, N.J. Sphingolipids: membrane microdomains in brain development, function and neurological diseases. *Open Biol* **7** (2017).
4. Tayebi, N., Lopez, G., Do, J. & Sidransky, E. Pro-cathepsin D, Prosaposin, and Progranulin: Lysosomal Networks in Parkinsonism. *Trends Mol Med* **26**, 913-923 (2020).

REVIEWER COMMENTS

Reviewer #1 (Remarks to the Author):

The authors have in good faith aimed to address all the key points raised by reviewers in this revised manuscript which provides a comprehensive evaluation of the role of prosaposin in dopaminergic neuron function in PD supported by patient data. Upon revision there are just two new points raised by the data/discussion added that will need to be satisfied:

1) PSAP correlates with motor status but does not distinguish PD from controls. The authors need to be clear that this is not diagnostic but could be used for staging progression.

2) In response to point 8 about PSAP and alpha-synuclein's relationship, new data have been added for pser129 alpha-synuclein. The authors now lead as this is a main result in the abstract discussing "pser129 aggregations" and at other points in the manuscript. From the data the only conclusion that can be made is that there are pser129 increases that are decreased with PSAP overexpression in cKO mice. No aggregation data (biochemical insolubility etc, presence of fibrils by EM, PK resistance) is shown, and there is plenty emerging literature suggesting that pser129 may not directly affect aggregation status. Authors either need to do the aggregation experiments to demonstrate true alpha-synuclein/lewy body formation or change language particularly the use of "giant ones" and "pser129 aggregations" which is non-relevant/not demonstrated. I would point out however the fact that pser129 increases are observed in the cKO only I think strengthens the link between PSAP's specific dopaminergic neuron role and a potential link to a-syn and vulnerability in PD.

Reviewer #2 (Remarks to the Author):

Major concerns:

1. The linkage between PSAP mutations and PD risk has been challenged, making statements about the ability of saposin D variant to cause autosomal PD suspect. The authors need to acknowledge this fact. Moreover, why should PSAP levels be altered in idiopathic PD cases in which there aren't PSAP mutations? Were the changes in PSAP MFI correlated with the TH MFI? In the analysis presented in Figure 1, TH appears to have been used as simply a binary marker, rather than a continuous variable.

Response: We agree with the reviewer and apologize for not adding references that failed to detect saposin D variants in autosomal PD. We have now corrected this in the Introduction and Discussion. According to published data, it seems that the ethnic heterogeneity and/or age of the studied populations caused discrepancies in results among studies¹. Thus, PSAP mutation-related PD may not be a common genetic form of PD in the whole population. However, since the Japanese study has shown

robust evidence of Saposin D-related PD in their population, the importance of PSAP in the dopaminergic system has at least been demonstrated. Corroborating this, we now citing a recent study presenting several decreased lysosomal-associated proteins, including PSAP, in iPSC-derived DA neurons of young onset sporadic PD patients². Taken together with our data, it is suggested that PSAP could be a critical modifier of the pathogenesis of PD.

Regarding PSAP decrease in idiopathic PD patients, we found that AAV- α -synuclein overexpression induces downregulation of PSAP in DA neurons providing some mechanistic insights. Since α -synuclein has different impacts on the cellular system, including the synaptic nerve terminals, nucleus, ER (ER stress), lysosomes, and mitochondria (oxidative stress), the regulation of PSAP could be multifactorial.

We have now added data on the correlation between PSAP and TH when measured as continuous variables and found that PSAP MFI correlated negatively to TH MFI (Extended Data Figure 1D). Moreover, there is a negative correlation between TH and neuromelanin levels (Extended Data Figure 1B). Accordingly, PSAP significantly correlates with both TH and neuromelanin levels in human DA neurons.

Response: The evidence does not justify the statement that 'Variants in the saposin D domain of PSAP cause autosomal dominant PD'. At most, these variants are associated with PD. It is also worth noting that in early onset PD patients, particularly those with recessive mutations, there often is little or no discernible aSYN pathology.

2. The extent of PSAP recombination achieved in the DAT-CreERT2 mouse following tamoxifen treatment needs to be more rigorously documented. There is one image. The mediolateral and rostrocaudal extent of the deletion is important, particularly given the 'global' disruption of lipid metabolism. Do mice with a single copy of the PSAP gene have a phenotype?

Response: We now have analyzed PSAP transcripts in DA neurons of a series of sections covering the whole substantia nigra of cPSAPDAT mice. Consistent with global disruption of lipid metabolism, PSAP has been successfully deleted in DA neurons of the whole substantia nigra pars compacta (Extended Data Figure 4C).

Regarding the influence of PSAP haploinsufficiency on mice, it would be interesting to study. However, this would require the establishment of a new mouse line heterozygous for the conditional targeted PSAP allele and the BAC-DAT-CreERT2 allele and extensive studies. This will take several years to accomplish and is not reasonable for this study.

Response: Fig. 2A does not show deletion of PSAP in DA neurons across the SNc/VTA. Neither do the subsequent figures, which focus on high magnification images. Fig. 2L appears to be mislabeled. The data in Extended Data 4C is unconvincing; why have the authors chosen to show hippocampus in these images?

3. What is the half-life of the PSAP protein? This is important assessing whether the temporal pattern of PSAP levels makes sense or not.

Response: This is a good point. We have now added text on this topic in the Supplementary Discussion. The turnover of PSAP in neurons has been reported in the rat dentate gyrus³. According to the study, after P14, immature neurons in the SGZ of the hippocampus show decreased PSAP immunoreactivity;

after P21, immature neurons do not produce PSAP by themselves, instead, they uptake PSAP secreted by mature neurons and cleave PSAP into saposins. At P28, full-length PSAP is undetectable in immature neurons, which means, in a maximum of 7 days, PSAP has been cleaved into saposins or degraded. In our case, we presented a decrease in PSAP two weeks after the tamoxifen injection, then we observed PSAP recovery at later time points. Therefore, in our study, PSAP uptake is activated earliest two weeks after PSAP genetic deletion and the detected PSAP in later time points could be instantly uptaken PSAP. Thus, the temporal changes in PSAP level found in our study are consistent with the aforementioned study.

Response: I find this argument extremely indirect and unconvincing. This should be addressed more directly given the statements by the authors.

4. The effects of PSAP deletion in DA neurons on histochemical properties and transmitter release at relevant time points should be presented before the behavioral data. This is important for many reasons, not the least of which is that the extent of DA depletion seen in the PSAP model is very modest and significant only at the 18 month time point; these changes do not explain the behavioral deficits reported. It is also worth noting that loss of TH staining (or that of DAT) is not equivalent to degeneration. Were stereological methods used for cell counting?

Response: This is a well-taken point. We have now switched the order of these two parts of data and agree that the mild dopamine deficits cannot fully explain the behavioral data. The new order of presenting the data better connects the behavioral defects with lipid dyshomeostasis.

We agree that reductions in TH/DAT stainings are not equivalent to neuronal loss. Actually, in the manuscript, regarding this, we now used the term “reduced levels of dopaminergic markers” instead of “neurodegeneration”.

The reason why we did not count the cell numbers was the mild loss of TH staining. If there is a significant change in cell numbers, we would probably have detected a more severe loss of TH staining in the striatum⁴. Therefore, we did not count the cell numbers since it is not likely that we would gain more knowledge from it.

5. The synaptic plasticity experiments lack appropriate controls or evidence of rescue with the addition of DA; field potential recordings were used for these experiments, which have limited interpretability in the striatum because it is not a laminated structure. Furthermore, the sample sizes (N=3) are too small to draw conclusions.

Response: LTP measurement in the striatum has been widely used to determine synaptic plasticity although there is a limitation indicated by the reviewer.⁵⁻⁷ We have now added another group with D1 agonist (SKF38393) treatment (Figure. 2E-G), which has been shown to restore LTP in PINK1-deficient mice⁸. However, SKF38393 did not restore LTP in the cPSAPDAT mice. Some sentences on these new data have now been added to the Supplementary Discussion. Moreover, for all groups, we have now increased the sample sizes to N=5-6 in our study.

Response: The LTP data are NOT consistent with the statements made in the text. The insensitivity of the putative plasticity to D1 agonist treatment clearly argues whatever is going on, it has nothing to do with dopamine. This whole line of investigation should be deleted. It is also worth mentioning that field

potential measurements of plasticity ARE NOT regarded as a rigorous method of assessing striatal synaptic plasticity.

6. One of the major strengths of the paper is the use of MALDI-MSI. However, the widespread alterations in lipid metabolism in response to a limited, cell-specific deletion of PSAP is unexplained. If, as suggested by the authors, there is a strong 'horizontal flux' of appropriately processed lipids, why should there be any deficit at all in the mice with selective deletion of a group of neurons that constitute less than 0.1% of the total? Where there changes in lipid metabolism in regions not innervated by DA neurons?

Response: From what we observed, we think the horizontal flux is a consequence of lipid dyshomeostasis initiated from the DA system rather than a way to alleviate the dopaminergic dysfunction. Moreover, considering the minor proportion (<0.1%) of DA neurons in the brain, our revealed global lipid dyshomeostasis highlights the importance of PSAP in modulating the lipid metabolism of DA neurons and subsequently many brain regions. Vice versa, lipids from other regions do not counteract the changes seen in cPSAPDAT mice. We have now added this to the Supplementary Discussion.

To further examine the horizontal flux, we have now also profiled lipids in the cerebellum that is regionally isolated from the SN and sparsely innervated by DA neurons. We found fewer lipid changes in the cerebellum than striatum (Extended Data Figure 11). However, the significantly altered lipids in the cerebellum showed similar changing direction and amplitude as in the striatum. Combining these data, we think that DA-innervated areas have stronger alterations in lipid composition in cPSAPDAT mice supporting a horizontal flux hypothesis.

Response: Without further evidence of spreading of lipid dyshomeostasis selectively from dopaminergic neurons, this argument is unconvincing. The cerebellar data makes the situation worse, not better.

7. As the changes in dopaminergic signaling do not explain the behavioral deficits, the authors turned to the lipidomics. But these data do not explain the evolution of behavioral deficits either as best as I can tell.

Response: Very good point. We are excited by the massive changes in several classes of lipids in the cPSAPDAT mice. We present longitudinal changes in both behaviors and lipids in our study. The behavior defects are similar across different time points, and the lipid changes gradually exacerbate. In the revised manuscript, we have now examined levels of proteins/enzymes that regulate lipid metabolism and found changes in the cPSAPDAT mice. As expected, several different proteins/enzymes are altered. Future works are needed to target them individually to better understand the role of each lipid class for the behavioral phenotype in the cPSAPDAT mice.

It should be remembered that changes in markers of the dopamine system does not correlate so well with the symptoms of PD9. For instance, in PD patients, some of them will only show motor symptoms until they lose 50% of DA neurons in the SN, indicative of a non-linear relationship between pathological changes and behavioral changes.¹⁰ Similarly, the progressive lipid changes may not necessarily linearly present as progressive behavior defects. Moreover, the existing behavior tests for PD study are not very sensitive and thus sometimes do not reflect minor differences among time points. Some sentences on this topic have now been added to the Supplementary Discussion.

Response: There is a strong correlation between dopaminergic markers and motor deficits in models of PD and in patients. The lack of correlation is with non-motor symptoms. Certainly in animal models of PD where dopaminergic cell loss is as modest as described here, there are no significant deficits in motor behavior. The authors do not demonstrate that any of the motor deficits they report are responsive to levodopa administration, further compromising their claim that they are due to dysfunction of dopaminergic neurons.

8. The comparison between the inducible DAT-CreERT2 model and the SERT-Cre model is problematic: one involves gene deletion in puberty, the other during embryonic development. As consequence, the conclusions about the cellular specificity of PSAP function are not compelling. Had the authors used the DAT-Cre line to delete PSAP earlier in development, would they have seen the same thing they saw in the SERT-Cre model?

Response: To avoid developmental influence and compensations in the dopamine system, we decided to use the DAT-CreERT2 model. We believe that this system is more relevant for studies of the physiology of the mature dopamine system particularly in relation to parkinsonism that initiates in adulthood. Indeed, our cPSAPDAT mice, to some extent, reflects idiopathic PD since these patients show PSAP decrease in adult age². If the PSAP gene was deleted already in the embryonic stage using a DAT-Cre line, we believe that we would have detected an even stronger phenotype both behaviorally and pathologically in these mice. A DAT-Cre line may present more new changes, including neurogenesis of DA neurons, which were not detected in our present study and not particularly relevant to parkinsonism.

We would have preferred to perform the experiments using SERT-Cre under an inducible promotor. However, such mice are not available from Jackson Laboratories, and we cannot find any citation in PubMed using SERT-CreERT2 mice. We still think that the dramatic differences between the two mouse lines are important to report. Since the changes in lipids are so dramatic in the cPSAPDAT mice, it is justified to ask if this happens when PSAP is deleted in any other neuronal population. By including the data with the SERT cre mice, we provide data indicating that this is not the case. We believe that our comparison using these two mouse lines did not compromise the conclusion we made, instead, it indirectly strengthened our conclusion. The good point raised by the reviewer is now added to the Supplementary Discussion.

Response: I understand that doing the DAT-Cre crosses or coming up with an inducible SERT model may be beyond the scope of this study, but without these experiments the conclusions about selective vulnerability are fundamentally compromised. Germline deletion very frequently leads to compensations that mask the normal adult role of a gene.

9. What is the nature of the protection afforded by PSAP expression in the PSAP null DA neurons? The only outcome measures are related to phenotypic markers and not to synuclein pathology. Does the impact of PSAP 'rescue' have anything specifically to do with alpha-synuclein or would the same outcome have been seen with any stressor?

Response: We have now measured the pathological form of α -synuclein, p-Ser129 α -synuclein, in the studied mice (Figure 5G-H; Extended Data Figure 14F). It turned out that the aggregation of p-Ser129 α -synuclein is dramatically exacerbated in AAV- α -synuclein overexpressing cPSAPDAT mice, compared to AAV- α -synuclein overexpressing WT mice. Moreover, AAV-PSAP overexpression reversed this situation

in cPSAPDAT mice. These facts indicate that p-Ser129 α -synuclein aggregation plays an important role in the vulnerability of cPSAPDAT mice to AAV- α -synuclein overexpression.

To investigate if PSAP also protects mice against oxidative stress, we performed striatal 6-OHDA partial lesion and tested if AAV-PSAP could protect WT mice against this stress. To our excitement, AAV-PSAP protects WT mice against 6-OHDA-induced toxicity both behaviorally and pathologically (Figure 6). Therefore, as discussed in the revised manuscript, the rescue effects of PSAP involves alleviation of p-Ser129 α -synuclein aggregation and oxidative stress.

Response: 6-OHDA lesions are completely useless for drawing inferences about PD pathogenesis. This has been shown over and over again.

10. The use of encapsulated cell bio-delivery devices to normalize regional PSAP levels is interesting. However, it is complicated by the fact that the ERG cells used co-release neurotrophic factors and levodopa. Moreover, the evidence that the ECBs alter the impact of aSYN OE is not very convincing. The aggregate distance traveled data is not different among treatment groups at the longest time point if the one outlier for the ECB-PSAP data is set aside. The data on DAT and VMAT2 should not have been analyzed with parametric statistics given its distribution and sample sizes.

Response: The secreted contents of RPE cells did confound the study, especially the rescue effect of PSAP, PGRN, or PSAP-PGRN on TH (Extended Data Figure 15J). However, comparisons among ECB-PSAP, ECB-PGRN, ECB-PSAP-PGRN, and control ECB still showed a protective effect of PSAP or PSAP-PGRN, especially on DAT levels (Figure 6G). Moreover, ECB-PSAP showed a significant protective effect on AAV- α -synuclein overexpressing mice both behaviorally and pathologically, while control ECB failed both behaviorally and on DAT and VMAT2 levels. Therefore, the confounding factors did not hinder our demonstrating the protective effect of ECB-PSAP and PSAP per se.

Regarding the presumable outlier in aggregate distance traveled data, this rat was an outlier if we only look at the first timepoint. If we consider also the second and third timepoint, it is not an outlier. Moreover, even if we remove this presumable outlier, the difference is still significant, and the conclusion does not change (see attached graph below). However, in this study, since we focused more on the progression of the rotation behavior and it is a repeated measure (RM) study, this time point is not an outlier in terms of three timepoints of the same rat. Thus, we prefer to keep data from this rat, which does not affect the conclusion.

We double-checked the statistics of DAT and VMAT2. There is no problem to do parametric statistics for DAT, since the data sets are normally distributed (Shapiro-Wilk test) and have equal variances. While indeed, for VMAT2, we neglected that the data sets were not normally distributed, so we now have reanalyzed this set of data. However, the difference between AAV- α -synuclein + ECB-PSAP and AAV- α -synuclein only group is still significant, which means the conclusion stays unchanged. All statistics have been updated in corresponding figure legends and Extended Data File 5.

Response: The point is that PSAP expression's impact may depend upon the release of neurotrophic factors and levodopa. There is no control for this confounding factor. Demonstrating that expression of other proteins doesn't rescue anything doesn't address the core problem.

Reviewer #4 (Remarks to the Author):

The authors have addressed the main points raised by the original referees. I have additional points to consider, mostly suggestions for text amendments.

1. As the authors mentioned in the manuscript, PSAP is processed into saposin peptides A-D, each of which regulates a subset of lysosomal hydrolases. For instance, SapC is known to regulate GCase, as nicely reviewed in recent reviews (PMID 32948448 and 36244875). The authors should comment on that and make specific predictions on the type of lysosomal lipid storage that should accumulate from loss of all saposins. They should then specifically address whether (glycosphingo)lipid substrates accumulate based on the known regulatory activities of each saposin. For instance, one may expect an increase in GCase substrates GlcCer and/or GlcSph (or more generically, HexCer and HexSph given the limitations of the approach used) in the Psap KO. The authors should finally comment on these changes (or lack thereof).

2. Can a loss of dopaminergic neurons explain in part the loss of lipids in the cPSAP DAT mice?

3. The authors should comment on whether they detect full length prosaposin and progranulin or, more likely, a combination of full length and cleaved saposin and granulin peptides, resp., with the immunoreagents they use.

4. The relationship between neuromelanin and PSAP levels in human tissue is interesting. Since SapD is a lipofuscin marker and neuromelanin contains some lipofuscin-like material, might the authors be looking at actual lipofuscin within the neuromelanin with the anti-PSAP antibody?

5. The authors should also emphasize how a standard LC/MS-based lipidomic analysis of mouse brain tissues would nicely complement the MALDI imaging approach, and perhaps help resolve some of the lipids that cannot be resolved with MALDI, like BMPs, GlcCer/GalCer etc...

Point-to-point responses to the comments from the Reviewers (round 2):

Reviewer 1:

The authors have in good faith aimed to address all the key points raised by reviewers in this revised manuscript which provides a comprehensive evaluation of the role of prosaposin in dopaminergic neuron function in PD supported by patient data. Upon revision there are just two new points raised by the data/discussion added that will need to be satisfied:

We thank the reviewer for the appreciation of our effort in improving our study and the manuscript.

1) PSAP correlates with motor status but does not distinguish PD from controls. The authors need to be clear that this is not diagnostic but could be used for staging progression.

Response: We agree. We state in the **Results** that CSF and plasma levels of PSAP are not different than controls. PSAP cannot serve as a diagnostic biomarker, but it has some potential as an indicator of disease progression. We have now also added this point to the **Discussion**.

2) In response to point 8 about PSAP and alpha-synuclein's relationship, new data have been added for pser129 alpha-synuclein. The authors now lead as this is a main result in the abstract discussing "pser129 aggregations" and at other points in the manuscript. From the data the only conclusion that can be made is that there are pser129 increases that are decreased with PSAP overexpression in cKO mice. No aggregation data (biochemical insolubility etc, presence of fibrils by EM, PK resistance) is shown, and there is plenty emerging literature suggesting that pser129 may not directly affect aggregation status. Authors either need to do the aggregation experiments to demonstrate true alpha-synuclein/lewy body formation or change language particularly the use of "giant ones" and "pser129 aggregations" which is non-relevant/not demonstrated. I would point out however the fact that pser129 increases are observed in the cKO only I think strengthens the link between PSAP's specific dopaminergic neuron role and a potential link to a-syn and vulnerability in PD.

Response: We have now modified the text and use the terms levels or accumulation instead of aggregation and giant ones to describe p-Ser129 α -syn immunoreactivity.

Apart from this, we have also performed proteinase K (PK) resistance experiment using striatal sections from cPSAP^{DAT} KO mice with AAV- α -syn overexpression. Prior to p-Ser129 α -syn staining, we incubated the striatal sections with PK (10ug/ml in 0.05% SDS in PBS) at 60 degree for 5min¹. As shown by the following image, PK digested most of the small α -syn accumulations but spared some larger accumulations, indicating possible aggregation of p-Ser129 α -syn in these mice. This result has now been added to **Extended Data Fig.15G**.

We totally agree that the data showing p-Ser129 α -syn increases in the cPSAP^{DAT} mice further link PSAP with PD and would like to thank the reviewer for suggesting these experiments.

p-Ser129 α -syn staining on PK-treated cKO+AAV- α -SYN striatal section (Extended Data Fig.15G).

1. Spinelli, K.J., *et al.* Presynaptic alpha-synuclein aggregation in a mouse model of Parkinson's disease. *The Journal of neuroscience : the official journal of the Society for Neuroscience* **34**, 2037-2050 (2014).

Reviewer 2:

Major concerns:

1. The linkage between PSAP mutations and PD risk has been challenged, making statements about the ability of saposin D variant to cause autosomal PD suspect. The authors need to acknowledge this fact. Moreover, why should PSAP levels be altered in idiopathic PD cases in which there aren't PSAP mutations? Were the changes in PSAP MFI correlated with the TH MFI? In the analysis presented in Figure 1, TH appears to have been used as simply a binary marker, rather than a continuous variable.

Response: We agree with the reviewer and apologize for not adding references that failed to detect saposin D variants in autosomal PD. We have now corrected this in the **Introduction** and **Discussion**. According to published data, it seems that the ethnic heterogeneity and/or age of the studied populations caused discrepancies in results among studies¹. Thus, PSAP mutation-related PD may not be a common genetic form of PD in the whole population. However, since the Japanese study has shown robust evidence of Saposin D-related PD in their population, the importance of PSAP in the dopaminergic system has at least been demonstrated. Corroborating this, we now cite a recent study presenting several decreased lysosomal-associated proteins, including PSAP, in iPSC-derived DA neurons of young onset sporadic PD patients². Taken together with our data, it is suggested that PSAP could be a critical modifier of the pathogenesis of PD.

Regarding PSAP decrease in idiopathic PD patients, we found that AAV- α -synuclein overexpression induces downregulation of PSAP in DA neurons providing some mechanistic insights. Since α -synuclein has different impacts on the cellular system, including the synaptic nerve terminals, nucleus, ER (ER stress), lysosomes, and mitochondria (oxidative stress), the regulation of PSAP could be multifactorial.

We have now added data on the correlation between PSAP and TH when measured as continuous variables and found that PSAP MFI correlated negatively to TH MFI (**Extended Data Figure 1D**). Moreover, there is a negative correlation between TH and neuromelanin levels (**Extended Data Figure 1B**). Accordingly, PSAP significantly correlates with both TH and neuromelanin levels in human DA neurons.

Response from reviewer 2: The evidence does not justify the statement that 'Variants in the saposin D domain of PSAP cause autosomal dominant PD'. At most, these variants are associated with PD. It is also worth noting that in early onset PD patients, particularly those with recessive mutations, there often is little or no discernible aSYN pathology.

Response: According to the original study from Japan, these variants were found in three families with autosomal dominant PD. However, considering that PD related to PSAP variants is not seen in other ethnic groups, we agree with the reviewer that it may not be a common genetic contributor of PD. This is now more clearly stated and cited in the **Introduction**. We have now changed the introducing sentence to "Variants in the saposin D domain of the PSAP gene are associated with PD in Japanese patients, but not in other ethnicities".

The reviewer is correct in stating that aSYN pathology is absent or low in early onset autosomal recessive PD, eg caused by biallelic *parkin* mutations. However, there is evidence supporting aSYN pathology in other cases of PD with early age of onset. For example, there are familial forms of PD with aSYN gene mutations and triplications

that start in the thirties or forties^{3, 4}. Furthermore, aSYN also plays a critical role in the common GBA-related PD, a condition characterized by an earlier onset of disease than idiopathic PD⁵.

We also would like to point out that in the iPSC cell study of DA neurons from PD patients that we cited², it is shown that PSAP is decreased while aSYN is increased. We have now added this information to the **Introduction**, **Results**, and **Discussion**.

2. The extent of PSAP recombination achieved in the DAT-CreERT2 mouse following tamoxifen treatment needs to be more rigorously documented. There is one image. The mediolateral and rostrocaudal extent of the deletion is important, particularly given the 'global' disruption of lipid metabolism. Do mice with a single copy of the PSAP gene have a phenotype?

Response: We now have analyzed PSAP transcripts in DA neurons of a series of sections covering the whole substantia nigra of cPSAP^{DAT} mice. Consistent with global disruption of lipid metabolism, PSAP has been successfully deleted in DA neurons of the whole substantia nigra pars compacta (**Extended Data Figure 4C**).

Regarding the influence of PSAP haploinsufficiency on mice, it would be interesting to study. However, this would require the establishment of a new mouse line heterozygous for the conditional targeted PSAP allele and the BAC-DAT-CreER^{T2} allele and extensive studies. This will take several years to accomplish and is not reasonable for this study.

Response from reviewer 2: Fig. 2A does not show deletion of PSAP in DA neurons across the SNc/VTA. Neither do the subsequent figures, which focus on high magnification images. Fig. 2L appears to be mislabeled. The data in Extended Data 4C is unconvincing; why have the authors chosen to show hippocampus in these images?

Response: We have now changed the arrangement of these images and show deletion of PSAP in DA neurons across the SNc/VTA (**Fig. 2A**) and in 5-HT neurons across the dorsal raphe nuclei (**Fig. 2E**). We have now also corrected the labeling in Fig. 2L (now changed to **Fig. 2I**) by removing the superscript of "+". In Extended Data Fig. 4C, we aimed to show that PSAP has been specifically deleted in DA neurons of SNc/VTA, but not in other regions of the brain, including hippocampus. To make this message clearer to the reader, we now also show the zoom-in view of SNc/VTA with separated channels in the figure (**Extended Data Fig. 4C**).

3. What is the half-life of the PSAP protein? This is important assessing whether the temporal pattern of PSAP levels makes sense or not.

Response: This is a good point. We have now added text on this topic in the **Supplementary Discussion**. The turnover of PSAP in neurons has been reported in the rat dentate gyrus⁶. According to the study, after P14, immature neurons in the SGZ of the hippocampus show decreased PSAP immunoreactivity; after P21, immature neurons do not produce PSAP by themselves, instead, they uptake PSAP secreted by mature neurons and cleave PSAP into saposins. At P28, full-length PSAP is undetectable in immature neurons, which means, in a maximum of 7 days, PSAP has been cleaved into saposins or degraded. In our case, we presented a decrease in

PSAP two weeks after the tamoxifen injection, then we observed PSAP recovery at later time points. Therefore, in our study, PSAP uptake is activated earliest two weeks after PSAP genetic deletion and the detected PSAP in later time points could be instantly uptaken PSAP. Thus, the temporal changes in PSAP level found in our study are consistent with the aforementioned study.

Response from reviewer 2: I find this argument extremely indirect and unconvincing. This should be addressed more directly given the statements by the authors.

Response: We have now tried to clarify this point. First of all, we now state at the end of the **Discussion** that studies determining the half-life of PSAP in DA neurons are warranted, as we have not found a specific reference on this topic. However, after a more careful review of the literature, we found a paper with relevance to the posed question⁷. In this study, it was shown, in great detail, how PSAP is processed in different cell types. As shown below, several pathways are described, and it is clearly shown that the proteolytic cleavage of PSAP into individual saposins varies between cell types. This is now stated in the **Discussion**.

We have removed the previous text and citation⁶ speculating on the half-life of PSAP from the **Supplementary Discussion**.

4. The effects of PSAP deletion in DA neurons on histochemical properties and transmitter release at relevant time points should be presented before the behavioral data. This is important for many reasons, not the least of which is that the extent of DA depletion seen in the PSAP model is very modest and significant only at the 18 month time point; these changes do not explain the behavioral deficits reported. It is also worth noting that loss of TH staining (or that of DAT) is not equivalent to degeneration. Were stereological methods used for cell counting?

Response: This is a well-taken point. We have now switched the order of these two parts of data and agree that the mild dopamine deficits cannot fully explain the behavioral data. The new order of presenting the data better connects the behavioral defects with lipid dyshomeostasis.

We agree that reductions in TH/DAT stainings are not equivalent to neuronal loss. Actually, in the manuscript, regarding this, we now used the term “reduced levels of dopaminergic markers” instead of “neurodegeneration”.

The reason why we did not count the cell numbers was the mild loss of TH staining. If there is a significant change in cell numbers, we would probably have detected a more severe loss of TH staining in the striatum⁸. Therefore, we did not count the cell numbers since it is not likely that we would gain more knowledge from it.

5. The synaptic plasticity experiments lack appropriate controls or evidence of rescue with the addition of DA; field potential recordings were used for these experiments, which have limited interpretability in the striatum because it is not a laminated structure. Furthermore, the sample sizes (N=3) are too small to draw conclusions.

Response: LTP measurement in the striatum has been widely used to determine synaptic plasticity although there is a limitation indicated by the reviewer.⁹⁻¹¹ We have now added another group with D1 agonist (SKF38393) treatment (**Figure. 2E-G**), which has been shown to restore LTP in PINK1-deficient mice¹². However, SKF38393 did not restore LTP in the cPSAP^{DAT} mice. Some sentences on these new data have now been added to the **Supplementary Discussion**. Moreover, for all groups, we have now increased the sample sizes to N=5-6 in our study.

Response from reviewer 2: The LTP data are NOT consistent with the statements made in the text. The insensitivity of the putative plasticity to D1 agonist treatment clearly argues whatever is going on, it has nothing to do with dopamine. This whole line of investigation should be deleted. It is also worth mentioning that field potential measurements of plasticity ARE NOT regarded as a rigorous method of assessing striatal synaptic plasticity.

Response: We have now clearly stated in the **Abstract, Results and Discussion** that the dopamine levels and markers are mildly affected in cPSAP^{DAT} mice.

The inability of SKF38393, a postsynaptic D1 agonist, to rescue LTP indicates that either pharmacological activation of postsynaptic D1 receptors is insufficient to recover LTP, or that presynaptic dopamine transmission or other neurotransmitter systems, in particular the glutamatergic/NMDA receptors¹³, may be altered. A speculation is that this is an indirect effect from the strong lipid changes in the cPSAP^{DAT} mice.

We have now changed the statement related to striatal LTP in the **Results** and in the **Supplementary Information**. Since we believe that this part of data helps understand the wide influence of PSAP deletion in DA neurons, instead of deleting these data, we now have moved it to **Extended Data Fig. 5F-H**.

6. One of the major strengths of the paper is the use of MALDI-MSI. However, the widespread alterations in lipid metabolism in response to a limited, cell-specific deletion of PSAP is unexplained. If, as suggested by the authors, there is a strong ‘horizontal flux’ of appropriately processed lipids, why should there be any deficit at all

in the mice with selective deletion of a group of neurons that constitute less than 0.1% of the total? Where are there changes in lipid metabolism in regions not innervated by DA neurons?

Response: From what we observed, we think the horizontal flux is a consequence of lipid dyshomeostasis initiated from the DA system rather than a way to alleviate the dopaminergic dysfunction. Moreover, considering the minor proportion (<0.1%) of DA neurons in the brain, our revealed global lipid dyshomeostasis highlights the importance of PSAP in modulating the lipid metabolism of DA neurons and subsequently many brain regions. Vice versa, lipids from other regions do not counteract the changes seen in cPSAP^{DAT} mice. We have now added this to the **Supplementary Discussion**.

To further examine the horizontal flux, we have now also profiled lipids in the cerebellum that is regionally isolated from the SN and sparsely innervated by DA neurons. We found fewer lipid changes in the cerebellum than striatum (**Extended Data Figure 11**). However, the significantly altered lipids in the cerebellum showed similar changing direction and amplitude as in the striatum. Combining these data, we think that DA-innervated areas have stronger alterations in lipid composition in cPSAP^{DAT} mice supporting a horizontal flux hypothesis.

Response from reviewer 2: Without further evidence of spreading of lipid dyshomeostasis selectively from dopaminergic neurons, this argument is unconvincing. The cerebellar data makes the situation worse, not better.

Response: We have now added more text on this topic to the **Discussion**. Specifically, we now state that the phenomenon of widespread lipid changes needs further investigation but might involve a direct “lipid horizontal flux” between cells or an indirect role of “diffusible metabolic factors” predisposing cells to later lipid composition. Similar mechanisms have been suggested in the pathogenic spread of tau protein aggregation¹⁴.

Concerning lipids in the cerebellum, we have clearly stated in the **Results** that there are fewer changes of lipids in this region than in striatum.

7. As the changes in dopaminergic signaling do not explain the behavioral deficits, the authors turned to the lipidomics. But these data do not explain the evolution of behavioral deficits either as best as I can tell.

Response: Very good point. We are excited by the massive changes in several classes of lipids in the cPSAP^{DAT} mice. We present longitudinal changes in both behaviors and lipids in our study. The behavior defects are similar across different time points, and the lipid changes gradually exacerbate. In the revised manuscript, we have now examined levels of proteins/enzymes that regulate lipid metabolism and found changes in the cPSAP^{DAT} mice. As expected, several different proteins/enzymes are altered. Future works are needed to target them individually to better understand the role of each lipid class for the behavioral phenotype in the cPSAP^{DAT} mice.

It should be remembered that changes in markers of the dopamine system does not correlate so well with the symptoms of PD¹⁵. For instance, in PD patients, some of them will only show motor symptoms until they lose 50% of DA neurons in the SN, indicative of a non-linear relationship between pathological changes and behavioral changes.¹⁶ Similarly, the progressive lipid changes may not necessarily linearly

present as progressive behavior defects. Moreover, the existing behavior tests for PD study are not very sensitive and thus sometimes do not reflect minor differences among time points. Some sentences on this topic have now been added to the **Supplementary Discussion**.

Response from reviewer 2: There is a strong correlation between dopaminergic markers and motor deficits in models of PD and in patients. The lack of correlation is with non-motor symptoms. Certainly in animal models of PD where dopaminergic cell loss is as modest as described here, there are no significant deficits in motor behavior. The authors do not demonstrate that any of the motor deficits they report are responsive to levodopa administration, further compromising their claim that they are due to dysfunction of dopaminergic neurons.

Response: We fully agree with the reviewer that dopamine deficits explain most of the motor symptoms in PD patients and in animal models with strong dopamine deficits and that the hypokinesia can be reversed by L-dopa.

The dopamine levels and markers are mildly affected in cPSAP^{DAT} mice, but the mice still exhibit significant motor and non-motor impairments. We have indeed stated in the manuscript that dopaminergic changes do not fully explain the behavioral deficiencies and the electrophysiological malfunction. This is unusual, but intriguing. There are studies, for example using aSyn transgenic approaches, showing motor deficits in mice with little dopaminergic loss¹⁷. In our study, we suggest that there is a contribution of the strong lipid alterations to the behavioral deficiencies. Since the connections between lipids and behavioral changes are largely unknown, the correlation of lipid alterations and symptoms may turn out to be non-linear.

To better understand the dopaminergic component in the behavioral deficiencies, we now have tested the locomotor responsiveness of cPSAP^{DAT} mice to dopaminergic stimulants, including L-dopa/benserazide, cocaine, and a D1 agonist, in the open field test (following figure). Agreeing with previous studies in WT mice or mice with mild dopaminergic deficiency^{18, 19}, we found hypolocomotion of both WT and cPSAP^{DAT} mice upon L-dopa treatment. In contrast, cocaine induced hyper-locomotion both in WT and cPSAP^{DAT} mice, which was significantly stronger in WT mice. Since cocaine acts at dopaminergic terminals, these experiments, together with the baseline data, provide further evidence that dopamine neurons *per se* are mildly dysfunctional in cPSAP^{DAT} mice. SKF81297, a D1 agonist stimulating striatal dopaminergic neurons, significantly increased locomotion in both genotypes with a similar potency. We have now added this piece of new data in the **Results (Extended Data Fig.5I)**.

8. The comparison between the inducible DAT-CreERT2 model and the SERT-Cre model is problematic: one involves gene deletion in puberty, the other during embryonic development. As consequence, the conclusions about the cellular specificity of PSAP function are not compelling. Had the authors used the DAT-Cre line to delete PSAP earlier in development, would they have seen the same thing they saw in the SERT-Cre model?

Response: To avoid developmental influence and compensations in the dopamine system, we decided to use the DAT-CreERT2 model. We believe that this system is more relevant for studies of the physiology of the mature dopamine system particularly in relation to parkinsonism that initiates in adulthood. Indeed, our cPSAP^{DAT} mice, to some extent, reflects idiopathic PD since these patients show PSAP decrease in adult age². If the PSAP gene was deleted already in the embryonic stage using a DAT-Cre line, we believe that we would have detected an even stronger phenotype both behaviorally and pathologically in these mice. A DAT-Cre line may present more new changes, including neurogenesis of DA neurons, which were not detected in our present study and not particularly relevant to parkinsonism.

We would have preferred to perform the experiments using SERT-Cre under an inducible promotor. However, such mice are not available from Jackson Laboratories, and we cannot find any citation in PubMed using SERT-CreERT2 mice. We still think that the dramatic differences between the two mouse lines are important to report. Since the changes in lipids are so dramatic in the cPSAP^{DAT} mice, it is justified to ask if this happens when PSAP is deleted in any other neuronal population. By including the data with the SERT cre mice, we provide data indicating that this is not the case. We believe that our comparison using these two mouse lines did not compromise the conclusion we made, instead, it indirectly strengthened our conclusion. The good point raised by the reviewer is now added to the **Supplementary Discussion**.

Response from reviewer 2: I understand that doing the DAT-Cre crosses or coming up with an inducible SERT model may be beyond the scope of this study, but without these experiments the conclusions about selective vulnerability are fundamentally compromised. Germline deletion very frequently leads to compensations that mask the normal adult role of a gene.

Response: We have removed the text relating our data to “selective vulnerability” in PD and focus the manuscript on effects of PSAP on DA neurons. The actions of PSAP in 5-HT neurons are indicated to mainly reflect the possibility of “different effects of PSAP loss on different cell populations”. We have also added the reviewer’s text on the limitation of the comparison of the DAT-CreERT2 and SERT-Cre mouse lines both in **Results** and **Discussion**. We have deleted the previous paragraph on this topic from the **Supplementary Information**.

9. What is the nature of the protection afforded by PSAP expression in the PSAP null DA neurons? The only outcome measures are related to phenotypic markers and not to synuclein pathology. Does the impact of PSAP ‘rescue’ have anything specifically to do with alpha-synuclein or would the same outcome have been seen with any stressor?

Response: We have now measured the pathological form of α -synuclein, p-Ser129 α -synuclein, in the studied mice (**Figure 5G-H; Extended Data Figure 14F**). It turned

out that the aggregation of p-Ser129 α -synuclein is dramatically exacerbated in AAV- α -synuclein overexpressing cPSAP^{DAT} mice, compared to AAV- α -synuclein overexpressing WT mice. Moreover, AAV-PSAP overexpression reversed this situation in cPSAP^{DAT} mice. These facts indicate that p-Ser129 α -synuclein aggregation plays an important role in the vulnerability of cPSAP^{DAT} mice to AAV- α -synuclein overexpression.

To investigate if PSAP also protects mice against oxidative stress, we performed striatal 6-OHDA partial lesion and tested if AAV-PSAP could protect WT mice against this stress. To our excitement, AAV-PSAP protects WT mice against 6-OHDA-induced toxicity both behaviorally and pathologically (**Figure 6**). Therefore, as discussed in the revised manuscript, the rescue effects of PSAP involves alleviation of p-Ser129 α -synuclein aggregation and oxidative stress.

Response from reviewer 2: 6-OHDA lesions are completely useless for drawing inferences about PD pathogenesis. This has been shown over and over again.

Response: We agree that 6-OHDA-induced model does not mimic the progressive nature of PD and does not recapitulate several molecular and cellular aspects of PD pathogenesis. We have now added text on these major drawbacks of using 6-OHDA in relation to PD pathogenesis in the **Discussion**.

However, oxidative stress has long been proven to be one of the key aspects underlying DA neurodegeneration. The reviewer requested studies on another “stressor” and we therefore used 6-OHDA as it seems appropriate. We have clearly written that the 6-OHDA experiments are done to show a potential role of PSAP against oxidative stress and not as another model of parkinsonism.

10. The use of encapsulated cell bio-delivery devices to normalize regional PSAP levels is interesting. However, it is complicated by the fact that the ERG cells used co-release neurotrophic factors and levodopa. Moreover, the evidence that the ECBs alter the impact of aSYN OE is not very convincing. The aggregate distance traveled data is not different among treatment groups at the longest time point if the one outlier for the ECB-PSAP data is set aside. The data on DAT and VMAT2 should not have been analyzed with parametric statistics given its distribution and sample sizes.

Response: The secreted contents of RPE cells did confound the study, especially the rescue effect of PSAP, PGRN, or PSAP-PGRN on TH (**Extended Data Figure 15J**). However, comparisons among ECB-PSAP, ECB-PGRN, ECB-PSAP-PGRN, and control ECB still showed a protective effect of PSAP or PSAP-PGRN, especially on DAT levels (**Figure 6G**). Moreover, ECB-PSAP showed a significant protective effect on AAV- α -synuclein overexpressing mice both behaviorally and pathologically, while control ECB failed both behaviorally and on DAT and VMAT2 levels. Therefore, the confounding factors did not hinder our demonstrating the protective effect of ECB-PSAP and PSAP *per se*.

Regarding the presumable outlier in aggregate distance traveled data, this rat was an outlier if we only look at the first timepoint. If we consider also the second and third timepoint, it is not an outlier. Moreover, even if we remove this presumable outlier, the difference is still significant, and the conclusion does not change (see attached graph below). However, in this study, since we focused more on the progression of the rotation behavior and it is a repeated measure (RM) study, this time point is not an

outlier in terms of three timepoints of the same rat. Thus, we prefer to keep data from this rat, which does not affect the conclusion.

We double-checked the statistics of DAT and VMAT2. There is no problem to do parametric statistics for DAT, since the data sets are normally distributed (Shapiro-Wilk test) and have equal variances. While indeed, for VMAT2, we neglected that the data sets were not normally distributed, so we now have reanalyzed this set of data. However, the difference between AAV-α-synuclein + ECB-PSAP and AAV-α-synuclein only group is still significant, which means the conclusion stays unchanged. All statistics have been updated in corresponding figure legends and **Extended Data File 5**.

Response from reviewer 2: The point is that PSAP expression's impact may depend upon the release of neurotrophic factors and levodopa. There is no control for this confounding factor. Demonstrating that expression of other proteins doesn't rescue anything doesn't address the core problem.

Response: We agree that interdependence cannot be ruled out, but it is very challenging to address this type of problem in cell-based delivery systems. All cell lines secrete factors. We have now added text on this limitation in the **Discussion**. However, since our AAV-PSAP experiments have already shown the effect of PSAP, we believe that the ECB study corroborates the therapeutic potential of PSAP and have therefore kept these data in the manuscript.

References:

1. Oji, Y., Hatano, T., Funayama, M. & Hattori, N. Reply: PSAP intronic variants around saposin D domain and Parkinson's disease. *Brain : a journal of neurology* **144**, e4 (2021).
2. Laperle, A.H., *et al.* iPSC modeling of young-onset Parkinson's disease reveals a molecular signature of disease and novel therapeutic candidates. *Nature medicine* **26**, 289–299 (2020).
3. Singleton, A.B., *et al.* alpha-Synuclein locus triplication causes Parkinson's disease. *Science (New York, N.Y.)* **302**, 841 (2003).
4. Polymeropoulos, M.H., *et al.* Mapping of a gene for Parkinson's disease to chromosome 4q21–q23. *Science (New York, N.Y.)* **274**, 1197–1199 (1996).

5. Alcalay, R.N., *et al.* Comparison of Parkinson risk in Ashkenazi Jewish patients with Gaucher disease and GBA heterozygotes. *JAMA neurology* **71**, 752–757 (2014).
6. Morishita, M., *et al.* Temporal changes in prosaposin expression in the rat dentate gyrus after birth. *PloS one* **9**, e95883 (2014).
7. Leonova, T., *et al.* Proteolytic processing patterns of prosaposin in insect and mammalian cells. *The Journal of biological chemistry* **271**, 17312–17320 (1996).
8. Laguna, A., *et al.* Dopaminergic control of autophagic-lysosomal function implicates Lmx1b in Parkinson's disease. *Nat Neurosci* **18**, 826–835 (2015).
9. Pascoli, V., Turiault, M. & Lüscher, C. Reversal of cocaine-evoked synaptic potentiation resets drug-induced adaptive behaviour. *Nature* **481**, 71–75 (2011).
10. Calabrese, V., *et al.* A positive allosteric modulator of mGlu4 receptors restores striatal plasticity in an animal model of l-Dopa-induced dyskinesia. *Neuropharmacology* **218**, 109205 (2022).
11. Shen, W., Flajolet, M., Greengard, P. & Surmeier, D.J. Dichotomous dopaminergic control of striatal synaptic plasticity. *Science (New York, N. Y.)* **321**, 848–851 (2008).
12. Kitada, T., *et al.* Impaired dopamine release and synaptic plasticity in the striatum of PINK1-deficient mice. *Proceedings of the National Academy of Sciences of the United States of America* **104**, 11441–11446 (2007).
13. Nouhi, M., Zhang, X., Yao, N. & Chergui, K. CIQ, a positive allosteric modulator of GluN2C/D-containing N-methyl-d-aspartate receptors, rescues striatal synaptic plasticity deficit in a mouse model of Parkinson's disease. *CNS neuroscience & therapeutics* **24**, 144–153 (2018).
14. Walsh, D.M. & Selkoe, D.J. Amyloid β -protein and beyond: the path forward in Alzheimer's disease. *Current opinion in neurobiology* **61**, 116–124 (2020).
15. Nandhagopal, R., *et al.* Longitudinal progression of sporadic Parkinson's disease: a multi-tracer positron emission tomography study. *Brain : a journal of neurology* **132**, 2970–2979 (2009).
16. Lang, A.E. & Lozano, A.M. Parkinson's disease. Second of two parts. *The New England journal of medicine* **339**, 1130–1143 (1998).
17. Aniszewska, A., Bergstrom, J., Ingelsson, M. & Ekmark-Lewen, S. Modeling Parkinson's disease-related symptoms in alpha-synuclein overexpressing mice. *Brain Behav* **12**, e2628 (2022).
18. Fleming, S.M., *et al.* Behavioral effects of dopaminergic agonists in transgenic mice overexpressing human wildtype alpha-synuclein. *Neuroscience* **142**, 1245–1253 (2006).
19. Oksman, M., Tanila, H. & Yavich, L. Behavioural and neurochemical response of alpha-synuclein A30P transgenic mice to the effects of L-DOPA. *Neuropharmacology* **56**, 647–652 (2009).

Reviewer 3:

No comments in Round 2

Reviewer 4:

The authors have addressed the main points raised by the original referees. I have additional points to consider, mostly suggestions for text amendments.

We would like to thank the reviewer for her/his positive comment towards our previous revision and constructive suggestions for improvements of the manuscript.

1. As the authors mentioned in the manuscript, PSAP is processed into saposin peptides A-D, each of which regulates a subset of lysosomal hydrolases. For instance, SapC is known to regulate GCase, as nicely reviewed in recent reviews (PMID 32948448 and 36244875). The authors should comment on that and make specific predictions on the type of lysosomal lipid storage that should accumulate from loss of all saposins. They should then specifically address whether (glycosphingo)lipid substrates accumulate based on the known regulatory activities of each saposin. For instance, one may expect an increase in GCase substrates GlcCer and/or GlcSph (or more generically, HexCer and HexSph given the limitations of the approach used) in the Psap KO. The authors should finally comment on these changes (or lack thereof).

Response: Regarding the role of saposin C in regulating GCase activity, we have cited PMID32948448 (reference 14) in previous version of our manuscript. We have now also added the newly published review PMID36244875 (reference 15), which summarized the relationship of PGRN and PSAP and their role in lipid regulation, especially GCase activity. Furthermore, regarding a comprehensive view of saposins on lipid metabolism, we now also cite “Lysosomal Glycosphingolipid Storage Diseases” by Konrad Sandhoff and Bernadette Breiden¹, where they summarized the role of saposins on glycosphingolipid metabolism (see the following figure).

According to this literature¹, saposin A modulates β -galactosylceramidase (GALC); saposin B modulates GM1- β -galactosidase, neuraminidase (Neu), α -galactosidase A (GLA), β -galactosidase (GLB), and arylsulfatase A (ASA); saposin C modulates GLB

and glucocerebrosidase (GBA); saposin D modulates ceramidase; saposin D has also been shown to modulate sphingomyelinase (SMase)². Based on this, PSAP loss should have triggered accumulation in gangliosides (especially GM1 and GM3), globosides (especially Gb3), LacCer, GlcCer, GalCer, sulfatides, and sphingomyelins (SMs). There may also be a decrease of ceramide due to reduced activity of glycolipid pathways. Because of saposin D deficiency, a decrease of sphingosine (Sph) may also be found. We now have added this summary of saposins' role in sphingolipid metabolism and the prediction of lipid changes due to saposin loss to the **Results**.

However, as shown by our data (**Extended Data Fig. 9A-D and Fig. 3H-I**) and discussed in the manuscript, we did not detect accumulation of GlcCer/GlcSph or GalCer/GalSph by UHPLC-MS/MS. However, we found universal decrease of gangliosides (GM1, GM2, and GM3), ceramide-1-phosphate, and sphingomyelins (SMs) by MALDI-MSI in cPSAP^{DAT} mouse brains. We did not identify globosides, LacCer, or sphingosine with our MALDI method.

For sulfatides, we now found that we have missed pointing out one piece of information in the lipid list of our previous version of the manuscript. That is, we noticed increased polyunsaturated sulfatides (SHexCer(t43:2), SHexCer(t42:2), SHexCer(t44:2)) and decreased mono-unsaturated sulfatides (SHexCer(d40:1), SHexCer(d36:1)) especially in the cortex in cPSAP^{DAT} mouse brains. These data are consistent with the desaturation pattern identified in CLs and PCs. We have now added this information to the **Results**.

We conclude in the **Discussion** that these sphingolipid changes may be a consequence of decreased *de novo* synthesis, increased degradation, and lack-of-saposins-induced accumulation of sphingolipids.

2. Can a loss of dopaminergic neurons explain in part the loss of lipids in the cPSAP DAT mice?

Response: Good point. As we have shown, there was no TH loss in the striatum of 4-month-old cPSAP^{DAT} mice (**Fig. 2C,D**). However, we have detected dramatic lipid alterations in 4-month-old cPSAP^{DAT} mouse brains (**Extended Data Fig. 9E, H**), which suggests that loss of dopaminergic neurons may not explain loss of lipids. To directly explore if acute DA neuronal loss can trigger similar lipid alterations in WT mouse brains, we have now performed MALDI-lipids experiment in mouse brains with 6-OHDA medial forebrain bundle (MFB) lesion. We specifically identified and analyzed several lipids that were either increased (CL(72:9)), decreased (CL(76:9), GM1(36:1), GM1(38:1)), or unchanged (CL(74:9)) in cPSAP^{DAT} mouse brains. Only CL(74:9) and CL(76:9) displayed a mild decrease in 6-OHDA lesioned mouse brains (Extended Data Fig. 12B, C), which was different from lipid alterations in cPSAP^{DAT} mice. These data suggests that the lipid alterations found in the cPSAP^{DAT} mouse brains are not secondary to a DA neuronal loss. This point has now been added to the **Results** (**Extended Data Fig. 12**) and **Discussion**.

3. The authors should comment on whether they detect full length prosaposin and progranulin or, more likely, a combination of full length and cleaved saposin and granulin peptides, resp., with the immunoreagents they use.

Response: We have now performed experiments showing PSAP antibody specificity in N2a cells with PSAP knockdown by PSAP-siRNA (see following Figure A). As predicted by the reviewer, this antibody detects both full-length PSAP and several saposins. This information has now been added to the **Results** and **Discussion** and are shown in **Extended Data Fig. 7**. The bands shown in the blot below represent different forms of PSAP/saposins, as have been described in both mammalian and insect cells (see following Figure B)³.

Figure A. PSAP antibody specificity validation

Figure B. Proteolytic processing pathways of PSAP in mammalian and insect cells

Regarding PGRN, studies have shown that the antibody (AF2557) we used detects both full-length PGRN and individual granulins^{4, 5}.

4. The relationship between neuromelanin and PSAP levels in human tissue is interesting. Since SapD is a lipofuscin marker and neuromelanin contains some lipofuscin-like material, might the authors be looking at actual lipofuscin within the neuromelanin with the anti-PSAP antibody?

Response: This is an interesting point. In the previous version of the manuscript, we ruled out the influence of lipofuscin autofluorescence by the negative control staining in the human brain sections, where we used a laser intensity that is unable to excite the autofluorescence to image the stainings.

However, to fully address this point, we have now used a much higher laser intensity but still only detected a very weak autofluorescence signal at 594nm. It may be derived from lipofuscin (see attached Fig.A), but it is very unlikely that it corresponds to the PSAP levels (see attached Fig.B).

Fig. A. Autofluorescence in neuromelanin

Fig. B. PSAP staining in neuromelanin

5. The authors should also emphasize how a standard LC/MS-based lipidomic analysis of mouse brain tissues would nicely complement the MALDI imaging approach, and perhaps help resolve some of the lipids that cannot be resolved with MALDI, like BMPs, GlcCer/GalCer etc...

Response: Valid point. While we obtain spatial information of the lipid changes using MALDI-MSI, LC-MS-based lipidomics would be complementary for the analysis of a more comprehensive and high-throughput lipidomics analysis of mouse brain tissue^{6, 7}. Further, when coupled with LC and ion mobility or applied with chemical derivatization strategies^{7, 8}, electrospray ionization (ESI) can provide specific information on the isomeric and isobaric species, which would enhance the biological interpretation of the lipidomics analysis. We have now added this information to the **Discussion**.

References

1. Breiden, B. & Sandhoff, K. Lysosomal Glycosphingolipid Storage Diseases. *Annual review of biochemistry* **88**, 461-485 (2019).
2. Morimoto, S., Martin, B.M., Kishimoto, Y. & O'Brien, J.S. Saposin D: a sphingomyelinase activator. *Biochemical and biophysical research communications* **156**, 403-410 (1988).

3. Leonova, T., *et al.* Proteolytic processing patterns of prosaposin in insect and mammalian cells. *The Journal of biological chemistry* **271**, 17312-17320 (1996).
4. Zhang, T., *et al.* Differential regulation of progranulin derived granulin peptides. *Molecular neurodegeneration* **17**, 15 (2022).
5. Du, H., Zhou, X., Feng, T. & Hu, F. Regulation of lysosomal trafficking of progranulin by sortilin and prosaposin. *Brain Commun* **4**, fcab310 (2022).
6. Köfeler, H.C., *et al.* Recommendations for good practice in MS-based lipidomics. *J. Lipid Res.* **62** (2021).
7. Wang, M., Wang, C., Han, R.H. & Han, X. Novel advances in shotgun lipidomics for biology and medicine. *Prog. Lipid Res.* **61**, 83-108 (2016).
8. Xia, F. & Wan, J.B. Chemical derivatization strategy for mass spectrometry-based lipidomics. *Mass Spectrom. Rev.*, e21729 (2021).

REVIEWERS' COMMENTS

Reviewer #1 (Remarks to the Author):

The authors have satisfied reviewer comments.

Reviewer #2 (Remarks to the Author):

I have no further comments.

Reviewer #4 (Remarks to the Author):

The authors have done a great job addressing all my questions and concerns.

Point-to-point response to the Reviewers:

Reviewer #1 (Remarks to the Author):

The authors have satisfied reviewer comments.

Response: Thank you! We appreciate the constructive comments over the review process by the reviewer.

Reviewer #2 (Remarks to the Author):

I have no further comments.

Response: Thank you! We appreciate the constructive comments over the review process by the reviewer.

Reviewer #3

No comments.

Response: Thank you! We appreciate the constructive comments over the review process by the reviewer.

Reviewer #4 (Remarks to the Author):

The authors have done a great job addressing all my questions and concerns.

Response: Thank you! We appreciate the constructive comments over the review process by the reviewer.